# What advice would I give a starting graduate student interested in robot learning?
# Models! Model-free! ... Both!

Paper for the RSS workshop on Robotics Retrospectives
Chris Atkeson, CMU Robotics Institute, July 2, 2020

The latest version of this paper is at `www.cs.cmu.edu/~cga/mbl`

**Abstract:** This paper provides a personal survey and retrospective of my work on robot control and learning. It reveals an ideological agenda: Learning, and much of AI, is all about making and using models to optimize behavior. Robots should be rational agents. Reinforcement learning is optimal control. Learn models, and optimize policies to maximize reward. The view that intelligence is defined by making and using models guides this approach.

This is a strong version of model-based learning, which emphasizes model-based reasoning as a form of behavior, and takes into account bounded rationality, bounded information gathering, and bounded learning, in contrast to the weak version of model-based learning, which only uses models to generate fake (imagined, dreamed) training data for model-free learning. Thinking is a physical behavior which is controlled by the agent, just as movement and any other behavior is controlled. Learning to control what to think about and when to think is another form of robot learning.

That represents my career up to now. However, I have come to see that model-free approaches to learning where policies or control laws are directly manipulated to change behavior also play a useful role, and that we should combine model-based and model-free approaches to learning.

Everyday life is complicated. We (and robots) don't know what aspects of the world are relevant to any particular problem. The dimensionality of the state is potentially the dimensionality of the universe. The number of possible actions is vast. The complexity barrier for AI is like the speed of sound for airplanes, where we have to punch through it as a field to go supersonic and develop useful robots in our homes, daily lives, and on the road [The Right Stuff, book 1979, movie 1983].

I advocate exploring life-long learning in the context of rich knowledge, such as libraries of past observed, taught, actual, and imagined behavior, rather than focusing on impoverished and isolated learning where a single representation is adapted to do a single behavior once from a single training set, and independent of any larger context. I advocate working with real robots in the real world to address the complexity barrier. Working with real robots is much easier and more fun than trying to figure out how to get simulators to capture the complexity that human learning effortlessly takes advantage of.

*"I just want to say one word to you ... one word ... are you listening? ...* **plastics!***"* (from the movie The Graduate [Nichols 1967]). My word for you in 2020 is "**Models!**" No, wait. it's **Model-free!** ... urr um hmm **Both!**

# 1   Advice to students

New PhD students need to pick a research area that will make a splash in five to ten years. Ideally, you should develop and become known for a new problem or approach. For this reason, I would not advise you to start doing what is hot now (deep X). Look for areas or problems that are being ignored (see the discussion of state estimation for robots below, for example) or will become hot. One way to get inspiration is to find a paper you think is wrong, and try to show why it is wrong, and how the work can be improved. Another source of inspiration is to pick a paper you think is good, and ask yourself how that work could be even better, and how to extend or build on that work (which is what we will do in the first section of this paper).

In addition to reading papers, look at videos and demos (lots on the web). I encourage you to send email to authors. They will typically respond positively to the attention. Ask "How often does the demo work?", "What makes it fail?", "Why is it so slow?", "Does it work every time?", and "Why does it keep making the same mistake?". A metric for how well a robot or demo is working is how much a human is involved or closely supervises it. The more involved or closer the human, the more fragile the demo. This is especially true in robot legged locomotion demos. Ask if the human can move further away from the robot. Look for flaws, and ask "**why?**".

If you want to have practical impact, you should look for areas where it is likely your research will both lead to better performance, and that performance will improve on state-of-the-art performance generated by other approaches, including non-learning engineered performance. The criteria of state-of-the-art performance suggests exploring (domain) **model-based** approaches to control and learning, which use predictions of what will happen to make decisions and generate robot commands. (Domain) **model-free** approaches do not try to make predictions of future situations (or states), but instead keep score of which commands did well or badly in each situation that arises.

This paper focuses on the role of domain-specific continuous-state models, such as models of robot kinematics and dynamics, and models of how objects move when manipulated in a task, in robotics, reinforcement learning (RL), and AI. Domain models support more powerful and efficient reasoning and also support simulation. Ideologically pure domain-model-free robot learning does not learn in simulation, but only using a real robot. Appendix 1 discusses what these words and concepts mean in more technical detail with respect to reinforcement learning. A summary is that in this paper the word model means domain-specific known or learned information that enables prediction of a future state or situation, rather than the broader interpretation of "a model is any internal representation of anything".

The first part of this paper discusses some implications of a recent model-based learning paper that also uses model-free learning techniques applied to a learned model in the form of a simulator [Hwangbo et al. 2019] to see what connections and ideas are inspired. The second part of the paper describes the evolution of my thinking about model-based control and learning, based on my own work. The point of this section is to examine what inspired me, how I chose problems to work on, how robotics, AI, and science and engineering work, the somewhat stumbling and opportunistic nature of making forward progress in research, and what a career in research looks like. The third part of the paper discusses some issues and open questions, to inspire further research and thesis topics. I end with some final thoughts.

This paper is too long. I am like that quiet guy in a bar, sipping away, but when you get me talking I can't stop. You can just read the first and third parts. My heart, and the jokes, are in the second part. Technical details are in the appendices, if you like details and data.

## 2 Inspiration from a paper

A recent excellent example of model-based control and learning is the hybrid approach of Hwangbo, Lee, Dosovitskiy, Bellicoso, Tsounis, Koltun, and Hutter [2019], where engineering approaches and substantial prior knowledge were combined with model learning. Surprisingly accurate models resulted. The most interesting part of this work for me was that model-free learning used the learned model in simulation to plan how the robot should get up after falling from a wide variety of awkward positions sprawled on the floor. The learned model was so good that the policy learned in simulation worked well on the real robot without further learning (in technical terms the Sim2Real gap [Sim2Real 2020] was small).

1) **The more an agent knows, the more and faster it can learn:** In [Hwangbo et al. 2019], learning was used to make a better model of an aspect of the system (actuator dynamics), rather than create a complete model from scratch (*tabula rasa* model learning). It was assumed that the rigid-body dynamics of the robot were already known (which was valid in this case). This decomposition focused, simplified, and sped up learning. Building on a foundation of engineering knowledge is often a route to high performance. Take advantage of what you already know about the system. Don't start learning from scratch. This principle applies more generally. I learned this principle in Patrick Winston's undergraduate AI class at MIT (6.034) around 1980. Expert systems were hot in the early 1980s, and we kept reciting slogans like "Knowledge is Power!" during that era. Later in this paper I will advocate that the field of robot learning should focus on learning with extensive domain knowledge and rich libraries of experience.

2) The paper you are reading now will argue that **planning in advance is an important form of robot learning.** Looking again at [Hwangbo et al. 2019], we see that in addition to learning a better model, plans (in the form of a policy) were computed to get up from many different starting positions. Why is planning in advance a form of learning, if it just involves thinking (which is whatever a brain or computer does), and not physical behavior such as movement, talking, or sweating?

One definition of learning is the improvement of performance from experience or behavior. I believe planning in advance is a form of learning for two reasons: **Better reactions improve performance.** No question about that. Planning in advance such as considering alternative strategies and storing plans, which can be quickly accessed when needed, can improve future performance. **Thinking is a physical behavior.** Hmmm, got to think about that one ...

In AI, learning from thinking has sometimes been called "speedup" learning to distinguish it from other forms of learning [Fern 2017], and relegate it to a lesser status, or maybe it isn't really learning at all. However, thinking is a physical process that plays out in real time. If a robot can't react appropriately or think of a desirable behavior fast enough, too bad. We should recognize the importance of learning from planning or simulation in robotics. It is another form of behavior by an agent that leads to improved performance based on simulated or mental experience. Mental

practice is still a form of practice. Performance improves by making good choices in advance, as well as taking into account any improvements in the learned model by updating all stored plans.

Thinking is a physical process, just like movement. Robots brains are like today's smart phones, where computation has a more obvious cost. Computational resources are limited and computation takes time and energy (measured in Watts) and generates heat which needs to be dissipated, all of which are limiting. Agents must eat, recharge, or refuel to think, as well as move. Computational hardware takes up space and adds weight. These factors change for computation that is moved offboard or to the cloud, but there are still costs and limits for both computation and now communication. Thinking must be controlled as much as movement is. Don't waste time and energy daydreaming! Agents must allocate computational resources, and make choices as to what to think about, and when. In addition to planning to act, we must plan to think.

A more radical argument, which I will save for another day, is that **any precomputation is learning.** In computer graphics and video games it is common to precompute as much as possible and store the results, so interactive animation and games execute in real time. Robotics has the same need to execute in real time. We found that precomputing all possible optimal trajectories for any pattern of stepping stones resulted in sets of policies we could easily represent with one global quadratic function [Kim et al. 2013]. In fact, almost anything we wanted to know based on a set of several hundred thousand plans could be represented accurately with global quadratic functions. In this case optimization simplified a complex range of possible behaviors into something simple. We didn't even have to use a table lookup approach. Just simple global polynomials. I would call that robot learning. We humans also learned something surprising about the domain which might be generally true of sets of optimized plans in many domains.

One of the mysteries of how people think about learning and AI is why generating heuristics or creating a policy for a game you cannot solve is learning (and is hailed as an impressive achievement [Schrittwieser et al. 2019]), but completely solving a game is not considered learning [Wikipedia 2020q]. This does not make any sense. In both cases computation is applied to a perfect model of the game. Another mystery is from control theory, in which adaptive control only applies when a perfect model is not available, so it is possible for a system to learn based on a known model (such as the rules of a game) but not be considered adaptive.

3) This paper will also argue that **library approaches to storing plans are useful:** Library-based learning is trivial: both executed plans with the associated experience and un-executed plans are simply stored in memory. In fact, all experience and the associated robot commands are stored, since those commands produced that behavior. It is important to keep in mind that the simplest form of learning (which is also model-free) is to avoid making the same mistakes over and over again by storing experiences, detecting repeating patterns of behavior, and somehow changing the choices being made. The second simplest form of learning (also model-free) is to recognize the same or similar situations, and re-use a previous behavior. Recognizing similar situations can be limited and based on how a situation looks or other perceptual cues, or it can be powerful where deep causal theories drive reasoning by analogy. Representations that store training data explicitly avoid forgetting and negative interference when something new is learned or when the training data distribution changes. The big challenge is indexing: finding relevant plans to provide a starting point for solving a new problem. This approach has many names, such as memory-based

learning, and case-based reasoning or learning [Richter and Weber 2013]. Experience graphs are another form of a plan library or cache [Phillips 2015].

3A) **We can think of a policy as a library of local plans**, indexed by state, instead of thinking of a policy as representing a single plan and how to deal with likely disturbances to that plan. Even with a single large scale policy, starting from quite different initial states results in different behaviors whose state trajectories are not similar until they approach a common goal state. In [Hwangbo et al. 2019], the common goal state for getting up was the standing position. It makes more sense to think of these behaviors as a set of trajectories, rather than thinking of the different initial states as perturbations of each other [Atkeson 1994, Atkeson and Stephens 2008, Schaal and Atkeson 2010, Atkeson and Liu 2013]. Many of these initial positions cannot be reached from other initial positions without getting up off the floor (lying on one side or another, for example). Alternatively, separate plans and policies could have been created for each sufficiently distinct starting point [Atkeson and Stephens 2008].

3B) **A library of local plans can be used to build a globally optimal policy by making sure the plans agree at the borders:** Just as we can decompose a policy into parts, we can combine the parts into an overall or global policy. Locally optimal policies are easy to create by optimizing greedy policies that always try to go straight to the goal. Locally optimal policies can be combined to form a policy that is optimal over a region of interest. If that region is the entire state space, then the overall policy is globally optimal. If the overall policy and corresponding value function Sutton and Barto [2018] solve the Bellman equation over a region that is bounded by a constant cost-to-goal surface, and some other technical conditions, the overall policy is optimal for that region [Atkeson 1994].

4) **Model-free reinforcement learning is a good way to do model-based planning.** It is still unclear whether model-free reinforcement learning can be directly applied to robots rather than simulations, without large amounts of domain knowledge in the form of highly structured policies with few parameters (for example [Zhang et al. 2017]). Too many samples of the robot's dynamics (states, actions, and outcomes) and too much real time running the robot are required. However, it has become clear that model-free reinforcement learning applied to faster than real time robot simulations (or perfect models of games) can solve difficult planning problems, and outperforms state-of-the-art planning methods in some domains (especially games).

**There will be occasional asides. This is one of them.** The success of model-free reinforcement learning is not magic. This is a form of sample-based planning [LaValle 2006], using a massive amount of search. I am told that each run of training AlphaZero, a "model-free" reinforcement learning program that learns to play games using perfect models of each game) costs a million dollars in computer time. OpenAI spends $5000 every time it trains a policy for its Shadow robot hand. These numbers will decrease as computing gets cheaper, but these costs are a measure of how much computing is involved. If the model-free paradigm prevails, future robot brains will have to be located next to dams to get enough power and cooling to think, and not have unaffordable carbon footprints. Really intelligent brains will not be "in the cloud" or anywhere else on Earth, but in space with kilometer-scale solar panels and spewing out molten material to dissipate waste heat. I was an eager member of the L5 society as a teenager, where we planned to launch lunar material into space to build space colonies using magnetic rail guns. My girlfriend dumped

me because she wasn't interested in living in an orbiting tin can. The dream of space colonies may not happen any time soon, but we will probably have to launch lunar rocks into space anyway to provide coolant for orbiting AIs.

In [Hwangbo et al. 2019], model-free learning was used with the learned model to plan future behavior. Robots need to work in a wide range of states, and manually programming the response to a large number of possible errors and possible initial states is expensive. This is an area where learning in the form of using models to pre-plan rare behaviors is useful. It turns out model-free learning approaches make good offline planners which use models in the form of simulators, if one is willing to pay for a lot of computation [Irpan 2018, Wang et al. 2019]. Unfortunately, model-free learning typically involves too much computation to be used as a real time planner. Many of the great successes of model-free reinforcement learning, such as learning to play games, can be viewed as planning behaviors using a model in the sense of optimal control, and then applying the plan to an actual robot or in a contest with others, rather than learning on the actual robot or during the chess championship.

**Another aside:** Surprisingly, learning is usually turned off when we care about performance. If one is not careful, negative interference in parametric representations such as neural nets can cause a policy to get worse by learning from experience. In a game an opponent can degrade the performance of some approaches by manipulating the distribution of training data by playing badly, for example. Many current approaches to machine learning have to be very careful what data they learn from.

# 3  A personal retrospective on model-based learning

I realize this paper is supposed to be a retrospective, so I thought I would talk about the evolution of my thinking about the role of models in planning, control, and learning, and the lessons I learned along the way. Talks (with videos) that cover some of this are [Atkeson 2017ab 2019]. The utility of models has long been recognized (here is a quote that is older than I am) [Craik 1943]: "If the organism carries a 'small-scale model' of external reality and of its own possible actions within its head, it is able to try out various alternatives, conclude which is the best of them, react to future situations before they arise, utilize the knowledge of past events in dealing with the present and future, and in every way react in a much fuller, safer, and more competent manner to the emergencies which face it."

## 3.1  Why I do what I do

I have been interested in skill learning and model-based X learning (where X is human, motor, robot, or reinforcement) since I started graduate school in 1981. Why? A psycho-babble answer is that I was clumsy as a child and had a difficult time learning to do things. Sports and physical education in school were a disaster. So learning physical skills is impressive to me. To this day I get excited to see a robot working in the lab or the real world. I spend a lot of time watching my robot vacuum cleaners do their thing. I was also jealous of my athletic siblings. One has college running records that have not been broken yet. I credit myself for that, in that I spent a lot of time

chasing him around to beat him up. He eventually became an economics professor, focused on what is fair.

Why do model-based approaches appeal to me? Maybe because I found everything difficult, learning the secret or trick to being able to do something is very appealing. I compensated for my lack of effortless learning by applying more thinking about better ways to do things. I remember vividly times when people told me "tricks": My father told me when I started wearing glasses "Don't leave your glasses on a chair, people are likely to sit in chairs without looking, and break your glasses." When a friend and I were trying to build a structure for a school project and it kept coming apart, my friend's father (an engineer) explained to us that we should use tape so that the likely forces on the tape will pull the tape on to the surface it was stuck to, rather than shear the tape or lift it off the surface. Books on how to juggle provide a sequence of subgoals to shape behavior, rather than trying to learn to juggle 3 balls all at once from scratch. Magic tricks are usually based on a secret, which once you know, perceiving and doing the magic trick becomes easy. My math teacher in high school said I was the laziest person she had ever met, since all I did was look for ways to do homework that minimized my work. It turned out that always looking for better ways to do things has served me well in my career.

Although I have a PhD in psychology, enough psycho-babble! A more intellectual answer is that I had read in many places that the essence of intelligence is learning models, becoming better at predicting what will happen, and not repeating errors.

## 3.2   The context of model-based and model-free ideologies

The notion of model-based control was well known when I started graduate school, and the term "model-based control" appears several times in my papers from the 1980s as well as the book that included my thesis work [An et al. 1988]. At the MIT AI Lab we were explicitly working towards the goal of making model-based robot control practical. My work on learning from practice for my PhD thesis (1986) showed me the role of models in learning [An et al. 1988] and my thesis title was "Roles of knowledge in motor learning". By knowledge I meant models. My first use of the term "model-based robot learning" seems to be in 1988 [Atkeson et al. 1988]. My first use of the term "model-based reinforcement learning" (1997) seems to be a paper that attempted to get the field of reinforcement learning to be more balanced and shift from its fixation on model-free approaches (I failed) [Atkeson and Santamaria 1997]. This was followed up by a paper (1998) I talk about below, which implemented reinforcement learning from practice based on a single human demonstration of the task [Atkeson 1998].

To put this in context, let's look at related fields. In American psychology, there was a civil war (1950s – 1970s) between advocates that the mind is model-free (behaviorists) and those who advocated that the mind both learns models and computes with internal representations using those models (cognitive scientists) [Bargh and Ferguson 2000, Miller 2003, Sinha 2007, Skinner 1985]. Many universities ended up with two psychology departments because of this civil war, one for behaviorists and one for cognitive scientists (first one in 1986 at UCSD), I note that I did my graduate work in MIT's Psychology Department, which became the Brain and Cognitive Science Department while I was there (1986). At MIT, the war was over. Clearly, I was in a hotbed of model-based Cognitive Science that strongly rejected the old-fashioned model-free psychology. Interestingly,

the study of reinforcement learning moved over to AI and machine learning for a while, and only recently has come back to psychology and neuroscience with the discovery of dopamine-related neurons and areas, some of which match the characteristics of model-based reinforcement learning, and others which match model-free reinforcement learning [Sutton and Barto 2018]. It seems both the behaviorists and the cognitive scientists were right.

As a graduate student I was part of a neuroscience group focusing on understanding the biological control of movement and behavior. My advisor, Emilio Bizzi, was known for showing that animals with no sensation or feedback could still move to targets, showing that feedback control was not necessary for biological control of movement, and suggesting that there may be learned commands stored in memory or internal models generating movement [Bizzi 2020]. Emilio went on to explore how motor synergies were controlled and used in biological motor control. Emilio is a strong supporter of computational approaches to understanding the brain, and was a strong influence advocating a model-based ideology. McNamee and Wolpert [2019] provide a recent survey on the role of internal models in biological control.

I also spent a lot of time in the engineering part of MIT. This had a major impact on how I view the world, and engineering approaches to control dominate my thinking. Things that are difficult for folks with a computer science background (such as Kalman filters and Dual Control) are obvious to me, while the same thing from an AI point of view (Bayesian approaches to perception and exploration) are a mystery to me. I also have more awareness of the huge engineering literature that is relevant to issues in robotics and AI, so I often end up saying to students "Take a look at the large literature in engineering on X". In terms of the little things, such as signalling to the world that I am an engineer, I use $\mathbf{x}$ for state and $\mathbf{u}$ for action in my teaching and writing, instead of the AI convention of $\mathbf{s}$ for state, and $\mathbf{a}$ for action.

In control theory the internal model principle, which states that a good feedback controller embeds a good process model, is a strong argument for model-based approaches, and that learning a policy is also learning a model [Conant and Ashby 1970, Wonham 1975]. The field of adaptive control had a debate between model-based (indirect) and model-free (direct) approaches since the 1960s, with the eventual conclusion that they could be made equivalent [Narendra and Valavani 1978], as long as the system being controlled was linear so what state you were in and what goal you wanted to achieve didn't matter. However, the ability of model-based approaches to rapidly adapt to new goals rather than start learning all over again was attractive to me. Indirect (model-based) adaptive control is the source of the meme "**Learn models, compute policies!**".

The internal model principle that good controllers implicitly represent good models suggests another way to think about model-based and model-free approaches. Model-based approaches have or learn an explicit model, and can use it for many purposes such as to make predictions and speed up computation. Model-free approaches also learn a model, but it is implicitly represented in the learned policy and/or value function (actually a Q function) [Sutton and Barto 2018]. Model-free systems can't use the implicitly represented model for other purposes.

The model-free/model-based controversy was unavoidable in the field of reinforcement learning. The model-free position was inspired by behaviorism from psychology. The model-based position was inspired by optimal control from engineering. Sutton and Barto's reinforcement learning text provides good surveys of the history of model-free and model-based reinforcement

learning [Sutton and Barto 2018], as does this review [Kaelbling et al. 1996]. Polydoros and Nal-pantidis [2017] survey model-based reinforcement learning in robotics. Recht [2019] excellently surveys reinforcement learning and strongly advocates model-based reinforcement learning. Bertsekas [2019] surveys the connections between reinforcement learning and optimal control.

I want to highlight that the field of reinforcement learning was shaped by two scientists who also provided funding as program managers, Harry Klopf and Paul Werbos. Harry Klopf had a strong interest in the neural basis of psychological phenomena such as conditioning, and believed that human behavior arose from the cellular physiology of neurons [Wiki 2020, Scholar 2020a]. Klopf had a strong influence on Barto and Sutton (including funding them) [Barto and Sutton 1981, Sutton and Barto 2018], although Barto and Sutton did not model biological neurons but instead worked on "artificial" neurons and neural networks in the early years, before moving to more computational and algorithmic levels of analysis in recent decades (in terms of Marr's levels [Marr 1982], see below). Such neural models are greatly simplified models of biology, and probably missing key aspects of real neurons, but have had a resurgence in the last decade due to the success of deep learning. I would describe Klopf's influence as supporting a model-free ideology of policies being altered directly by rewards and punishments at a mechanistic or what Marr called the "implementation" level. Paul Werbos is a strong advocate for Adaptive and Approximate Dynamic Programming, as well as having invented backpropagation and done a lot of work on recurrent artificial neural networks [Wikipedia 2020k, Scholar 2020b]. Paul supervised NSF funding for many researchers including myself. I would describe his influence as supporting a model-based ideology of optimal control using learned models, and emphasizing computational and algorithmic levels of analysis, despite having a commitment to implementing algorithms in neural networks and even special purpose neural network hardware.

**Yet another aside:** When IBM came out with special purpose hardware that more accurately emulated biological neurons including "spiking" or action potentials, there was great irony in Yann LeCun [Wikipedia 2020g] criticizing that hardware as not being useful because it did not speed up the use of the cartoonish and definitely non-biological artificial neural networks that powered deep learning. Yann had been talking about biological inspiration of his work for 20 years, and was a major presence in the Conference on **NEURAL** Information Processing Systems (now NeurIPS). Artificial neural networks were no longer being inspired by biological neural networks, but had attained a usefulness independent of biology. The field of artificial neural networks had left biological neural networks and its roots behind. Nobody cared any more how the brain worked.

Chris Watkins, who merged the behaviorist and optimal control ideologies, discussed "model-based learning" and "primitive learning" in his thesis [Watkins 1989]. Later, he replaced the term "primitive learning" with "model-free reinforcement learning" [Watkins and Dayan 1992], but it is not clear Watkins considered model-based learning as a form of reinforcement learning. Barto, Sutton, and Watkins attributed the term "model-based" in the context of sequential decision making to the engineering literature on adaptive control of Markov processes [Barto et al. 1989]. A model-based approach to learning involved learning a model, and then computing a policy using methods such as dynamic programming. Barto et. al. refer to adjusting a parametric policy in the absence of a model as a direct approach, borrowing terminology from the field of adaptive control:

> "... *Our concern is with other approaches to learning how to solve sequential de-*

*cision tasks, which we call direct approaches. Instead of learning a model of the decision task, that is, instead of estimating state transition and payoff probabilities, a direct method adjusts the policy as a result of its observed consequences. Actions cannot be evaluated unless they are actually performed. The agent has to try out a variety of decisions, observe their consequences, and adjust its policy in order to improve performance. We call this process reinforcement learning after Mendel and McLaren [Mendel and McLaren 1970] who describe its relevance to adaptive control.* " [Barto et al. 1989]

In Mendel and McLaren's view, reinforcement learning was about increasing and decreasing the probability of executing a behavior, which was controlled by a probabilistic policy.

*"By reinforcement learning we will mean the process by which the response of a system (or organism) to a stimulus is strengthened by reward and weakened by punishment. In the former situation, positive reinforcement occurs and increases a response probability, whereas in the latter situation negative reinforcement occurs and decreases a response probability."* [Mendel and McLaren 1970]

Sutton also emphasized the relationship of reinforcement learning to direct (model-free) adaptive control, and thus took a position that reinforcement learning is (or should be) model-free in [Sutton et al. 1992].

The possibility of **using a model to assist in adjusting a policy**, rather than computing a policy, was not acknowledged then and is rarely acknowledged now, except for the weak version of model-based learning where a model is just another source of training data. I believe that simultaneously learning models and improving or learning policies with the aid of models is an important future direction for optimal control and reinforcement learning. In thinking about this, I think the emphasis in computer science and reinforcement learning on problems with discrete states and actions is partly to blame. It is said that roboticists from computer science want to do discrete search and sorting, and roboticists from engineering want to do calculus and take derivatives. Robotics is not like games with discrete board positions and discrete moves. The success with AlphaZero is not predictive of success in robotics. Perhaps because of the lack of emphasis on continuous state and action problems where derivatives can be used instead of exploration to rapidly compute updates across an entire local region, there has not been a lot or recognition of what known or learned global or local models can do for reinforcement learning.

Since Barto and Sutton were immersed in the study of conditioning and reinforcement (reinforcement learning from the point of view of psychology), it was a small step to using the term "reinforcement learning" to apply AI and robotics [Sutton and Barto 2018]. I note that Sutton's thesis was entitled "Temporal Credit Assignment in Reinforcement Learning" [Sutton 1985], reflecting Minsky's focus on credit assignment as an important issue in AI and Minsky's use of the term reinforcement learning [Minsky 1961]. Sutton's thesis has a nice discussion of the origin of the term "reinforcement learning". Work in psychology focused on associative conditioning: context dependent temporal associations of stimuli and appropriate responses. The version of reinforcement learning studied in psychology often trivialized control problems, for example, to a binary choice such as a rat choosing to go left or go right in a T maze. Reinforcement learning

was not used to address the question of improving physical skills from thinking and practice, as it is now in robotics and machine learning. Improving performance of physical skills was what I wanted to know how to do.

The evolution of Barto and Sutton's thinking on what learning is is also interesting. In one of the early (1981) reports to funders, they say:

> *"... it is misleading to view all adaptation and learning tasks as function optimization tasks."* [Barto and Sutton 1981]

but also

> *"It is very clear, however, that many of the prime examples of adaptation in nature involve extremum search."* [Barto and Sutton 1981]

Looking at their body of work [Sutton and Barto 2018], I would say their point of view has shifted over time towards "reinforcement learning is optimization". Here are some quotes from the 2nd edition of their book:

> *"We define a reinforcement learning method as any effective way of solving reinforcement learning problems, and it is now clear that these problems are closely related to optimal control problems ... Accordingly, we must consider the solution methods of optimal control ... to be reinforcement learning methods. ... it feels a little unnatural ... On the other hand, many dynamic programming algorithms are incremental and iterative [like learning methods]."* [Sutton and Barto 2018]

In this early report, they acknowledged that evolution and much animal behavior are optimization processes [Barto and Sutton 1981]. However, they were concerned that in situations where the environment had state that the agent was not aware of, the optimal behavior would seem to vary and change over time. They were also concerned that there was additional information in the environment that would be useful to the agent beyond the features known to the agent. At the time this appeared to make optimization-based generation of behavior less relevant to understanding animal behavior. In retrospect, I believe their view was incorrect, but these issues do present challenges even now. Both of these problems have to do with defining the state vector and then estimating it, described below as still a major problem for reinforcement learning and robot learning in general.

So where did the term "model-based reinforcement learning" actually come from? When did model-based methods become accepted as a form of reinforcement learning? An early instance of model-based reinforcement learning is Barto and Sutton's work on "internal models". Sutton and Barto describe the sources for their thinking about internal models, and an approach where an internal model predicted rewards for simulated behavior to generate "fake" or "virtual" training data for a policy [Sutton and Barto 1981, Sutton and Pinette 1985], similar to Dyna which was presented a decade later [Sutton 1990 1991]. I will refer to the use of fake training data as the "weak" version of model-based reinforcement learning, where any models are just used to augment real training data with fake (virtual, imaginary, dreamed) data. Examples include [Andersen et al. 2018, Andrychowicz et al. 2017, Feinberg et al. 2018, Gerken and Spranger 2019, Gu et al. 2016, Ha and Schmidhuber 2018, Kalweit and Boedecker 2016, Pascanu et al. 2017, Piergiovanni et al. 2018,

Thabet et al. 2019, Weber et al. 2017, White 2018, Zhong et al. 2019]. There is no use of models to more efficiently compute or improve learned value functions or policies (see Appendix 1 for a discussion of how models can accelerate reinforcement learning). [Bertsekas 2019] categorizes weak model-based approaches as model-free. It is interesting to compare Sutton's views in [Sutton et al. 1992] and [Sutton 1991], one of which advocates the use of models in reinforcement learning, and the other does not. A recent talk by Sutton which describes his current thinking on model-based reinforcement learning is [Sutton 2019].

Andrew Moore's thesis was about using models in robot learning, and has a nice survey of the roots of this approach [Moore 1990]. The distinction between using inverse dynamics models which directly compute an action and using optimal control or reinforcement learning to design policies that optimize a criterion was not clearly made. Moore, Schaal, and Atkeson (1992-4) clearly articulate "Learn models, compute policies", the strong version of model-based reinforcement learning, and almost actually say "model-based reinforcement learning", except they use the name of a particular type of model instead of just the word "model": "memory-based reinforcement learning" [Moore and Atkeson 1992 1993, Schaal and Atkeson 1993 1994]. Many later papers refer to Sutton's Dyna papers and these papers when introducing the concept of model-based reinforcement learning.

The earliest use of the term "model-based reinforcement learning" I have found is in 1994 by [Tadepalli and Ok 1994]. By the late 1990s the use of the term "model-based reinforcement learning" was firmly established [Forbes and Andre 1999, Kuvayev and Sutton 1997, Mahadevan 1996, Precup and Sutton 1998, Precup et al. 1998, Szepesvari and Littman 1996]. Gordon and Atkeson organized a workshop on "Modelling in Reinforcement Learning" at ICML in 1997. Surveying what has happened since then in model-based reinforcement learning will have to await another paper, as I am running out of space and time. In the mean time, these are useful recent surveys [Polydoros and Nalpantidis 2017, Recht 2019].

> *"Model-based methods rely on planning as their primary component, while model-free methods primarily rely on learning. ... planning uses simulated experience generated by a model, learning methods use real experience generated by the environment. ..."* [Sutton and Barto 2018] (Q-learning applied to simulated data becomes Q-planning.)

> *"Our choices [of which applications to cover] reflect our long-standing interests in inexpensive model-free methods that should scale well to large applications."* [Sutton and Barto 2018]

Model-free approaches have dominated research on reinforcement learning. These days, students who are interested in learning want to work on model-free deep learning. The position that model-based reinforcement learning is not reinforcement learning, or learning at all if the model is known, was widely held [Woergoetter and Porr 2008]. It is interesting to speculate why. The notion of an agent learning from practicing a task with little domain knowledge (such as an agent locked in a room with a chess board and a rule book) is attractive to many people. The appeal of "something from nothing, with no work on my part (the computer does all the work)" is very powerful. We like to think that we, being as intelligent as anyone else, can figure things out. We tend

to underestimate the role of expertise, cultural knowledge, and what we already know. Later in this paper I advocate studying learning in its natural habitat, surrounded by rich knowledge about the world.

> "The role of models in RL remains hotly debated. Model-free methods ... aim to solve optimal control problems only by probing the system and improving strategies based on past rewards and states. Many researchers argue for algorithms that can innately learn to control without access to the complex details required to simulate a dynamical system. They argue that it is often easier to find a policy for a task than it is to fit a general-purpose model of the system dynamics ..." [Recht 2019]

In many situations policies may be less complex than models. However, using models can greatly accelerate learning even a low complexity policy. I talk about model-based policy optimization in Appendix 1. If you are using a simulator, you have already done the work to make a complex model. Why not use it more effectively?

The debate between strong AI (knowledge-based and focused on a particular domain) and weak AI (domain independent methods) continues today, between advocates of using large amounts of domain-specific knowledge and representations, and advocates of more general AI. While I was at the MIT AI Lab, there were two schools of thought. One was to turn AI into applied mathematics and for robotics, mechanical engineering. This view encouraged careful analysis and trying to take advantage of the physics of the world, and making models. David Marr articulated this point of view [Marr 1982]. The right way to do AI was to first understand the problem in terms of computational theory, which described what inputs were useful, what needed to be computed, and what the outputs should be. One should use that understanding to address the next level which was algorithmic and representational, specifying the information processing to be done, independent of the computational hardware on which the algorithms were going to be implemented (brains, analog circuits, artificial neural networks, serial computers, dark matter gas clouds, ...). Only after sorting out the computational theories, algorithms, and representations was one allowed to address the last level, implementation, which mapped the algorithms to available computational hardware.

Marr had a huge impact on me. He was at MIT, and had died the year before I came to MIT as a graduate student. At MIT, Marr was considered a prophet and his book was our bible. He had defined the new and correct paradigm to both study the brain and build robot brains. His paradigm embraced model-based approaches. Note that this school of thought was a rejection of both neural networks and symbolic AI of the 1960s and 70s. Symbolic AI was felt to have no principles, and was just one hack after another. Every instantiation of interest in neural networks (1950s, 1980s, 2010s, note the 30 year periodicity) focused first and last on mechanism and implementation. I was keenly aware of the Kuhnian battle of paradigms [Wikipedia 2020e].

**An aside:** For those who like conspiracy theories, it is startling to find out that Marr, Frank Rosenblatt (Perceptrons), and Kenneth Craik (see quote above) all died young. Perhaps there are things the space aliens among us don't want us to know. Those of us who are still around are not on the right track.

**Another aside:** The AI Lab doing mechanical engineering annoyed the mechanical engineers at MIT, who felt they were doing mechanical engineering just fine, and resented a bunch of computer scientists stumbling around and rediscovering their field, and getting a lot of good press doing

it. As now, AI was hot and the wave of the future, and applied math (vision), mechanical engineering (robotics), and statistics (learning) were boring and old fashioned. I can now feel their pain, as I feel the same about the deep reinforcement learning folks, who are invading robotics. Now knowledge and model-based approaches are out of fashion, and the know-nothingism [Wikipedia 2020d] of deep reinforcement learning is hot hot hot! Just wait for the next swing of the intellectual pendulum, however.

The other school of thought at the AI Lab was led by Rod Brooks, whose message was "When we examine very simple level intelligence we find that explicit representations and models of the world simply get in the way. It turns out to be better to use the world as its own model." [Brooks 1991]. Note the caveat of "very simple level intelligence". That the world is its own best model is a true statement reflected in current work on the Sim2Real gap [Sim2Real 2020], In the end, Rod rejected models, representations, abstraction, symbols, and modular design approaches [Brooks 1990 1991], all of which I advocate and are now practical. I do agree with Rod's position that we must be committed to evaluating our work in as complete a system as possible with physical inputs and robots for outputs, rather than only measure the performance of an isolated module that we are not even sure we need acting on carefully constructed inputs (the benchmark!) and often only in crude simulations.

An interesting question is "When do explicit representations and models start becoming useful?" Does it have something to do with what is computed and its reliability, or is it simply a reflection of the cost of current technology (sensors and computers)? It is interesting to note that the debate between model-free and model-based approaches played out in the robot vacuum cleaner market over the last 20 years. In 2002 the company iRobot founded by Rod and members of his group came out with the Roomba, a model-free robot vacuum cleaner that used randomized actions to make full coverage of the floor likely. Since fewer sensors and less computation were needed, this robot wiped out attempts to commercialize model-based robot vacuum cleaners, whose price was about ten times Roomba prices. Millions of model-free Roombas have been sold and continue to sell, and model-free Roombas continue to be the most common robot on the planet. However, in the 2010s, prices for sensors and computers for computer vision and LIDAR decreased sufficiently for model-based vacuum cleaners to succeed in the market, and now even iRobot produces model-based vacuum cleaners (their highest-end and most expensive models). Model-based vacuum cleaners are more efficient, seem more intelligent, and are more fun to interact with in terms of looking into the robot's brain and seeing its model (a map of your house) and what the robot is thinking and doing. You can do brain surgery on the robot remotely with your phone! Now iRobot is having your robot vacuum share knowledge with your robot mop, which is difficult to do if all you represent are policies and Q functions. The natural things to share are maps and state estimates (Where are the obstacles, where am I, and where are you?).

I note that modular approaches to computer vision continue to be common, and a major objection to current work in deep learning is that it is not modular. I expect symbolic AI to come back in the form of abstraction, as discussed below [Atkeson 2020d]. Another note on the evolution of people's thinking: One could argue that Rod's message in the 20th century was similar to the behaviorist's position. Interestingly, in the new Millennium, Rod co-founded an AI company, Robust.ai, with Gary Marcus, a (radical) cognitive scientist [Pontin 2019]. The Apocalypse

is nigh!

Even computer animation has the debate between model-free and model-based approaches. Model-free approaches include manually programming behavior, or copying human behavior (with motion capture, for example). There is a model-based contingent in computer animation who actually simulate physics to animate objects, and develop controllers to generate character motion, just like robotics.

What we learn from economics is that robots should be rational agents, which optimize. This does not mean they are all-powerful. In 2000, I moved to CMU, where one of the revered figures is Herb Simon. Simon got a Nobel Prize for work on bounded rationality and satisficing (1978). Simon pointed out that people do not behave like rational agents with infinite computational power, who enumerate and fully model all possible sequences of actions. Instead, they seem to enumerate some alternatives, and then pick one which is good enough. They limit the amount of computation they do. This work is often understood as a rejection of optimization: people don't optimize. This is incorrect. If computation has a cost, it makes perfect sense to optimize the sum of physical and mental costs. One of the points I am trying to make in this paper is that computation has a cost, and a nuanced version of "robots should be rational agents" would take that into account. Later I discuss Dual Control Theory, which I also learned about during graduate school [Wikipedia 2020n]. Dual Control Theory puts a cost on ignorance, and tells us what to learn and how much to learn (bounded learning in addition to bounded rationality), based on models of what an agent does and doesn't know, and how that affects future rewards. A rational agent doesn't bother to learn about things that don't increase rewards. So much for universities.

Chaos theory raised doubts about model-based approaches, because although chaotic systems were easy to model, it was difficult to integrate the model through a local instability and make accurate long term predictions [Wikipedia 2020l]. This is true. However, once a controller is added to a chaotic system, typically the local instabilities, which are often a fragile feature of the system, are controlled away without much force or effort, and the system is no longer chaotic. Chaos theory (and fractals) turned out to be a distraction.

**Maybe this is an aside:** It is interesting that model-free ideology reverberates in American culture and politics, contrary to cultures more respectful of teachers, scholars, and other "experts". The appeal of model-free approaches is a form of technological populism. Anyone can do it. You don't need any domain knowledge or expertise. In fact, expertise just gets in your way. In American mythology the common people can fix anything. They don't need fancy degrees or experts. Our founding fathers were farmers, as was Abe Lincoln, who was largely self-educated. Our technological heros are often self-educated in terms of domain expertise. The founders of Microsoft and Facebook dropped out of college. Edison, importantly not an academic, famously used trial and error to invent the light bulb. Some forms of American exceptionalism, including a "can-do" attitude and Joe Biden constantly saying "America can do anything we set our minds to" are manifestations of this ideology. Our presidents are generally not technocrats, and one who was, Carter, is not viewed highly for it. Our heros do not have PhDs. In the 1850s there was an antecedent to the Republican party (after the Whig party collapsed) known as the "Know Nothings" [Wikipedia 2020d]. Part of their ideology was a rejection of elites and experts. Interestingly, we see an echo of the Know Nothings in today's politics, again in the Republican party and again rejecting elites,

experts, academia, and education. Hmmmmm. One way to view model-free ideology is as another instantiation of know-nothingism.

## 3.3  Traditional model-based approach: identify (learn) a model and use it

So that was the intellectual context in which I started out. When I entered graduate school in 1981 there was evidence that humans had internal models of some aspects of their movement system (in psychology and neuroscience this is called the "motor" system, which is confusing to engineers). The first question I asked as a graduate student was "Do humans use models?" and more specifically, "Do humans compensate for arm and load dynamics?" It turned out that no matter how I asked subjects to move at different speeds or carry different loads, the movements had the same path and speed profile, an almost symmetric in time bell shape [Atkeson and Hollerbach 1985]. This suggests that humans compensate for how forces change with speed and load. In fact, it was difficult to get subjects to change their velocity profiles. The only subjects who could make a significantly asymmetric speed profile were those who had had martial arts training. **Lesson 1: It appears that humans have internal models that predict what forces to apply with different movement speeds and loads.** This left me with the question "How do internal models work, and how are they learned?"

The early 1980s was the model-free era of robot control. Although the internal model principle was well known in control theory that "Every good regulator of a system must be a model of that system" [Conant and Ashby 1970], it was widely believed that model-based control was not practical for robot arms, much less full humanoids that had many more degrees of freedom and no fixed base (we were able to make and use models in real time for humanoids in the early 2010s). Problems for robot model-based control included: 1) **The equations were too complex to use in real time.** Researchers used to symbolically derive the dynamics equations for their robot arm using recently developed symbolic math programs (such as Macsyma, an AI accomplishment that was useful), print the equations out, and count the pages, which usually were in the hundreds. It was obvious that contemporary computers could not do that many computations in real time. Maybe early robotics was led astray by the poor performance of symbolic math programs on identifying common sub-expressions and detecting the recursive structure of robot dynamics on serial link arms. 2) **We would never know the model parameters accurately enough** (mass, location of the center of mass, and moment of inertia tensor for each link, Coulomb and viscous friction, and any parameters associated with the actuators and transmissions). Even if we could compute the dynamics fast enough, we wouldn't get it right, because the model parameters would be off. 3) **We don't need models of the full dynamics:** This is actually true for most industrial robots. 3a) Maybe all those pesky Coriolis and centripetal terms only matter for super fast motions (not true). 3b) Maybe robots move so slowly that only gravitational forces matter (this restricts robots to pretty slow movements). 3c) For highly geared robots (gear ratios above 100) with practical transmissions such as harmonic drives, **the only dynamics that matter are individual joint motor/transmission inertias and transmission friction.** Gravity compensation, compensating dynamic coupling between joints, or a full rigid-body model of the robot dynamics are not needed. For high gear ratios the loads and consequent forces and torques due to anything beyond the transmission become too small to matter relative to the forces and torques due to motor/transmission

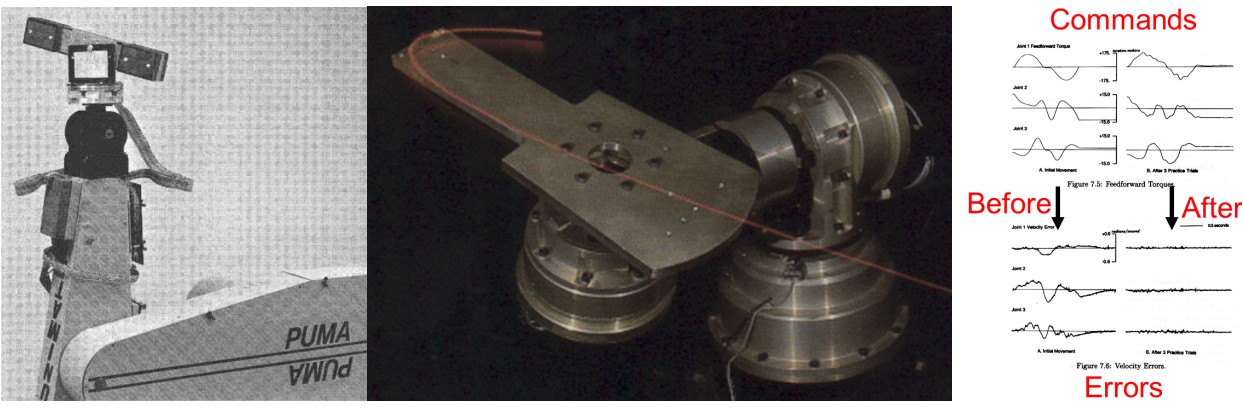

Figure 1: The robots I started with (left to right): 1) The PUMA arm used to test whether we could estimate the inertial parameters of a load. The load was bolted to a six-axis wrist force-torque sensor. 2) The 3 link direct drive robot arm we used to test learning a model, model-based control, and learning a trajectory from practice. 3) A slide showing the results of learning a trajectory from practice using the 3 link direct drive arm after 3 trials.

inertias and transmission friction. Robots at that time were controlled with high gain (stiff) PID servos (feedback gains for position, velocity, and integrated error feedback). Because this is an outcome of the transmission physics, it is still true today, and current high gear ratio robots still do not typically use model-based control using full coupled rigid-body dynamics. However, direct drive and low gear ratio electric robots, and hydraulic robots that do not have gears, all benefit from model-based control using full models of the robot (and actuator) dynamics.

The recursive rigid-body dynamics formulation was a breakthrough that allowed robot arm inverse dynamics to be computed in less than a millisecond in the early 1980s (computers are now a million times faster) [Hollerbach 1980]. Problem #1 was solved! Since the AI Lab had built a direct drive robot arm (Figure 1) to explore model-based control, we were now faced with problem #2: how do we identify the parameters of the model? This was the starting point for my PhD thesis.

I also wanted to understand how I could tell that milk was frozen in the refrigerator soon after I lifted the container. As a graduate student, I had a crappy refrigerator. If the sensed movements of the handle and forces on the handle of the container were consistent with a rigid body, then the milk was frozen. To figure that out, I could estimate the parameters of a rigid-body model for what I was holding on to, and see how well a learned rigid body dynamics model fit the data. I tried estimating load inertial parameters using a PUMA arm with a wrist force-torque sensor and a rigid load bolted to it (Figure 1).

I and others realized estimating dynamic parameters of rigid-body models was actually easy to do, if we reparameterized the problem by replacing the location of the center of mass parameters for a link (or handheld load) with a new set of parameters which were the product of the link mass and the location of the center of mass, and expressed the dynamics with respect to a known reference point fixed on the body, such as the reference point of a wrist force/torque sensor. Once all the inertial parameters were proportional to link mass, they all appeared linearly in the rigid-

body dynamics, since rigid-body dynamics are an elaboration of force = mass*acceleration (f = ma), and mass appears linearly in that equation. If the kinematic equations for the accelerations could be computed, finding the corresponding inertial parameters could be done using simple linear regression from Statistics 101 [An et al. 1988]. **Lesson 2: Reparameterizing models can make learning easier.** The features you use to express your model matter.

A remaining problem was that there were a lot of parameters: more than ten parameters for each link (depending on the friction model used). To identify all these parameters, each link had to be driven in more than ten different ways that were independent of any other links' movements. For robot arm links, close to a fixed base, this is impossible, because there are only a few joints between that link and the base. One could take the robot apart and identify a model of each link separately, but that does not achieve the goal of learning a model from actual robot movements. The corresponding parameters are thus unidentifiable from normal use data. Other parameters were hardly identifiable, due to the magnitude of the forces and torques that could be applied at the robot joints. Things looked bad for my graduation. Then I realized that if I couldn't identify parameters accurately, I didn't need accurate estimates for those parameters because they either don't matter at all, or matter very little [An et al. 1988]. **Lesson 3: If you can't learn particular parameters, relax, they don't matter.** The reason they don't matter is the same reason they are hard to learn: they have little effect on the forces and torques used to drive the robot.

**Yet another aside:** Although my thesis was all about statistical approaches to making models, I failed the statistics part of my PhD qualifying exam, as it focused on experimental psychology. The faculty let that slide. I also failed a standardized test on math in 8th grade, and the school threatened to hold me back. I avoided that by pointing out that I would carry a calculator around for the rest of my life, just like I wear glasses. I failed to turn in my undergraduate thesis, saying I would do it later, which I never did. Graduated summa cum laude. I thought re-publishing conference papers as journal papers to rack up publications was stupid, so I didn't do it, but I got tenure anyway (twice). I seem to have a career marked by evading requirements and getting away with it. My advice to students: you have much more power than you think you do to control your life. You are surrounded by people who want to help you succeed, and are willing to cut you some slack.

## 3.4   Modeling errors

The last problem I had to solve in order to get my PhD was to deal with imperfect models in the form of model structure errors, situations in which the model structure was missing important terms or effects. In this case there is no set of model parameters which fit all the training data, and the estimate of the parameters is different depending on the training data distribution. In control theory model structure errors are known as unmodeled dynamics. For example, our direct drive robot arm motors had cogging torques, which were not included in our model structure. Neither were the dynamics of the power amplifiers that turned information-level requests for a particular overall current into large currents flowing through each phase of a motor. The motors had large inductances and the power amplifiers were limited to their power supplies of 100V, so they often saturated and it took time for the currents to change. In addition, friction models are often too simplistic, and one can spend an entire career modeling friction accurately. You can see

the prediction errors made by a learned model for our direct drive arm in Figure 1-right. How could I use machine learning to avoid a career in electrical engineering or tribology, neither of which I was interested in?

Before I describe how I handled modeling errors, I want to recount some more recent experiences. I was involved in the DARPA Learning Locomotion Program (otherwise known as the Little Dog program). Several teams were given identical robots and identical carefully engineered test setups, and there was another identical robot and test setup DARPA used to evaluate our work. No matter how good our performance was on our own robots, all teams did poorly when our software was applied to another team's identical robot and test setup, as well as when we were evaluated by DARPA. What was wrong? Clearly there was sufficient variation among the "identical" robots, and our software was highly tuned to our own setups (often called overfitting to a particular robot). **Lesson 4: The model will never be perfect**, and **Lesson 5: Avoid overfitting to a model**, or taking advantage of "brittle" features of a model that are not likely to be exactly replicated in reality or stable across time.

No team had a good way to design control and learning algorithms that were robust to that small amount of variation, without having access to the other robot. **Lesson 6: We need to develop control and learning techniques that are robust to small modeling errors**, which should be expected.

Another approach is to rapidly adapt to the new robot [Clavera et al. 2018a]. **Lesson 7: We need to develop control and learning techniques that can rapidly compensate for small modeling errors**, before the robot falls or crashes. I find that when I go to a talk and I have trouble understanding the speaker's accent, within a few minutes I get much better at it. This is an example of rapid adaptation.

I was also part of a team that participated in the DARPA Robotics Challenge (DRC), using the high performance bipedal robot Atlas built by Boston Dynamics. At a DARPA meeting, all the teams using an Atlas, and even the folks from Boston Dynamics, admitted that none of us could estimate the location of the center of mass (COM) of the robot with less than 2cm uncertainty (which is quite large, and the center of mass plays an important role in balance) [Xinjilefu et al. 2015]. Furthermore, from one day to the next, the robot dynamics seemed to change. The DARPA program managers were horrified, and wanted to know what was wrong with the rigid-body dynamics they learned in high school. They also wanted to take our contracts (money) and robots away, since we were obviously incompetent. What they did not appreciate was that any number of the following could be true (including effects I haven't thought of): the measurements of joint torque were not perfect; the joint axes were not exactly the same as what the model assumed; there were significant actuator dynamics; the robots were not made up of rigid bodies, the deformations depended on the load, and the deformations changed the relationships of the joint axes as well as the rigid-body parameters; the robots were hydraulic and the hoses were definitely deformable; and there were thermal effects on the control valves, oil flow, and friction, and the robots heated up as they operated, so their temperature (which varied over the body) depended on elapsed run time and the weather. **Lesson 8: There are always model structure errors** (things you don't know you don't know), and **Lesson 9: It gets more and more expensive in time and effort to track down every effect and improve a model.** Additionally, control theory warns us that **Lesson 10: Arbi-**

**trarily small deterministic modeling errors can lead to arbitrarily large performance errors in adaptive control and learning** [Hussain 2017] and **Lesson 11: More learning cannot eliminate model structure error**. We can push our ignorance to higher and higher frequencies, but we cannot eliminate it. It is widely understood in control theory that from a frequency domain point of view, models will never be good at high frequencies [Wikipedia 2020c]. Real life physical systems have finite power limits, and thus transmit frequencies less as the frequency increases, The higher the frequency response you want to model (more bandwidth) the more power you have to drive the system with to accurately measure a response. Beyond just being able to measure a frequency response, the dynamics change at high frequencies due to thermal, non-rigidity, electromagnetic, chemical, and other molecular-level effects, so even if you had almost infinite power (Watts) to do system identification experiments, the model you identified would soon be out of date. This is why computer CPU performance is becoming thermally limited, and high performance audio speakers and microphones and terahertz electrical circuits and radar are so expensive. The work on the DARPA Robot Challenge resulted in me re-learning a previous lesson: **Lesson 12: Model-based approaches need to handle modeling error.**

## 3.5   Handling modeling errors: use models to correct commands

The paradigm I chose to focus on to deal with modeling errors in my thesis work was to have a robot arm try to do a known desired trajectory that it would learn from practice. I assumed a planner had produced a desired target point or trajectory, and my goal was to get the robot to achieve it as well as possible in a small number of practice trials. This is known as "error correction" learning. On each attempt, the robot would use data from previous attempts to refine the commands to the robot actuators. This paradigm greatly simplified learning by reducing the dimensionality of what had to be learned: all the robot needed to learn was a one dimensional array of actuator command corrections indexed by a variable that represented time from the start of (or phase) of a behavior.

I needed a learning operator that would map past errors into actuator command corrections. It turned out that the learning operator for error correction should be the inverse of the dynamics that mapped actuator commands to trajectories [An et al. 1988]. **Lesson 13: In learning to obtain a goal or execute a desired trajectory from practice, the best learning operator for error correction is a model of the system dynamics.** Hence, the title of my thesis was "Roles of knowledge in motor learning". If I had been in an engineering department instead of a psychology department, my thesis title would have been "Roles of knowledge in robot learning".

One flaw in my approach to learning from practice was that models are always imperfect, and thus **Lesson 14: Learning operators, like models, are always imperfect.** (In this paradigm, if the robot had a perfect model, it wouldn't need to learn ...) As mentioned previously, it is hard to make models accurate at high frequencies [Wikipedia 2020c]. My learning from practice approach turned out to amplify high frequency modeling error, so after a few trials learning introduced high frequency noise into the learned actuator commands. The physical robot arm did not respond much to high frequency inputs, and thus the inverse model had very high gains for high frequencies. The simple fix was to low-pass filter the actuator commands. From a theoretical point of view, the best learning operator turned out to be a model of the system, modulated by the model uncertainty which increases at higher frequencies. **Lesson 15: Explicitly representing model uncertainty**

**and using it to modulate learning helps.** Figure 1-right left column shows the initial commands and errors for our 3 link direct drive robot arm, while the right column shows the commands and errors after 3 practice movements.

## 3.6   Optimal control approaches to handling modeling errors

In later work, this approach to learning from practice (error correction learning, which is similar to root finding) was replaced by an optimization approach (extremum finding) which minimized both the trajectory error and the actuator commands (effort), a form of reinforcement learning [Atkeson and Schaal 1997a]. I was now doing model-based reinforcement learning! It took me a long time (more than a decade) to appreciate that the inverse dynamics approach to control popular in robotics was not robust. Optimization-based approaches, which penalized effort, were much more robust.

In addition to helping deal with high frequency modeling errors, penalizing effort in the optimization prevented actuator commands from "blowing up" due to other types of modeling errors. For example, when your system is non-minimum phase (cart-pole, unicycle, steering a bicycle) and the system has to move away from the goal in order to move towards it later. Systems that have to balance are often non-minimum phase. Which direction does the cart move, when the pole is vertical, to later balance the pole somewhere to the right? The cart first moves left, away from the goal, to tilt the pole to the right, so it can move to the right without knocking the pole over. It the cart does not move to the left relative to the center of mass of the pole, there is no way to keep the pole balanced. **Quiz:** Which direction do you steer a bicycle to turn to the right? **Answer:** You initially turn the bicycle to the left, so the bicycle is tilted to the right, and you can turn to the right and remain balanced. Error correction learning without a good model often blows up on non-minimum phase systems, since commands have to be corrected before errors appear (the inverse model is not causal) [An et al. 1988].

In comparison to model-free reinforcement learning and some forms of model-based reinforcement learning [Sutton 1991], where a policy is represented by a function approximator and is learned, I have explored reinforcement learning approaches where a model is learned, and the policy is computed, usually in real time. **Lesson 16: Learn models, compute policies.** Techniques for computing policies came from optimal control. Many people don't think this version of model-based reinforcement learning is actually *reinforcement* learning, but something else. I disagree.

We found it was necessary to add another term to the optimization criterion, which allows violations of the learned model in the early stages of learning when the learned model is not accurate in some regions of state space [Atkeson 1998]. We found that trajectory optimizers that are fully consistent with the learned model often have difficulty finding reasonable plans at that stage. Trajectory optimizers estimate derivatives to choose an optimization step. This process amplifies inconsistencies in the learned model. In addition, optimization-based planners choose commands that take advantage of modeling errors that reduce cost, which also amplifies modeling errors. This is a form of "maximization bias" [Sutton and Barto 2018]. We proposed that a planner should not be entirely consistent with the learned model during model-based reinforcement learning. Trajectory optimizers that balance obeying the learned model with minimizing cost (or maximizing

reward) often do better, even if the plan is not fully consistent with the learned model. Effectively, we are biasing the optimizer to oversmooth the trajectory in a way that favors reducing cost. This is a version of the "optimism-in-the-face-of-uncertainty" (OFU) principle. Violations of the model are gradually reduced as the model accuracy improves with more training data. This optimization approach avoids integrating the imperfect learned model in time to generate a trajectory, which allows errors to build up, but instead hypothesizes a trajectory represented by spline knot points, and penalizes violations of the learned dynamic model. **Lesson 17: In areas of state space where a learned model is not yet accurate, an optimization-based planner should downweight obeying that model in favor of optimistic planning that minimizes cost. Lesson 18: Estimates of local model accuracy are critical to this approach.**

Even with my best modeling efforts, sometimes this approach to learning from practice did not converge to small errors even with model learning as part of learning from practice. With parametric models, this was often due to model structure errors. With nonparametric models [Atkeson et al. 1997ab], this was caused by training data from previous attempts not affecting the regions of model input space used in planning the next attempt. In these cases, adding model-free learning in the form of updating the actuator command trajectory directly with some function of the errors that is not a model (such as the output of the feedback controller on the last trial [Nakanishi and Schaal 2004]) helped move the distribution of training data collected from practice closer to the desired trajectory, and prevented learning from practice from getting "stuck" with large errors [Atkeson and Schaal 1997b]. I should note that feedback error learning is a very effective model-free approach to error correction learning all on its own [Nakanishi and Schaal 2004]. It relies on a good feedback controller, which according to the internal model principle, embeds a good model. **Lesson 19: Combining model-based and model-free learning can take advantage of the benefits of both.** Recent papers discussing such combinations include [Chebotar et al. 2017, Gu et al. 2016, Krueger and Griffiths 2017, Mordatch et al. 2016, Nagabandi et al. 2018, Pong et al. 2018].

In retrospect, I would do a different approach today, combining error correction and optimization-based learning. As in the first approach which mapped errors into one dimensional arrays of command corrections, I would maintain a policy offset in the form of one dimensional arrays. On each trial, I would update this policy offset as if I was doing error correction learning on a known desired trajectory, which was the desired trajectory generated by the most recent optimization. I would also update the model and compute a new optimization-based policy as previously described, and add in the policy offset. If I had enough trajectories, I would also learn a one dimensional array of linear mappings from desired trajectories to policy offsets at each time or phase of the behavior. A full correction would require N trajectories where N = state_dimensionality + 1. One could reduce the dimensionality of the linear mapping from errors to command offsets using some form of principal component analysis.

This leads to a revision of an earlier lesson: Instead of "learn models, compute policies", **Lesson 20: Simultaneously learn a model and a policy.** The learned model can be used to update the policy, and the learned policy can compensate for persistent modeling errors such as unmodeled dynamics. We are still assuming the cost/reward function is known perfectly and are computing the value function. It may be the case that learning everything and trying to maintain consistency between the representations using computation is a useful approach. **Lesson 21: Represent a**

**learned model, learned cost function, learned value (V and Q) functions, and learned policy simultaneously. During learning gradually reduce the inconsistencies between these representations**, that for the most part will improve them. When fine-tuning behavior put more weight on the model-free components of the representations. Maintaining multiple learned representations which are inconsistent provides a way to incorporate prior knowledge and enforce biases and constraints. One can think of this approach as similar to optimal control approaches that simultaneously represent inconsistent control and state trajectories, enforce the dynamics as a constraint, and allow constraint violations. Taking this approach to the extreme, an agent could also separately represent the quality of local models, the cost function, local regions of the value function, and local regions of the policy, and train these "quality maps" separately as well, in addition to using local training data density and the fit of local training data to a simple function (linear or quadratic polynomial, for example) to directly estimate the quality of represented quantities. Other things that can be learned or tuned during this type of learning are meta-parameters and features to use.

## 3.7 Linear dynamics parameterization leads to adaptive control "guarantees"

The observation that rigid-body dynamics models could be formulated to be linear in the unknown parameters was picked up in the field of adaptive control, where there were approaches that could be proved to converge to the right answer under those conditions [Niemeyer and Slotine 1988, Slotine and Weiping 1988]. This application of real-time model learning worked well. However, my experience with model structure errors made me cynical about the convergence proofs. **Lesson 22: Be skeptical about proofs that assume a perfect model exists or perfect knowledge of the model structure.**

## 3.8 Nonparametric models: memory-based and locally weighted learning

So by the end of my thesis, I had a robot that performed well on tracking any smooth desired trajectory, and could reduce errors on a particular trajectory down to the repeatability level of the robot by practicing that trajectory. I was left with the question, "How can a robot learn from similar trajectories to perform a trajectory as well as possible on the first try?" The approach I chose is to apply more modeling and system identification, from data collected while the robot is trying to do a task. But I needed to find a way to increase the representational capacity of the robot model, without losing the fast learning due to using idealized engineering models with small numbers of parameters. I chose the paradigm of learning a second nonparametric model to model the errors of the idealized model [Atkeson et al. 1997ab]. **Lesson 23: One can have multiple models for the same thing, with each model correcting the others.** Using a second model to correct a first model is known as "modeling the residual error (residuals)".

Global parametric models learn from small amounts of data (if they have few parameters) and can be evaluated quickly to make a prediction, provide clean estimates of derivatives, but suffer from model structure errors. Nonparametric models that make weak assumptions about model structure can represent almost any function, can asymptotically be perfect if they have the

correct input vector, and are not sensitive to training data distributions, but they need more training data, may be slow to evaluate, and typically generate noisy estimates of derivatives. Combining models provides the best of both worlds. Global parametric models with few parameters learn fast and generalize far providing clean outputs and derivative estimates, but will never have zero error. Nonparametric models learn more slowly, but can eventually get errors down to as small as possible (the level of the repeatability of the robot). This combination learns fast, provides a quick approximate answer with clean derivative estimates, and then a more accurate answer a little later. One can have multiple models for the same thing, with each model complementing the others. In our work a global parametric model based on idealized physics is typically used, as well as a memory-based nonparametric model to increase modeling capacity. We could have also used a parametric model with more parameters, with intermediate properties, such as a neural net, in this modeling hierarchy. At any point, the data stored in the memory can be used to update the parameters of the parametric models, which then requires updates of the model corrections in the non-parametric model of the residuals. The different types of models have to periodically re-balance what each is representing. **Lesson 24: Different types of models complement each other.**

I initially considered neural networks (which were hot in the 1980s) to model the residuals. I found that forgetting and negative interference due to learning new information and changing training data distributions made attaining consistently good performance difficult. Also, large neural networks were slow to train and even evaluate at that time. Computers are a million times faster now. One way to avoid undesired forgetting is to retain all of the training data and cycle through it during training. I then realized that if I was retaining all the training data, why not just operate on the training data directly to answer queries? This was the inspiration for me to look at memory-based learning, also known as locally weighted learning, and lazy learning [Atkeson et al. 1997ab]. **Lesson 25: If you are retaining the training data anyway, why not also use memory-based approaches?** At the least a memory-based model can help you estimate local model uncertainty based on the local density of training data and the quality of the local model fit.

In memory-based learning training data is simply stored in memory [Atkeson et al. 1997ab]. When a query needs to be answered, the memory is searched to find training data similar to the query. A "local" model is generated from that data, and is used to answer the query. Then the model is discarded. Usually the training data more similar to the query is weighted more during the model training, leading to the term "locally weighted". This type of learning is called "lazy", because as little work is done as possible until a query needs to be answered. The training data is just stored in memory. Training data that is not similar to any query is never processed. It is like a student who waits until an exam, asks what is on the exam, and only studies that material. This is a nonparametric representation and learner, in that there is not a fixed set of parameters to learn, such as weights in a neural network. Instead, the representational capacity of the learner scales with the training data, and resolution is automatically allocated according to the training data density.

Our experiments using this approach to control robots and have them learn interesting and fun dynamic tasks such as juggling were successful [Atkeson et al. 1997b, Atkeson and Schaal 1997ab, Schaal and Atkeson 1994, Schaal et al. 2002]. However, memory-based learning was ahead of its time then, and is probably still ahead of its time. One question that is often asked is "Won't you

run out of memory?" At that time it was a real problem. Now it is not. Terabytes are cheap, I have a 10 terabyte disk on my desk. Training sets of millions of data points are routinely used in neural network learning. If a training set can be stored for parametric learning, it can be stored for nonparametric learning. There is also no question that we can now form complex local models in real time.

The remaining question is "How can we find relevant data fast enough to answer a query?". Clearly the brain has no problem doing this, accessing our own vast memories. We need a way to access all of memory simultaneously in parallel, with the amount of bandwidth scaling with the amount of memory. With current computer architectures, what makes the most sense is to associate a part of the data set to each available core and have them search through their part of the data serially. Fancy data structures like kd-trees don't help much in high dimensions, and we want to access the memory with many different types of queries. The cores operate in parallel. Ideally, the memory should be on-chip with a high bandwidth path between the CPU and local memory, to reduce access time.

Processors tighly bound to memory is exactly what is happening with the increasing size of on-chip memory caches. I envision future machines on my desk and controlling robots with millions of cores and vast total memory in the form of on-chip cache. One view is that our processor chips will have memory on them and another is that our memory chips will have processors on them. We won't be able to say what is a processor chip and what is a memory chip. Top of the line consumer Intel CPU chips have 8 cores and a 16 Mbyte cache, so 2 Mbyte/core. A current wafer-scale chip, the Cerebras Wafer Scale Engine, has 400,000 cores and 18 Gigabytes, so 45 Kbyte/core. When using multiple chips, the processing power scales with the amount of memory, so total search time is almost constant, with a small logarithmic increase to find the overall best answers across all the cores. I built prototypes of this architecture in the late 1980s and early 1990s, but gave up because the rapid improvement in standard computers kept outpacing my special purpose computer development efforts. **Lesson 26: Until we are really done with Moore's law, don't build special purpose computing hardware.**

I note that local models can be of arbitrary complexity. I typically use linear or quadratic polynomials. If the neural network model structure has some special advantage, one can use neural networks as local models. Another note is that this is like various forms of experience replay [Lin 1992] and Gaussian Processes, which also store and cycle through the training data. My collaborator Stefan Schaal led the development of parametric forms of locally weighted learning, which could be implemented in real time [Schaal et al. 2000, Schaal and Atkeson 1998, Schaal et al. 2000 2002, Schaal and Atkeson 2010]. **Lesson 27: It is okay to be ahead of your time.**

Previously in this paper I suggested representing how well you know what you know. Memory-based representations are well suited for computing local estimates of the quality of models, value functions, policies, Q functions, and learned cost functions, based on the density of nearby points and the fit quality of the local model [Atkeson et al. 1997a]. Another approach to representing how well you know what you know is to maintain separate representations that are trained with various types of prediction error: predicting the next state, predicting the cost to attain a goal, assessing how fast the value function and policy are changing on each re-planning step, assessing the update magnitudes in updates to the value function, and assessing the inconsistency between computing

an optimal action using a stored model and value function and the stored policy. **Lesson 28: Represent how well you know what you know.** This knowledge may be learned and represented independently of learning what you know.

## 3.9 Building globally optimal plans out of many simple local greedy plans

At this point (late 1980s/early 1990s) I felt we had solved the problem of representing dynamic models. The next problem was to rapidly compute new policies as new data came in to the memory. There was an obvious answer to this question from optimal control. If local models were linear, the theory of linear quadratic regulators (LQR [Wikipedia 2020f]) could be used to compute policies. If local models were quadratic polynomials, an extension of LQR to quadratic models, known as Differential Dynamic Programming (DDP [Jacobson and Mayne 1970a, Wikipedia 2020a]), was appropriate. This led to work on combining locally optimal plans or policies based on trajectory optimization into globally optimal policies [Atkeson 1994, Atkeson and Morimoto 2003, Morimoto et al. 2003, Atkeson and Stephens 2007 2008, Liu and Atkeson 2009, Liu et al. 2013, Schaal and Atkeson 2010, Atkeson and Liu 2013]. **Lesson 29: We can build globally optimal plans by getting many greedy local planners to agree.** I think the idea that we can solve hard problems by combining the answers of many simple problem solvers will have a broad impact. **Lesson 30: Take advantage of related fields** such as optimal control, which means you have to know what is going on in them.

## 3.10 Multiple-model-based policy optimization handles unmodeled dynamics

A warning sign about my gung-ho pursuit of model-based approaches was that often the local policy generated by optimization such as LQR or DDP had too high feedback gains for things like humanoid standing. I kept getting hammered by model structure errors and unmodeled dynamics. This was my confrontation with the Sim2Real gap [Sim2Real 2020]. I needed a way to automatically generate local policies, but somehow limit the maximum feedback gains (the derivatives $\partial \text{command}_i / \partial \text{state}_j$ of the policy) based on information that was currently not available to the policy design methods: how wrong could the models be? Providing manually designed low-level controllers is standard for commercial robots partly for this reason.

I didn't solve this problem until the early 2010s, when I realized I could represent possible model variations by providing a set of possible models rather than a single model [Atkeson 2012]. If a policy worked on all of the possible models, it was likely but not guaranteed that it would work on the real robot. In the special case where the variation in the model was caused by variability in parameters that appear linearly in the model, one can provide a guarantee that the policy that worked on extreme model variations would work on any interpolated model [Boyd et al. 1994]. However, this is another example of assuming that the model structure is known perfectly, and that guarantee is not worth the paper it is printed on. More recent work on multiple models and robust control includes [Buckman et al. 2018, Clavera et al. 2018b, Kurutach et al. 2018, Recht 2019]

The multiple model approach forces the use of policy optimization rather than dynamic programming. One can find a separate policy for each possible model using dynamic programming, but this multiple model approach is based on finding one policy for all models. I have not found a good way to get dynamic programming to correctly handle multiple models. **Lesson 31: Optimizing over multiple models solves the model overfitting problem. Lesson 32: I will be focusing on policy optimization rather than dynamic programming-based algorithms** such as value iteration in the future because policy optimization can handle multiple model approaches to making the policy robust, as well as optimizing the selection of behavioral primitives. **Lesson 33: Selecting a policy from a set of alternatives can support fast learning** [Bentivegna et al. 2004].

## 3.11   Midlife crisis

At the turn of the century I had my midlife crisis. My father had always told me that one should change what one is doing at least every 10 years, so in the late 1990s it was time for a change. Switching from arms to legs was part of that change. I also broadened my research to try other areas out. I started a new effort focusing on using what I had learned in robotics to help people, particularly older adults. As part of a group effort on assistive devices and environments, I could contribute knowledge about sensing and embedded computing, and I could actually build stuff, which was useful [Group 2000]. At Georgia Tech, we built an actual house, that was supposed to be smart [Tech 2020]. I viewed it as just another robot with a lot of sensing. At CMU, I worked on capturing behavior in an Alzheimer's facility, with the hope of understanding drug interactions and doing a better job controlling medication levels in people who can't tell you how they feel, or about side effects [Informedia 2012]. This emphasis on technology to help people led to my efforts in human-scale soft robotics, in that I wanted to develop a human-sized robot that was intrinsically safe. This led to inflatable robots and the work inspiring the character Baymax in the Disney movie Big Hero 6, which was a lot of fun [Atkeson 2015]. More recently I have been working on assistive exoskeletons (using a model-free approach to reinforcement learning!) [Zhang et al. 2017]. Originally I was developing assistive technology for my mother (which she always rejected) and now I am developing it for myself. I figure I have about a 20 year deadline.

## 3.12   Hierarchical optimal control

As part of my midlife crisis, I moved from working on robot arms to legs. In this work on humanoid legged locomotion, we and others developed hierarchical optimization approaches to use real time optimization of complex models to choose commands [Feng et al. 2015 2016, Zucker et al. 2011]. We developed a range of models, from the most complete and detailed models we could think of (the low-level full model) to more simplified and less accurate models (the high-level simple models). We separated optimizations so that the optimization of simple models can look ahead in time to optimize the path and footstep locations. The output of that high-level optimization provides a value function to the more complex optimization of what torques to apply at this instant using the full model, so low-level full-body optimization can make choices and tradeoffs that take the future into account. Lower-levels of control (the more complex and accurate models) are actually re-optimized at a faster rate than higher-levels of control (the simpler and less accurate models),

such as choosing a route or footstep planning. This is multi-time-horizon or multi-level receding horizon control, also known as model predictive control (MPC). This hierarchical approach is based on abstractions (simple models). As described below in the section "What can be learned?", these abstractions can be learned from demonstrations or from slow or imperfect planners.

Optimizing this hierarchy of humanoid models in real time was a substantial achievement. Among the teams using the Atlas robot from Boston Dynamics in the DARPA Robotics Challenge there was a consensus that this hierarchical optimization approach was the way to go, and we each developed variants of it. As far as I know, this is also how the impressive performance of Boston Dynamics' Atlas II is achieved as well. **Lesson 34: An optimization hierarchy can both look far into the future using simplified models, and optimize complex models with a shorter planning horizon and faster reflexes.**

One problem we have not solved is what happens when the optimization of the full model fails, and simple model optimization must generate a different high-level plan, based on information it cannot represent using the simple model. Right now we increase the cost of plans that are similar to the current high-level plan, and re-optimize at the high level. We refer to this as poisoning specific plans. This is too crude, wiping out many plans that would work. If the robot just barely hit a limit, we don't want to eliminate plans that would just avoid the limit. Ideally, the low-level full-body model would generate information about local capabilities, in the form of optimization constraints, in terms that the high-level simple model planner can make use of. Unfortunately, you can't constrain maximum joint torques in a planner that doesn't represent joint torques at all. Perhaps this is an area where a model-free form of planning that gets punishment for making bad high-level plans is useful. Another way to deal with this problem is to do full-model optimization (optimize only using the most detailed model) for several seconds in to the future (the needed time horizon for humanoid walking). It will be interesting to see in 20 years or when full-model optimization over the necessary time horizon is possible whether the hierarchical optimization approach is still useful.

## 3.13   Model-based safety supervision

Our Atlas robot was the only robot not to fall down or need human rescue in any tests in the DARPA Robotics Challenge. Our secret was to use model-based safety supervision at all times while the robot is operating. If the models predicted the robot was about to fall or engage in risky behavior, the robot was put in a safe state and remote human assistance was requested. I do not think it is possible to do that level of safety checking without explicit models. A safety policy could be learned using a model offline, but that is still model-based, and the online model-based safety checking is likely to be more accurate, as it doesn't suffer from the training and representational issues of learning a policy offline. **Lesson 35: Model-based safety supervision is critical to reliable robot behavior.** One of the arguments in favor of model-based learning is **Lesson 36: Models have many uses.** Predicting when things are going to go bad is an important one. Communicating with humans and other robots is another one.

## 3.14 The quest for robust performance: Learning does not play a role

A comment on the learning work I have done, as well as other's work:

> *"They are dealing with simplistic or 'playful' tasks, such as playing table tennis, badminton and pendulum swinging. Even when more challenging tasks are addressed, e.g. maneuvering of autonomous vehicles, what is actually reported is successful tests of very limited practical applicability. Relevant literature evidently lacks usage examples of model-based RL in more 'serious' fields requiring reliability and robustness, such as in service or industrial robotics."* [Polydoros and Nalpantidis 2017]

As part of my midlife crisis, I changed my emphasis in robotics, away from 'playful' robotics to 'serious' robotics. There led to a more subtle change in my thinking than just whether I worked on arms and legs. I wanted to focus on getting state-of-the-art performance from the best humanoid robots we could find. How close to human-level competence could we get?

Although the work on robot learning had been helpful to get state-of-the-art performance out of various robot arms, the work on legged locomotion involved almost no work on robot learning. Even the DARPA Learning Locomotion program turned out to be mostly about planning and control, and none of the teams did a lot of robot learning research. I am still trying to figure out why the field of learning seemed to have little to contribute to help us with high-performance robust humanoid walking, so that we can work on areas of robot learning that would be helpful. As a counterpoint, Kober et al. [2013] extensively survey successful robot learning in many areas.

In our work on legged locomotion, we had working simulations and crude robot walking early on (Appendix 2), but it was a huge challenge to get robots to walk robustly. In some sense this is a reflection of a Sim2Real gap [Sim2Real 2020], but it was also a reflection that **Lesson 37: having a policy based on a simple model that works on the robot for a demo is not enough.** This is one of the lessons we learned in both the DARPA Learning Locomotion program, the DARPA Robotics Challenge, and all our other work in legged locomotion.

Reinforcement learning only gave us fragile policies, and we didn't know how to use weak AI methods (domain knowledge-free and model-free) to produce robust policies. In legged locomotion we had much less of a clear idea of what we should do compared to working on robot arms. It did not make sense to impose arbitrary desired trajectories, or servo feet to arbitrary positions and orientations. Instead, we wanted robust behavior that both balanced and walked. Sorting out the right way to control legged locomotion in a dynamic torque-controlled humanoid, and then implementing it, took more than a decade. We used domain knowledge in a fairly traditional engineering approach. Focusing on understanding the problem (computational theories in the sense of Marr [1982]) was critical. An example of a computational theory is this work on passive dynamic walking which emphasizes the role of mechanics in generating behavior [Anderson et al. 2005], and raised the question of what paradigm or strategy was going to guide our work.

**Another aside:** It is interesting to note that Marc Raibert, one of the founders of Boston Dynamics, which is the world leader in legged locomotion and dynamic torque-controlled robots, did his PhD thesis on robot learning, but swore off robot learning after that (my thesis was about a decade after his in the same department, and had similar goals of learning and using robot dynamics). Only now is Boston Dynamics beginning to do work on robot learning, and it is mostly

focusing on perceptual learning.

Part of this course change involved changing what conferences I attended. I stopped going to NeurIPS (or whatever it was called then) and helped found a new conference on Dynamic Walking, as well as attend the IEEE Humanoids conference. I wasn't interested in churning out yet another variant of some learning algorithm that wasn't going to matter on a real robot. If I was going to explore robot learning, it would need to improve actual robot performance. I guess I wasn't interested in studying learning just for the sake of understanding how learning might work in the abstract. I was a faculty member in the Robotics Institute, not the Machine Learning Department, which reflected my interests.

So why didn't robot learning in general or reinforcement learning have more to offer to legged locomotion? Answering this question might help us do better work on robot learning.

1. **Fixation on model-free approaches:** Reinforcement learning seemed fixated on model-free approaches. My group wasn't willing to start again from scratch, with little ability to use our hard-won knowledge about the robot and the task.

2. **Computational limits:** We certainly couldn't run model-free reinforcement learning directly on a robot, the sample efficiency is orders of magnitude not good enough. So we would have to simulate our robot and learn in simulation. In the 20th century optimization of any kind was still slow (computationally expensive), and homework assignments in our classes today were challenging problems then. I am told that each run of training AlphaZero, a "model-free" reinforcement learning program that learns to play games using perfect models of the games) costs on the order of a million dollars in computer time. Since computers were about a thousand times slower at the turn of the century, that's a billion dollars in 2000. We couldn't and can't afford that. **Aside:** The claim that model-free reinforcement learning democratizes anything or makes machine learning accessible to all is not quite true, if you have to own a data center near a dam to actually run the algorithms on non-trivial problems.

3. **We are working at the limits of the hardware, so subtle details matter:** Given joint velocity, torque, and power limits described below, the maximum walking speed of our humanoids is about $1/3$ human walking speed. In order to go faster, and respond to perturbations more effectively, the robot needs to be able to swing its leg and place its foot faster. We operate close to the limits of the robot actuators in order to get robust behavior, where subtle effects like actuators interacting through the hydraulic pressure lines and changes in how smoothly oil is flowing in and out of pistons start to affect behavior (Appendix 2).

4. **Trying to make a robot look like a simple textbook model is the wrong way to go,** unless you have lots of performance you can throw away and still be happy. To make a machine appear to obey simple textbook models, it needs to be over-engineered. Precision (as with industrial robots) is typically obtained by having excess structure to make the machine stiff. That structure also makes a robot heavy. This is tolerable (barely) for a robot bolted to the floor. Legged robots (at least the ones whose performance approach human performance) have to be light, so structure has to be minimized (as on an airplane). Airplane wings bend

(watch them next time you fly), airplane bodies deform, and humanoid body parts bend and deform as well.

Electric direct drive actuation, which approaches an ideal torque source, has to use motors that are too large and heavy to get force and torque levels comparable to a human. High gear ratios, which make robots non-backdrivable and stiff, allow small light motors to be used. However, gearing limits how fast a joint can move, since electrical issues (back EMF) limit how fast we can spin the motor. Typical electric humanoid joints can attain less than half a revolution per second. Human leg joints can achieve more than two revolutions per second. Electric motors have thermal limits on how much mechanical power they can generate.

Force control can be used to compensate for gears, non-backdrivability, and stiffness. (**Aside:** You can't measure force directly, and it has no physical reality. You can only measure the effects of something you intellectualize as force. It is a story that was made up to explain movement of matter. In robots, what we actually measure is small compression and stretching of material, known as strain. In the early days what was measured was motion of the planets, to measure gravity.) The problem with force feedback is that the force sensor picks up vibration and every shock wave caused by contact such as putting a foot down. With enough amplification and filtering, you can use force sensors as microphones and recover speech frequencies. This forces us to low pass filter the sensors. Shock waves travel at the speed of sound that is material-dependent and along pathways defined by the shape of the part. Forces on a structure are therefore not uniform, and what a force sensor measures is not the force at all points in the structure. The low pass filtering, jello-like nature of structure, and delays in applying feedback limit how good our force and torque control can be before the force feedback gains cause instability (Appendix 2). To make the robot look like simple rigid bodies driven by torque sources at the joints, as in Open AI Gym and Mujoco, we have to operate our force control with as high feedback gains as possible, just below stability limits. It doesn't take much variation in the dynamics of the robot (say thermal effects) to push the robot over these limits.

Hydraulics (which we use to get large enough forces and fast enough motion (force*velocity = power)) are not backdrivable either, and high performance force feedback has to be used to make hydraulics approximate a source of pure force or torque. So I am left with the conclusion: **Lesson 38: Don't try to make robot hardware match an idealized simple model.** Trying to make the robot behave like something it is not has a performance cost. **Lesson 39: Take advantage of the robot's natural behavior and dynamics.** Focus on developing control that takes advantage of all a robot's complexities. Appendix 2 provides an example of what this means.

5. **We need robust policies:** What limited performance both in the DARPA Learning Locomotion Program and the the DARPA Robotics Challenge was not that we couldn't make policies. It was that the policies and plans were not robust. If anything slightly changed, the policy failed. A major question for learning approaches is not how to generate policies, but how to generate robust policies without losing too much performance. The work on using multiple models to make policies robust was my attempt to deal with this issue [Atkeson

2012].

**Lesson 40: Robustness involves loss of performance:** Not allowing policies to take advantage of every detail of an individual robot (overfit) lowers performance. Since we do not have a lot of performance we are willing to give up, this tradeoff is challenging. We don't have good design tools to make intelligent choices. We have no way to tell an optimizer or reinforcement learning algorithm what kind of robustness is needed where in the state space. As far as I can tell, this issue has not been adequately addressed in the field of robot learning, and it is not clear that reinforcement learning can deliver the policy robustness needed. I wanted to work on robots, and not have to focus on tinkering with learning algorithms in a quest to make them robust enough. Adding noise to a simulator is a crude tool to do this. Learning over multiple models is much better, but we still need even better tools. The fragility of the policies was a reflection of the complexity of the real world. All sorts of things were going on. Let me repeat my list of excuses as to why we couldn't accurately locate the robot center of mass: the measurements of joint torque were not perfect; the joint axes were not exactly the same as what the model assumed; there were significant actuator dynamics; the robots were not made up of rigid bodies, the deformations depended on the load, and the deformations changed the relationships of the joint axes as well as the rigid-body parameters; the robots were hydraulic and the hoses were definitely deformable; and there were thermal effects on the control valves, oil flow, and friction, and the robots heated up as they operated, so their temperature (which varied over the body) depended on elapsed run time and the weather.

6. **Robots are deterministic, and reinforcement learning emphasizes stochastic models:** This is where mechanical engineers and computer scientists part ways. Physical systems other than lava lamps are rarely stochastic. The emphasis in reinforcement learning on stochastic models has little relevance for high performance dynamic robotics. Every robot I have ever worked with has been highly repeatable on the time scale of an hour or so, and some for much longer. If you put any particular robot in the same initial state, and executed the same sequence of commands, the same thing happened, down to almost the smallest and highest frequency wiggle we could measure (Appendix 3). It often takes little adjustment to get a policy working again after some time apart from a robot. **Lesson 41: Individual robots are highly repeatable for at least a limited time,** so deterministic learning should help [Silver et al. 2014, Lillicrap et al. 2016]. The stochastic transition models used in reinforcement learning are not good models of the robots I have worked with, or their model errors. Small deterministic modeling errors can destabilize a control policy. Errors modeled as stochasticity do not have this effect. At most, a small additive process noise term, as found in Kalman filters, is the most stochasticity needed to model robots, and even that is often too much. **Lesson 42: Stochastic models are not the best way to model or think about robot dynamics, model errors, or unmodeled dynamics.** Polydoros et al. [2015] disagree, saying that lower cost robots that are missing key sensors, such as joint torque sensors, have to estimate quantities like joint torques, and those estimates are noisy. I bet that is not really true.

**Aside:** A related point which is highly controversial is: Don't use stochasticity to model ignorance, or what you don't know about a system, or to make up for not having direct measurement of the full state of the world. If you do an experiment several times, you typically don't get a particularly random distribution of effects, errors, or failures, and you usually can explain each one of them in mechanistic terms, which means you the experimenter have a model of what is going on. My car slid off the road because the road had oil on it from lots of cars driving on it, and it had just begun to rain [Smith 2017]. Heavy rain causes hydroplaning. Bridges ice over before the road does. These are models. I don't think I have ever seen a robot fail that I had to attribute to white noise in the dynamics. A more realistic approach is to model what you don't know in a deterministic way, such as multiple possible models or environments, with unknown parameters that are not learnable (they change too fast relative to the measurements you can learn from, or are dependent on the full environment state).

7. **Stochastic policies are bad for you and your robot:** The emphasis in reinforcement learning on stochastic policies makes current reinforcement learning approaches unusable in high performance dynamic robotics. Stochastic policies make control harder, damage robots, and drive research in unproductive directions.

   Commands to robots need to be smooth to avoid exciting high frequency unmodeled dynamics. Jerkiness (the derivative of acceleration) adds wear and tear on joints, actuators, and transmissions). Variations in loading cause material fatigue and failure [Wikipedia 2020b]. If I want to drop something, I shake my hand and fingers. When I want to break a ductile or flexible material like a branch off a tree and bending it doesn't work because it is not brittle enough, I bend it back and forth to generate changes in forces. It breaks, and I haven't applied a high force. An example of a robot part failing is shown in Appendix 2 (robot snuff porn). The part probably failed because we drove it hard with various frequency sinusoids while modeling the joint.

   It is true that by changing the actions to ever higher derivatives of the actual actions, one can make the actual actions smooth, but this also makes learning slower and more complicated. If you need to explore dynamics, you need to pump energy through the system at the same level at the original input points no matter how many integrators you use to smooth the command. It is a well-known principle in system identification and machine learning to match the training set data distribution with the test set and actual use data distribution. Using input signals with high frequency energy (added random noise) when you are not going to add the same noise when you run your robot violates this principle. **Lesson 43: Stochastic models are not acceptable to control highly dynamic robots.**

   **Lesson 44: Model-based approaches use derivatives instead of local exploration:** I understand that many reinforcement learning algorithms need stochasticity to sneak in the exploration they need, but it is the wrong way to explore. Model-based approaches use known or learned model derivatives to figure out what happens if commands are changed, rather than numerically estimating derivatives using stochastic methods. Model-based approaches don't need to explore where the model is already known well, even if the policy is not. "Exploration" to improve the policy can happen in "mental practice" or simulation. Silver et al.

[2014] discusses this issue from an algorithmic perspective and concludes that reinforcement learning is more efficient with deterministic policies. **Caveat:** If the random variations in commands are separated into separate trials or episodes, as in function optimizers such as CMA-ES or CEM that randomly choose parameters that are fixed during a trial, and the trials are separated in time, this added stochasticity is not exciting system dynamics.

8. **Known and unknown limits:** It is hard to get optimized policies to obey known and unknown limits. Policies have to obey performance limits such as maximum joint velocities and joint torques, which is achievable (but not often enforced) in current work. There are more subtle performance limits that are hard to specify, such as bandwidth limits on actuator commands, and minimizing the high frequency content (jerkiness) of command sequences, to avoid exciting unmodeled dynamics. Limits on behavior in order to maintain contact are not well understood. What are the friction limits for a foot that is on edge, such as during heel strike and toe off, or during a bad step? Finite element analysis might be necessary to understand when a rubber foot pad with varying temperature, covered with leaked hydraulic oil, and rapidly deforming due to shock waves and vibration will slip. What is the effect of a torque about the normal axis of the contact, or actual twisting motion, on horizontal slipping? What is the effect of varying types of ground texture, material, and material properties, which are not known to the robot?

9. **Optimized feedback gains based on textbook models don't work:** Optimization policy design using Open AI Gym/Mujoco-style rigid-body models produces feedback gains that are too high, and cause the robot to oscillate violently ("the robot goes unstable"). Appendix 2 describes this issue in more detail.

10. **We don't understand many of these effects:** Much of our effort on humanoid walking was directed at how to improve performance of the controller due to non-rigid-body non-perfect-torque-source non-simple-contact nature of the humanoid, including: actuator dynamics, contact models, and joint coupling through the power system, Little details mattered, like compliance of the footpad and in other parts of the structure. It was not clear we would be able to understand and model all the important effects to put in a simulator in our lifetimes. We could have spent years trying to understand all these effects, but we wanted to get our robots to walk!

11. **Accurate simulators will be too expensive:** It is not clear that simulators based on rigid-body models can capture many of the physical effects that caused us problems. Finite element models of deformable materials may be necessary to address them. Accurate simulators may be too expensive to create and too computationally expensive or slow to be worth using in learning. Current simulators can't even simulate a rigid object resting on another rigid object much faster than real time. Wait a little while, and the object starts moving, and any tower falls over. This has been true for decades. Do we really think this is going to get solved any time soon for faster-than-real-time simulators? (If you think you have solved it without a special purpose hack like turning off simulation of objects at rest, send me email cga@cmu.edu). It is much cheaper to work with the robot directly rather than simulate first,

then transfer to a robot, because handling the Sim2Real gap is expensive and time consuming [Sim2Real 2020]. Think about deformable bodies, liquids, and granular materials. Think about thermal effects. Even friction is badly modeled in today's simulators. Throw a few grains of sand on the floor, and suddenly friction is completely different. Model that! The world is its own best model, and has no Sim2Real gap. Solving the Sim2Real gap is AI complete: If we can handle Sim2Real gap with crappy simulators, we can do it with no simulation, and we can do anything.

12. **Managing and minimizing the effects of unmodeled dynamics and sensor flaws by shaping behavior is important:** In the DARPA Robotics Challenge we manually designed behavior that minimized the effect of unmodeled dynamics. We had to work hard to avoid shock waves due to impact causing feet to start slipping, for example. We chose to walk gently. Interestingly, other teams in the DARPA Robotics Challenge shaped behavior to make the sensors more reliable. An example is stomping feet, so it is easy to tell when contact begins. We had to understand the machine, and use our human ability to manually optimize low-level control on the actual robot, and reason about effects that are not well quantified or modeled. This was also true of our robot arms, but a full humanoid is much more complicated. Humans are still much better at manually performing reinforcement learning than our algorithms. Being a "robot whisperer" is still useful.

13. **Learning safely is a big issue:** I am not willing to turn our million dollar robots over to people who don't know anything about mechanical engineering or running robots. It is too easy to destroy them, or run up huge repair bills (An example of a broken robot is shown in Appendix 2). We train our robotics students to safely work with robots. So far, we have no reliable way to train learning algorithms to safely work with robots. Given we have to operate at the limits of the machine, current safe learning approaches are too conservative. We have to be able to operate at the safety boundary to get the most out of our robots.

All of this sounds like an argument for model-free reinforcement learning, and that we were doing a lot of human-guided model-free reinforcement learning. Yes, this is true. Automated learning has to run safely on the real robot and be sample efficient. I roughly calculate that we have run our humanoid robot less than one hundred thousand times ever, and certainly less than a million times. Humans are still much better at learning than our algorithms. Humans can ask specific questions and design experiments to answer them, as well as make simple mental models of what they have seen so far, and use the models to guide what they try next. Humans are also good at knowing when they are too close to a safety limit, and backing off.

My overall concern about current work in reinforcement learning that is supposed to be relevant to the kind of robotics that I do (dynamic tasks like walking and juggling) is that I am not seeing results that I would actually be able to apply on high performance robots, or that address what I see as the fundamental problem of complexity (see below). Other than [Hwangbo et al. 2019], I haven't seen any results in model-free or deep reinforcement learning that I would be willing to run on our million-dollar humanoid. It is true that various kinds of automated reinforcement learning have been run on robots, sometimes with almost 100,000 physical trials. I have not yet seen learned behaviors that I thought couldn't have been hand programmed given the same levels

of effort and computational resources. I look forward to getting email pointing to work that proves me wrong on this (cga@cmu.edu).

Some more philosophical points:

1. One of my main objections to reinforcement learning implementations on robots is that they often start with little knowledge, and flail around making movements that even the simplest models would eliminate. There is no reason to use real time on a real robot to flail. To a robot experimentalist, it is aesthetically displeasing. It is humiliating for the robot. When the robot revolution comes, researchers who abuse robots in this way will be the first to go. Start with more knowledge! A poor quality simulation would have worked just fine to let the robot know it has to align two Lego pieces in order to connect them. This issue comes up in psychology as well:

   *"Early animal learning researchers disagreed about the degree of guidance an animal uses in selecting its actions in situations like Thorndike's puzzle boxes. Are actions the result of "absolutely random, blind groping" (Woodworth, 1938, p. 777), or is there some degree of guidance, either from prior learning, reasoning, or other means?"* [Sutton and Barto 2018]

2. **Robotics as a field is not happy with black box learning methods:** Robotics rewards "understanding" the task, the robot, and what affects what in reality. Good engineering. Machine learning as a field rewards generating new algorithms and demonstrating success on something, where simulations are ok. These are different goals.

3. **Applying black box methods too early misses debugging and design opportunities:** Once a black box learner is let loose, it becomes really hard to realize that you have bugs in your software, since the robot is learning how to work around your bugs. It is also very difficult to trace down what the bug is. One stops thinking about better strategies to do tasks or achieve goals. Poor choices of states, features, inputs, outputs, goals, and whatever else you can think of don't get reviewed or improved.

4. **Relying on only simulation is bad for you:** Although simulation is great for debugging software and ideas, "Simulations are doomed to succeed". They give you a false sense of accomplishment and that you are making progress, and enables you to remain in denial about the complexity of the world. Simulation is an enabler of bad habits and attitudes. The real world is full of surprises and things one did not anticipate.

5. **If we let students use learning techniques, they will go soft:** We won't be able to get them to do the hard (and sometimes boring) work of modeling and calibrating what is going on. A good roboticist has to be trained to be anal retentive. Details matter. Think about that next time you are riding in a self-driving car. There will be a worldwide shortage of robot whisperers. Good robot control still depends on mechanical intuition and a human's ability to "be the robot".

6. **Most learning work in robotics involves human learning, not machine learning:** My experience is that graduate students take a long time fiddling with algorithms and setting things up, until finally something works. Nothing works out of the box, especially machine learning and computer vision.

7. **Work on simulations in OpenAI Gym and things like it haven't had impact on real robots:** I haven't seen any of that work implemented, or improvements in robotics credited to work in these simulators. Simulation-only work is not a very effective way to have impact. Simulations that have helped have been special purpose and accurately modeled the robot involved.

8. **Worst of both worlds:** In the worst case, simulation-based reinforcement learning might give us the worst of both worlds. Simulators might not be able to model the real world accurately, and might also present a harder problem for the learning algorithms, without any commensurate benefits.

9. **The bubble has not popped yet:** Simulation has enabled the model-free deep reinforcement learning bubble to continue. If the researchers had been forced to demonstrate performance on real hardware, that bubble would have popped long ago.

10. **Will you be around to help clean up when the party is over?** Reinforcement learning in simulation has helped a lot of people from other fields get into robotics. That's great. Will they stick with it to do the much bigger amount of work to get real robots working in the real world? Cars have been safely driving themselves in simulation for a long time now, but we still have a long way to go in reality.

11. **Is robot action fundamentally different from vision, speech, and language?** I am told by several friends who looked at this paper that much of the past work in computer vision, speech, and natural language has been abandoned. Deep learning has been such a transformative paradigm shift that decades of past work is left to collect dust. It is an interesting question as to whether there is something different about robot action, or whether what we have done will suffer the same fate. For robot action, physics, energy flows and storage, design constraints, and control engineering all matter. I find it hard to believe that deep mumble will wash it all away.

Wow. As I look at this, I can see myself as someone in the horse business just as cars started to take over. Or a guy shoveling coal when the internal combustion engine took off. Hmmmm ...

## 3.15  The complexity barrier

Robots are not ready for the real world. I believe **the complexity barrier is the problem. Anything can happen.** I bought three different brands of model-based robot vacuums to try out (including a Roomba). I found I had to change things in my house to even have a chance to make them work. All the electrical cords had to be made inaccessible, as well as anything else that could be sucked in (many small rugs). I found the robots sometimes could not enter a room because there

was a 1-2cm step up in the flooring. Before I run any of these robots, I have to patrol the area, pick up stuff up, remove dangerous objects and clutter, and move furniture around to make things more accessible. When I actually get around to running the vacuums, each one needs to be rescued by me basically every other run because something goes wrong. I am still learning new failure modes, and I am sure more will come. With vacuum cleaners we can tolerate a fairly high failure rate. With humanoids and self-driving cars we can't. **Rare cases kill.**

In a recent debate at ICRA 2019 on Sim2Real, Pieter Abbeel said "In real applications it is all about going from 90 or 99% to 99.9999% (reliability). Sim2Real is about getting initial performance (demo-level), not getting reliability. In real applications the customer typically has already achieved the initial performance, and wants better performance." Davide Scaramuzza pointed out "Edge cases are the issue." Aleksandra Faust emphasized model structure error: "The key is unknown unknowns." Only by working in the real world with real robots can you find out about unknown unknowns. I came away from these experiences with some lessons about the complexity barrier, in addition to hoping that the computer perception (vision, audition, and tactile sensing) folks can get their act together and address reality:

**Lesson 45: It is easier to work with real robots than to create simulators** of those robots and their environments that capture the true complexity that we humans take advantage of. I have watched robot demos and seen pieces of tape seemingly stuck at random on robot fingers. I ask, "What's up with the tape?" The answer is usually that just the right friction had to be created for the demo to work. In my own juggling work, the parasitic compliance and damping (energy loss) of the robot, especially in the fingers, hands, and wrists, had to be carefully modulated by trying out different materials, just like Edison did in his "invention" of the light bulb by exhaustive search. For Shannon juggling (a form of open loop ball juggling) we had to add damping materials to keep the balls from bouncing out of the hands. For devil stick juggling, we used adjustable model car shock absorbers to tune how the sticks were held to prolong the contact as long as possible. Thermal effects and the coat of leaked oil on all hydraulic robots makes a difference. What happens when we work with liquids and granular materials? What happens when material properties matter, or complex processes like "wetting" a material change its properties? I have seen robot grippers based on adhesion due to surface tension, as well as grippers that melt and solidify glue to grasp. Did you know our skin mechanical properties change according to how hydrated the skin is due to humidity? Simulating these effects well is more work and a bigger challenge than programming robots. Let's leave improving simulators to the simulator community, and move on with reality.

**Lesson 46: Reducing degrees of freedom in learning is useful**, but the learner needs to be able to choose and change what those degrees of freedom are, and needs full access to the lowest levels of control in order to do that. I have a theory about human skill learning that humans focus on learning how to control one or just a few directions at a time. Consider windsurfing. First, the unstable directions/modes have to be controlled, but the stable directions/modes can be ignored. Then, transient behavior such as raising the sail can be learned. Balance, moving forward, and then steering are learned sequentially. Only then are all the behaviors refined together in an end-to-end fashion. This is a version of shaping [Sutton and Barto 2018] where the learner chooses which behaviors to learn and when.

**Lesson 47: Massive amounts of multi-modal redundant sensing is the key to better robot performance.** What made self-driving cars possible? It wasn't control or planning algorithms. It was better sensing (GPS, maps (a form of sensing), better computer vision, LIDAR, radar, etc.). Perhaps the same will be true of other types of robotics. One area I am currently focusing on is superhuman sensing for robots. For example, why do we only have 2 eyes? Why not 100, all over our body? [Atkeson 2020b]

**Lesson 48: Current work in reinforcement learning works on problems that are too simple.** States and actions are known. At most, the physics is rigid-body dynamics. Not many people are trying things out on real robots.

**Lesson 49: The effect of Moore's law.** One of the most important lessons I learned is that don't think the world you work in now will be the world you see in the future. We are riding a roller-coaster of technical and societal change. In terms of technical change, Moore's law (and whatever succeeds Moore's law like wafer-scale integration) has a huge effect on what we should do in research. Computers are a million times faster and more powerful and memories a million times bigger than when I started out. For the last 50 years computers and memory became both exponentially cheaper and more powerful. Ideas that didn't work out in the past could be brought back to life later. It is important not to accept limits based on "what we can do now". Control theory and other aspects of engineering seem stuck with the worldview that computing and storage are expensive, and algorithms should have minimal computational cost and memory usage. This is similar to my father's attitude about long-distance phone calls. He never understood that long-distance calls had become cheap, if not free. When I phoned home, he would always cut the call short to save money. "Going big" and assuming plentiful (if not infinite) computational and storage resources changes how we think about problems.

In terms of societal change, don't assume some new technology won't be accepted (such as always-on personal cameras). I thought lawyers would prevent self-driving cars from ever happening. And then, almost overnight, it was happening, driven by Google. The Covid pandemic clearly is driving huge societal change, such as remote work and much higher levels of automation, which will not be reversed after the pandemic ends.

## 3.16   Contributions

You are supposed to list the contributions of your thesis. I think this is my fourth thesis-like document (original thesis, locally weighted learning survey papers, multiple model tech report). Here is my attempt to identify my contributions in this area:

1. Articulated and advocated (along with others) a number of intellectual positions (memes) including: "Learning is optimization"; "Reinforcement learning is optimal control"; "More control should be model-based"; "Focus on model-based reinforcement learning"; "Learn models, compute policies"; "Memory-based learning"; "Trajectory-based reinforcement learning"; "Plan optimistically, since models have errors"; "Combine many simple local plans to make a complex global plan"; "Represent models **and** policies"; "Use models more effectively in model-free learning"; "Reinforcement learning is not paying attention to model structure errors"; "Explicitly make models of model error: model$^2$"; "Model-based safety

supervision"; "Let robots be robots"; "Stochastic X is irrelevant", "Task-level learning"; "Learn in the context of rich knowledge"; "Library-based learning"; "Thinking is behavior, and robots should learn how to manage it"; "Complexity barrier"; "Symbolic learning will be back"; "Strategies matter"; "Reason about alternative strategies"; "Superhuman sensing";

2. Set standards: Work with real robots. Learn in less than 100 trials. Do challenging dynamic tasks like walking and juggling. Implemented model-based robot learning (including model-based reinforcement learning) on actual robots performing difficult dynamics tasks such as juggling, and demonstrated sample efficient learning requiring less than 100 trials.

3. Showed that learning robot models and using them for control is practical. Early work focused on low level learning of rigid-body models and feedforward commands to low-level control systems. Other work focused on task-level learning.

4. Showed the role of models in error correction learning to attain a goal or trajectory. Introduced the concept of "model-based learning" to the robotics community.

5. Showed how memory-based learning and other library-based learning approaches could be applied to robotics, and demonstrated their use on real robots. Developed meta-learning approaches to tune memory-based learning.

6. Developed and demonstrated on actual robots methods to learn from imitation, using one or a small number of examples, and then learn from practice using model-based reinforcement learning.

7. Introduced and/or demonstrated techniques from optimal control to reinforcement learning, such as Differential Dynamic Programming (DDP) to create local models of the value function and perform trajectory-based reinforcement learning. Demonstrated region-based reinforcement learning, using LQR and DDP.

8. Showed how model-based control and learning, as well as optimal control could handle modeling errors. One example is using iterative error correction learning to achieve a goal. Another is using multiple models during policy optimization.

9. Developed reinforcement learning based on sets of trajectories. Showed how to combine many simple locally optimal plans to build a complex overall plan that is globally optimal for a region, by getting the simple plans to agree at their borders.

10. Demonstrated a version of hybrid model-based and model-free reinforcement learning.

11. Developed an "optimization in the face of uncertainty" approach to trajectory-based reinforcement learning.

12. Developed memory-based selection-based learning from a set of behavioral primitives with parameters (arguments of subroutines).

13. In the early days I tended to focus on applying one approach to one task on one robot. Now I encourage combining multiple different approaches, to achieve the best of each. Different approaches can complement each other (using model-free learning to assist model-based approaches for example), and can appear in a hierarchy of learning at different levels. I am pushing to address multiple tasks with multiple robots, but haven't gotten there yet.

Folks doing deep reinforcement learning often get annoyed when old fogies rise up and say "We already knew X, you are just using more computation." While I do not fully subscribe to this view (AlphaZero and MuZero are qualitative steps forward), it is an interesting exercise to ask "What did we know, and when did we know it" (a question we ask on crime shows and when impeaching a president). Here is a list of concepts that were developed in the "old days", and continue to matter today (possible values for X are): models help control; modeling errors accumulate over time in trajectory optimization and simulation; model-based learning; models can be improved (system identification/model learning) during robot operation; improving models is a form of reinforcement learning if policies are computed from models; models are the ideal learning operator in error correction learning; use sets of trajectories to represent policies (and value functions), (the idea of fitting a global parametric policy to local policies in Guided Policy Search is new); deadbeat control and learning using inverse dynamics operators is not robust; optimization-based control and learning can be made robust; optimization in the face of uncertainty is useful; optimizing one policy for multiple models improves policy robustness; combine model-based and model-free learning: model-based learning rapidly gets agent into the ball park, and model-free learning is the closer; trust region optimization techniques; task-level modeling and learning; optimized behavior is typically simple; library-based learning; learning to select primitives, and choose parameters for the primitives such as sub-goals; and meta-learning/learning to learn.

# 4 Research questions and thesis topics

I am now focused on task-level learning, abstraction and symbolic reasoning and learning, learning in the context of rich pre-existing knowledge, enabling robots to improvise new strategies in more complex tasks, and superhuman sensing [Atkeson 2020cb]. What is my next thesis going to be on? I started out in neuroscience, where aging professors go from a career of studying a specific behavior driven by a specific part of the nervous system, to trying to figure out how consciousness works. I now understand that phenomena better. What do aging roboticists do?

1) **What levels should learning operate on?** Up to now this paper has focused on work that generated raw actuator commands based on detailed (low-level) measurements. This is what I call the atomic level of representation. As I watched humans rapidly learn tasks like juggling, I realized that if I wanted quick learning and to be less vulnerable to model structure errors, I needed to look at learning at higher levels. It seemed like humans were selecting and combining more complex behaviors than individual joint torques, and then doing "end-to-end" learning to refine the behavior [Wikipedia 2020o]. Task-level learning is focused on learning to do single tasks, in the context of a known parameterized policy. Task-level models map from task-level command parameters to task outcomes, without attempting to model any intermediate details. The agent

commands `throw(7.2, 9.5, 4.1)` and the ball lands at `(4.3, 5.8, 1.2)`. The goal of task-level learning is to figure out what arguments to the `throw()` policy hit the target. We did some exploratory work on task-level learning when I started out as a professor [Aboaf et al. 1989], but let it drop because it was not clear how to make it less heuristic. The work of Darrin Bentivegna that based learning on selecting from a set of manually designed policy primitives and generating arguments for them convinced me that this was an important avenue to pursue [Bentivegna et al. 2004]. Learning high-level policies, such as learning to select from previously learned behaviors, is a useful area for future research.

We have also found that learning at the atomic level is an inefficient way to learn a particular task, compared to task-level models specific to particular tasks. Many tasks, such as tasks like cooking, or tasks that involve liquids or granular materials are complex to model at the atomic level. Also, in the context of a task the degrees of freedom of the system are greatly reduced. We don't need to model fluid dynamics to learn about how to pour from a particular pitcher in a particular way [Yamaguchi and Atkeson 2015b 2016cb, Yamaguchi et al. 2014 2015]. Applying a planner and optimization criterion can reduce the dimensionality of the system further. We found that optimized walking behavior was much easier to model than arbitrary leg movements [Kim et al. 2013]. Parameterizing a behavior to do a specific task with a few control parameters supports rapid learning [Aboaf et al. 1989]. This leaves us with the question of where the parameterized behaviors and strategies come from. This issue, and its relationship to learning abstractions, is what I am currently focusing on [Atkeson 2020c]. I want robots to be able to rapidly acquire behaviors and strategies from other agents, and from thinking about tasks. I expect this work to primarily focus on task-level models, backed up by atomic-level models to answer the question "How well will an entirely novel strategy or a strategy from another task work on this task?" Task-level learning also makes taking advantage of complex sensor measurements including vision, auditory, and tactile sensing easier [Clarke et al. 2018, Yamaguchi and Atkeson 2016a 2019].

2) **What is the role of abstraction and symbols in reasoning and learning?** I am particularly interested in hierarchies of representation, such as modeling, planning, control, and learning at different levels of detail, ranging from finite element models of matter to concise statements of goals like "Go to the airport". It may be the case that there will be a hierarchy of task levels, and learning will occur on all of them simultaneously, similar to our hierarchical optimal control of humanoid walking.

3) **Features and representations matter:** For robot learning, features include the variables we use to describe the state and as inputs to and outputs from models, policies, and reward functions. In addition, action features are units of behavior that we can build more complex behavior out of. Action features have been reinvented many times with many names: primitives, options, chunks, macros, schemas, scripts, subroutines, subpolicies, basis functions, units of behavior, skills, etc.

Patrick Winston used to say about AI that if one described a problem with the wrong features, it was usually hard: no algorithms work well on it, and little progress is made. If one found good features, the problem becomes easy to solve, and a wide range of algorithms work well. I add to this that learning becomes much faster with good features.

Peng and van de Panne [2017] make the point that learning actions at a "higher" level rather than at the "lowest" level is often more effective: for example, learning in terms of sequences of

desired positions rather than raw actuator commands is easier. More generally, learning to select from a discrete set of policies, including policies with continuous arguments, is faster and more robust than learning how to generate a continuous stream of raw commands [Bentivegna et al. 2004].

4) **Maybe action features don't matter** as much as we think. A historical note: Policy optimization was well known to be useful early on, but no one had a good answer to the question "Where do good action features for learning a policy come from?". How do we analyze a task to select or generate appropriate action features? A big surprise is that maybe action features don't have to be carefully engineered. It turns out that essentially random segmentations of existing behavior provide reasonable action features for robot learning (for example [Liu and Hodgins 2017]). It is probably the case that random smooth behaviors work well for many tasks. This provides an interesting suggestion as to where Sutton's temporal options could come from [Sutton and Barto 2018]. The recent successes of neural-net-based reinforcement learning provide examples of learning features from large amounts of computation using models of the task.

5) **Finding better ways to solve problems (strategies):** Another motivation to explore high-level representations for behavior was the recognition that humans often find strategies to do tasks that I had not anticipated. Ask a human to teleoperate your robot to do the task for your next demo. 1) The human will get the robot to do it faster. 2) The human will get the robot do it better. 3) The human will find better ways to do it with the robot. In the DARPA Learning Locomotion Program, quadruped robots had to walk across rough terrain. Human teleoperators had no problem driving the robot across the terrain, and did it faster and better than our planners. Furthermore, they were able to contact the environment with any part of the robot's body, as in parkour. Our planning algorithms could only use the feet for contact. Giving our planners more actions to consider made them slower and thus stupider rather than smarter. Human drivers invented the "belly slide", where the robot retracted its legs and safely slid down the mountain, instead of painfully picking its way down the mountain and typically falling. My robot vacuum cleaners often push up a rug and get stuck when trying to roll onto that rug when they try to go straight on. They all then back off, turn 45 degrees and drive onto the rug at an angle, which usually works. If it doesn't, the robot backs off, drives a little further along the rug edge, and tries again. I doubt this behavior was generated by a robot planner. Instead, it was probably carefully crafted by human programmers. I also think the details of how the rug gets pushed up are not important, and thus not worth modeling.

Spending a lot of money, effort, and time on robot learning without thinking hard about strategies is usually a waste. It is better to look for easier ways to do a task than to do the task using a bad strategy. This reflects my appreciation of finding "tricks" that make a task easier.

Paddle juggling (hitting a ball vertically) provides a nice example of alternative strategies to do a task, and how being able to learn new strategies might be much more powerful than atomic-level learning. My first implementation of paddle juggling tracked the ball, predicted where it would land using a learned model, and learned a policy of how to hit it so it would land at the center of the paddle on the next bounce [Aboaf et al. 1989]. We manually designed the hit behavior, attempting to make the paddle swing have a constant upward velocity at the hit, so small timing errors would not have a big effect. Another group criticized our model-based approach, and proposed using a mirror law, so that the motion of the robot continually "reflected" the motion of the ball [Rizzi and

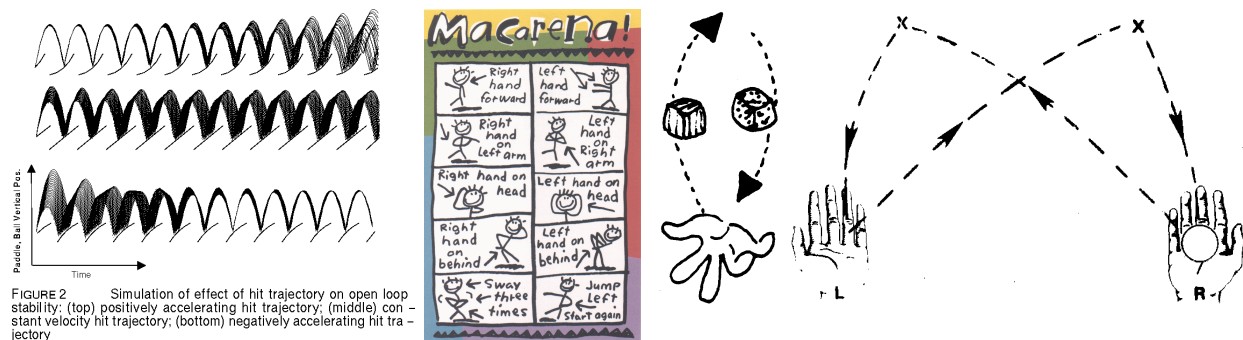

FIGURE 2          Simulation of effect of hit trajectory on open loop stability: (top) positively accelerating hit trajectory; (middle) con – stant velocity hit trajectory; (bottom) negatively accelerating hit tra – jectory

Figure 2: **Left:** Paddle juggling: Open-loop stability by slowing down during the hit trajectory. **Right:** Instructional material for doing the Macarena, and juggling. Note that the ball flights are physically incorrect in the juggling instructional material, and body and hand motions must be inferred.

Koditschek 1994], as a "Don't think, react!" and "The world is its best model" ideological stance. This synchronized the paddle with the ball, but meant the paddle was accelerating upwards at the hit, since the ball was accelerating downwards due to gravity.

This led us to realize that the characteristics of the paddle trajectory changed the nature of the task. If the paddle decelerates as it hits the ball, that stabilizes the ball flight height without any feedback or measurement of where the ball is (Figure 2). If the ball has too much energy, and lands late, it is hit with less velocity, and then has less energy. If the ball has too little energy and lands early, it is hit with more velocity, and then has more energy [Schaal and Atkeson 1993]. Adding a soft stretchy fabric paddle similar to a trampoline stabilized the horizontal directions as this type of paddle redirects the ball back towards the center of the paddle on each hit, and removes energy from any horizontal motion. These modifications of the task enabled open loop paddle juggling to work. Open loop in this case means there was no perception of the ball or feedback control based on the ball. This approach is also model-free, in that a fixed policy is executed. This is not to say that perception or feedback should be avoided, they can be added back in to make the behavior more robust. But the demands on perceptual processing are much less. Clearly the mirror law hit trajectory, which destabilizes the task, is less desirable. A similar result is true of walking, where swing leg retraction, having the swing foot moving backwards at heel strike, stabilizes the gait [Wisse et al. 2005].

We are left with the question "What learning algorithm would have discovered the alternative strategy of stabilizing trajectories?" Unless the human programmer anticipated that this feature of the task was important and parameterized the paddle or swing leg trajectory accordingly, a learning algorithm would not have been able to learn this new strategy. This is a general problem: program­mers have to anticipate what the robot will learn in the future when they choose how to represent tasks and behaviors, which reduces the value of robot learning. We do not yet have automated methods to define states, commands, and policy inputs for tasks or general behavior. Studies of human paddle juggling revealed that humans use a stabilizing hit trajectory, so good task represen­tations could be learned from demonstration using clustering and dimensionality reduction [Schaal

et al. 1996]. Unfortunately, relying on human teachers is not a general solution (there are a lot of things humans can't do well), and we should let robots be robots, and find the best strategies for robots, not humans. Generating, representing, and evaluating many possible strategies for a task is another fertile research area.

6) A related question is "**How can an agent represent, think, and learn about alternative strategies** to do a single task?" We often find that the second best strategy in simulation turns out to be the best strategy in reality. What algorithms will find the top N local optima of a function, or the top N locally optimal trajectories? It may be that some of these are trivial variations of other solutions. How is diversity enforced in the set of solutions? Dynamic programming produces only a single globally optimal policy. Policy optimization can maintain N searches for N policies, but has to enforce diversity to prevent all the policies ending up being effectively the same.

7) Another related question is **"How can an agent create or select a curriculum or design its own 'shaping' to more rapidly learn or learn more difficult tasks?"**

8) **Can deep learning be used to find good features, strategies, and curricula?** Neural networks have been a useful function approximation and representational tool for decades. The interesting claim of deep learning is that gradient descent applied to deep neural networks can actually find good features. This is worth exploring, not only for neural nets but for other representations as well. Can this approach find better task strategies, not through reasoning but through gradient descent or more general optimization of some task optimization criterion? Maybe Rosenblatt (the Perceptron guy) was right. What happens if random features are generated, and then gradient descent is applied?

9) **What can be learned?** We are now in a world where we can throw a lot of computation at a problem, and store a lot of data in large memories. Furthermore, robots can share experiences, greatly increasing the amount of training data available, and the utility of improving behavior. In situations where sets of agents solve similar problems over and over again, meta-learning, or learning to learn, has a large payoff, and is a fertile research area. There has been a great deal of work in tuning parameters of planning and learning algorithms based on data from using those algorithms [Atkeson et al. 1997a]. If we can recognize similar problems or classes of problems, we can imagine storing or caching the results of meta-learning in a library, so appropriate planning or learning algorithm parameters can be rapidly looked up when a new problem arises, and problem solving behavior (including thinking) can be specialized to the characteristics of the problem at hand. For hierarchical planners and learners this approach can be applied at each algorithmic level. This is an example of learning to learn that can be done in simulation.

It is important to keep in mind that the optimization criteria for meta-learning is reducing cost to do a task, and not some intermediate quantity such as perceptual outputs matching ground truth, or models making correct predictions [Lambert et al. 2020]. We don't care if perception or models are accurate. We only care if they help us reduce cost, and inaccurate perception and models may work better. Remember that in my work on identifying rigid-body dynamics models, I didn't care if we identified the "true" parameters. Some parameters were impossible to estimate, and others were badly estimated, because they had little effect. We found that our estimated parameters differed substantially from parameters estimated from the CAD models. So what! It just meant the parameter estimates were different in order to compensate for model structure errors.

The only thing that matters is how well models support optimal control. It is better to make a useful model than a correct model. This is why performing special calibration behaviors or trying to design maximally exciting trajectories to best identify parameters is not the right thing to do. Instead, one should try to identify model parameters from data from the task one wants to do. Try to get the training data distribution to match the test data distribution.

9A) **Cost functions can be learned:** We can use meta-learning to specialize aspects of optimization-based planning. Inverse optimal control focuses on learning more useful optimization criteria (cost or reward functions). We could imagine using inverse optimal control to learn useful cost functions from a teacher, or from watching another (potentially much slower or more expensive) agent solve problems. Note that the learned cost functions only need to generate similar plans, and could be much simpler than whatever cost functions are actually used by a teacher or another agent. This approach can be used to make agents more human-like, or adapt a particular style, by providing examples of desired behavior.

9B) **Constraints can be learned:** Constraints can be used to guide optimization-based planners. Constraints can be viewed as "stiff" terms in an optimization criterion, so meta-learning for cost functions can also learn constraints. This is especially true of soft constraints.

9C) **Behavioral units can be learned:** Meta-learning could also be used to suggest or select behavioral primitives in algorithms that select parameterized behaviors, based on context [Bentivegna et al. 2004]. The usual approach for this is clustering of observed behavior to generate candidate behaviors, followed by some form of principal component analysis to identify parameters for the candidate behaviors.

9D) **Heuristics can be learned** in algorithms that use heuristics to order search, such as A*, There is considerable work in this area. Here is the first result of a Google search for "learning heuristics" [Mittal et al. 2019], to help get you started on this topic.

9E) **Initialization parameters can be learned:** Good initial values for parameters, and good initial guesses for policies and value functions can be learned to speed up planners that refine policies and value functions. Optimization algorithms can be "warm-started" based on outcomes of previous optimizations on similar problems.

9F) **Low-level controllers can be learned:** In current state-of-the-art robot systems, actuators are controlled by "low-level" policies that are typically manually designed, and run on simple embedded microprocessors that have high bandwidth and low and constant latency communication with sensors and actuators for a particular part of the robot. Access to other sensors is typically over a computer network with variable latency, so use of those sensors is typically lower bandwidth. These policies typically have a PID servo component, where position, velocity, and integral error feedback gains provide ways for higher-level control elements to ask for desired positions, velocities, and forces. Low-level policies also typically have manually designed safety code which detects particular errors and changes control mode to handle them. Meta-learning can be used to improve on manual design for these elements.

9G) **State estimation and perception in general can be learned.** See below.

9H) **Library characteristics can be learned:** The design parameters of the library used to store or cache information can be improved with experience, as well as the organization of and any precomputed indexing for what is already in the library.

9I) **Simpler models (abstractions) can be learned.** See above.

9J) **Meta-learning plays the role of evolution** in our agent ecosystem. Presumably organisms have evolved to match the characteristics of the problems they have to solve (which is one reason why rapid climate change is so dangerous to current ecosystems). In the case of artificial agents, we can be more purposive, as meta-learning or learning to learn can be viewed as a form of intelligent design [Wikipedia 2020m].

10) **To optimize thinking as well as physical behavior, costs must be measured in the same units:** In order to fully optimize all aspects of robot learning, we need to put a comparable cost on all the work that goes into programming and learning the behavior, as well as the computational cost of perception, reasoning, and control to do the behavior. We also need a way to put a cost on uncertainty or ignorance. Once we do that, we can optimize how much time the robot spends collecting training data and practicing tasks. We can also change the overall policy to optimally balance minimizing costs and exploring to improve learned models and controllers (action policies whose output drives actuators). This is known as dual control [Wikipedia 2020n]. Here is a quick sketch as to how to put a cost on uncertainty or ignorance modeled as uncertainty. If $\mathbf{V_{xx}}$ is the local second derivative (Hessian) of the value function, then the cost of variance $\Sigma$ of the state $\mathbf{x}$ is trace$(\Sigma \mathbf{V_{xx}})$. This cost can be reduced by reducing the variance of the state, and by choosing a different action to go to a region of less curvature of the value function.

11) **Time to move on to more human-like learning:** I believe we should declare that the problem of learning policies for known tasks where we already know what strategy will be used as solved. We should declare victory and move on. It may be more productive to focus research on more human-like approaches. I believe the field of robot learning needs to address the amazing ability of humans to rapidly learn with a small number of examples, come up with a strategy or strategies to solve any problem they are presented with, and to deal with major deviations from any plan. Instructional material for humans is often missing details and wrong (Figure 2). No problem!

**Aside:** I was told when I was starting out that if I wanted to study robot learning, I should first have some (human) children and study them. It is true that children are amazing learners, and seem to need little training data (at least compared to current model-free reinforcement learning). Taking this point of view suggests several activities for new students.

12) **Robot learning should maximize prior knowledge:** I propose that humans are actually bad at learning from scratch on problems where the human has not seen similar problems previously. Instead, humans make use of experience with similar problems to solve slightly different novel problems. For example, we could have libraries of problem-solving experience that we have acquired by watching others, by being explicitly told, or from formal education. The example I like to use is opening a glass jar whose metal lid is stuck (for example [Rogers 2020]). When I ask an audience for ways to solve this problem at a talk, I get a range of answers including: increase friction (use a cloth or piece of rubber to grip the lid, or put a rubber band around the lid), heat the lid or the jar (typically by pouring hot water on the lid, or putting the entire jar in hot water), or use impact (tap the lid with a spoon or bang the lid on a countertop). When I ask the audience members whether they invented the strategy they use, the answer is almost always no (and I don't believe the people who say yes). Typically, they say that they were told how to do it, or saw someone else do it. A prominent example of humans adopting an observed strategy or style is the Fosbury

Flop in high jumping [Wikipedia 2020p], where everybody besides Fosbury learned it by watching others, or being coached. Sports are full of learned strategies (often referred to as techniques or styles). There are actually few Fosburys or Einsteins who come up with a genuinely new strategy or solution, so we should stop trying to make our robots as intelligent as Einstein, and settle for making them as smart as the average person, who learns by being told or copying. We should focus on understanding how skills are transferred, rather than on how to discover or invent skills.

What would robot learning look like if we pursued a library-based approach to learning, and built large libraries of how to solve problems? When presented with a new problem, relevant experience would be combined in some way to suggest a strategy to apply. We need to study robot learning in the context of rich libraries, rather than with little or no knowledge or prior experience [Atkeson 2020c].

13) A related question is **Can we make rich libraries work?** Given a problem to solve, can we develop algorithms that access just the right amount of relevant information from memory. Past attempts to do this led to algorithms that either find too few similar examples, or flood a planner with too many irrelevant examples. How well would you function if you decided what to do based on Google searches? That is what Siri does. How well does that work?

Our first attempts to build libraries to perceive and generate behavior have been based on storing information in trajectories (representations indexed by time). We see trajectory optimization techniques as key components of future reinforcement learning approaches, and we see reinforcement learning used to optimize trajectories and provide local control [Atkeson 1994, Atkeson and Liu 2013, Atkeson and Stephens 2008, Bellegarda and Byl 2019, Kahn et al. 2017, Levine and Koltun 2014, Liu and Hodgins 2018, Mordatch and Todorov 2014, Ota et al. 2019, Schaal and Atkeson 2010].

14A) **Do learners need to understand?** An interesting question is what role "understanding" plays in using prior knowledge. To what extent is using prior experience syntactic rather than semantic? Humans seem to acquire knowledge from watching others. They seem good at identifying the important aspects of the observed behavior. When one asks them questions about why the behavior works, the response seems to make sense. However, often the responses or theories about why something works are wrong. These errors do not impede adoption and learning of a new strategy. For example, many people say that heating a lid stuck on a new jar from the store works because metal expands more or faster than glass when heated. However, it is usually the reduction in the vacuum inside the jar due to heating which increases air pressure that releases the lid. Tapping the lid usually lets air in, and releases the vacuum, rather than breaking material gluing the lid on. It is clear that theories can play a role in identifying and adopting new strategies, but it is not clear the theories have to be correct or complete [MacMillan 2017].

14B) **Theory-free vs. theory-based AI:** It turns out trying to fix leaks is an excellent domain to show the usefulness of having theories or hypotheses to drive reasoning and learning. I am trying to stop my basement from flooding when it rains (Figure 3). Success depends on understanding where the water is coming from, and what path the water takes. Water could be coming from near the house, in which case improving the drainage by building a skirt of non-porous soil around the house that moves water about a meter away from the house would help. Or water could be coming from relatively far away through ground water. This is more likely because at one point the sewer

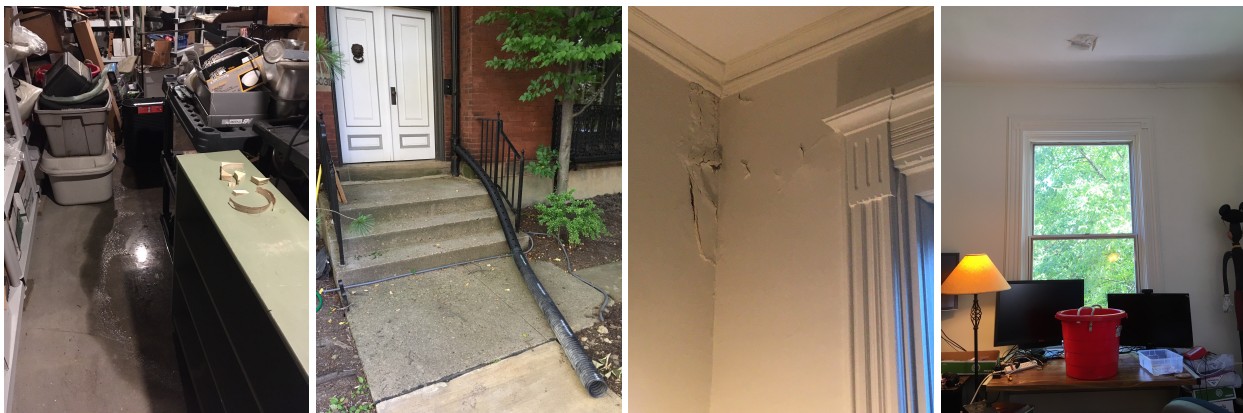

Figure 3: The learning challenges of owning an old home. Left to right: the waters 2 days after a basement flood, the downspout bypass, the bubbling plaster in the dining room, and the leak in my home office,

line from my house had to be replaced, so soil that had been packed solid was disturbed and now is much more porous, providing a pathway for ground water right up to the foundation. Water could be coming in to the basement at the joint between the wall and the floor, or through specific cracks, in which case caulking helps, or it could be coming through the porous stone foundation walls, in which case something more drastic needs to be done. Each theory suggests different actions. I have not solved this problem in many houses, so statistical approaches probably won't help me, but maybe I should ask "experts" who have more experience, or just get random good and bad advice by Googling. I need rich enough representations and reasoning abilities to understand what I see on the web.

Having the wrong theory may lead me to do actions that actually help. Having theories at all leads me to do actions that are relevant to the problem and perform useful exploration, rather than just flailing, and the behavior that results could be refined by theory-free exploration. However, my list of theories may be incomplete (see update below) and I may fail to execute key behaviors or experiments. Having alternative theories supports experimentation, where I use a hose to put water in particular places and see what happens. So far the ground water theory is winning. (Update: another pathway was a gutter downspout that ran into my concrete front steps, and then presumably through a pipe into the sewer. Just to eliminate that water path, I ran a hose into the downspout. The basement flooded. Now the other theories are less likely, so no need to move dirt or build a French drain, unless more than one theory is correct at the same time. I have installed a bypass so the water no longer goes into the pipe into the concrete, and am waiting to see if my basement stays dry (Figure 3)). Every day I pray for rain. Lots of it.

This kind of reasoning happens all the time. There is a spot in my dining room where the plaster is bubbling, a sign of water damage (Figure 3). Is the water coming in through exterior walls, so I need to point the bricks, or from leaking pipes or condensation on pipes, in which case I have to rip a hole in the wall? Yesterday my home office ceiling started dripping water well away from the walls (Figure 3). The interesting thing was that this was not on the top floor of my house.

I went upstairs and the room above was totally dry. What do I do in these cases? Old houses are a maintenance nightmare. Random exploration is expensive and frustrating. Even theory-driven exploration is expensive and frustrating, but less so, unless none of the available hypotheses are true. Note that this leads to another controversy: "**theory-free**" vs. "**theory-based**" learning and AI.

14C) **Do learners need to explain?** I am dubious that the mechanisms used by humans to explain their reasoning have much to do with the mechanisms used when reasoning. My bet is that as we understand human reasoning better, we will find that humans create an explanation after a decision is made, and completely separately, and that the true reasons why we make decisions often are not the reasons used to explain or justify those decisions. Psychiatrists and marriage counselors already know this, and the whole research area of "implicit bias" is about this disconnect. This paper is full of examples of explanations constructed years after the actual behavior. Decoupling deciding and explaining suggests that learning agents should learn to make good decisions and create good policies. They should learn to explain those decisions using different methods and representations. We don't want deciding and explaining to interfere with each other. Requiring decision mechanisms to be explainable to humans will degrade performance. In addition, explanations are often based on incorrect theories [MacMillan 2017]. However, I would like to develop a modern version of explanation-based learning [Thrun 1996].

15) **Let's start taking state estimation for robots seriously.** One relative blind spot in reinforcement learning is how to deal with imperfect feedback or measurements of what is going on when trying to control robots [Fujita and Ishii 2007, Gaon and Brafman 2019, Heess et al. 2015, McCallum 1996, Morimoto and Doya 2007, Whitehead and Lin 1995], Another blind spot is the assumption that the state vector is complete, and that there are no unmodeled dynamics. (see work on Partially Observable Markov Processes (POMDPs)). What happens if measurements are noisy? What happens if aspects of the full state of a situation are not measured? Dealing with noisy and limited measurements by explicitly doing state-of-the-art **state estimation** was **the** critical step to getting robot bipeds to walk robustly [Xinjilefu 2015, Xinjilefu and Atkeson 2012, Xinjilefu et al. 2015]. In general, a big part of learning is figuring out what aspects of the world are relevant to a task. Sutton and Barto [2018] in section 17.3 Observations and State, lists this area as one of the frontiers of reinforcement learning.

In reinforcement learning papers it is common to confuse the state of a system with what is measured or observed, and use a vector of measurements as an input to a policy, and to imply or refer to a limited set of measurements as the state vector. State is everything one could know that could improve prediction of the future. An example of the distinction between measurements and state is when only position measurements are taken, and velocities have to be estimated by differencing positions. In this case the measurements are a set of positions, and for mechanical systems the state is often positions and velocities of all the parts. A policy to balance the cart-pole system needs to take into account velocities, even if only the cart position and pole angle are measured. Past measurements need to be included in the inputs to a policy, if direct velocity measurements are not available, or the policy has to have internal state that remembers past position inputs. If only the cart position is measured, the pole angle as well as the velocities, or surrogates for those quantities, need to be estimated. This is also true if images of the cart-pole are used

to measure the cart-pole configuration, instead of joint angle sensors, and applies to the use of computer vision for control more generally. When I see single images used to control mechanical systems where momentum matters, I look for internal state in the policy, so proxies for velocity and thus momentum can be estimated and used for control.

The distinction between full state feedback and what is known as output feedback (using only the measurements as inputs to the policy) is important. What one can do with full state feedback greatly exceeds what can be done with output feedback, unless the policy has internal state. A state estimator is one way to add human comprehensible internal state to a policy. Also, there are good control or policy design techniques for full state feedback. Output feedback policies are typically designed using policy optimization. As a design process, optimization is more difficult to control and get what you really want. These are reasons why explicit state estimators are used to convert output feedback to full state feedback. As reinforcement learning matures, I expect that we will borrow techniques from the rich literature on state estimation in "traditional" engineering.

These issues get more complex when we need to estimate what humans will do next. For self driving cars, explicitly estimating the intent of other drivers is a major challenge. We would at least like to have probability distributions of what other cars will do next, and ideally know the policies that other drivers are following. These are based on estimating the internal or belief states of the other drivers.

Adding state estimation to reinforcement learning, and learning what the state vector is and how to estimate state while simultaneously learning a control policy are areas I would definitely recommend to new students. Given access to a limited set of measurements and the ability to do experiments, how can a learner construct a state vector (state vectors are not unique)? In addition, the reward may depend on hidden variables (aspects of the state vector that are not measured). How are the hidden variables made explicit? More generally, how can an agent simultaneously do perceptual learning and action learning?

The counterproposal is to give all the observations to a deep reinforcement learner with internal state, such as an LSTM network, and have it figure out what latent representations are useful.

16) **Model how well you know what you know, and what you don't know:** In order to reason about risk, predict the distribution of rewards, or even just provide a probability of success, an agent needs to represent and learn about uncertainties and modeling error. Models should be used to learn to predict process dynamics. Separate models should be used to model prediction error as well as reward prediction, value function, and policy errors, and how these errors depend on states and actions. Ideally, an agent would be able to predict reasoning errors and uncertainty as well. I call this approach "model$^2$", where we maintain models of models.

It is important that we don't confuse the parameter sensitivity of our representations with the true uncertainty of a prediction, independent of how a model is represented. Modeling errors should have nothing to do with how a model is represented. For example, perturbing weights of a neural net representation and looking at how outputs vary has nothing to do with the true modeling error. Trying to represent uncertainty as distributions of neural network weights doesn't make sense. Chua et al. [2018] discuss these issues in the context of neural network models in model-based reinforcement learning.

17) **How can model-based planning be used to handle modeling errors?** Dealing with the

Sim2Real gap can be viewed as specializing a pre-planned behavior to a particular situation. In game playing, learning a model of a particular opponent may be useful, for example. To the extent that uncertainty can be modeled, model-based planning can help handle uncertainty as well, The Kalman filter and most particle filters are examples of model-based techniques to handle uncertainty. [Peng et al. 2020] (discussed in [Atkeson 2020a]) describes an interesting approach to online system identification, which is pre-planned in advance using models. [Atkeson 2012] describes another approach to plan for a range of modeling errors in advance based on hypothesizing multiple models and optimizing a single policy using the sum of the costs on all of the alternative models.

18) **Observation: Reinforcement learning, the whole point of which is to avoid specific goals, may be most useful when given a specific goal.** It is interesting to note the increasing use of learning from demonstration (a.k.a. learning from observation or imitation) to guide reinforcement learning (for example [Atkeson 1998, Peng et al. 2020]).

19) **Model-free learning is unlikely to become sample efficient enough for online learning for most problems.** Sometimes motivations for research are negative, you want to prove someone else wrong. So, prove me and the following wrong: It is unlikely that model-free reinforcement learning applied directly to a real robot will ever be sample efficient enough to learn any complex behavior from scratch (perhaps this even defines complex behavior). This requires orders of magnitude reduction in the need for training data. We should stop beating our heads against that particular brick wall, and choose a different wall to work with.

The only way I can see learning a high complexity policy with little built-in domain knowledge working is if there is a way to regularize the policy, and start off with few policy degrees of freedom. As performance improves and more data is collected, the number of degrees of freedom of the policy can be gradually increased.

An example of where model-free approaches are actually useful directly applied to real robots is when humans are involved (human-in-the-loop optimization). For example, optimizing wearable robot behavior involves humans, who are poorly modeled if modeled at all. Here is some work on online model-free optimization of the behavior of an ankle exoskeleton, where a small number of parameters (4) were learned in less than 100 trials over roughly two hours [Zhang et al. 2017]. This worked because the parameterized policy was carefully chosen using domain knowledge.

20) **Why are robots so slow to do tasks, and to learn?** In the DARPA Robotics Challenge robots took an hour to do a set of tasks that a child could do in a few minutes (at most). It is true that self-driving cars and drones dodging obstacles have to act fast, but watching most academic robot demos is like watching paint dry.

21) **Why don't reinforcement learning folks worry about stability, as control theory folks do?** Is it that current work in reinforcement learning is based on vision, which usually has a low sampling rate and the robots move slowly? Or that low-level control policies to control the robot joint angles are manually designed by the robot manufacturer and not altered by learning? Can we better understand the relationship between reinforcement learning and adaptive control as practiced in control theory? See Appendices 1 and 2 for a more extensive discussion of this issue.

22) **Value driven exploration:** Random exploration is inefficient, as the agent spends time learning about things that don't matter in that they don't affect the policy. Can we be smarter about

exploration? Can exploration be purposive? Deterministic?

**Going Further:** I want to provide some entry points into relevant literature. First, I need to point out some distractions. If one Googles "Model-based learning", one finds at least two other meanings of the phrase, besides using a model to learn more effectively. It also refers to human learners who form mental models while they learn, which is of great concern in education, and to machine learning approaches that synthesize models specific to a problem [Bishop 2018, Winn et al. 2020]. There is also a large literature in psychology and neuroscience about model-based and model-free decision mechanisms in humans and other animals [Sutton and Barto 2018].

Back to robots: There are talks and classes on model-based reinforcement learning on YouTube [Finn 2017, Fragkiadaki 2019, Levine 2019, Poupart 2018, Silver 2015, Sutton 2019]. Recht [2019] excellently surveys reinforcement learning and strongly advocates model-based reinforcement learning. This is an excellent survey of issues in reinforcement learning in general [Irpan 2018]. Here is a recent workshop on the use of machine learning in model-based control [Mesbah and Houska 2020]. This is a nice survey of current issues in deep model-based learning [Janner 2019]. Here is a recent survey of model-based reinforcement learning in robotics [Polydoros and Nalpantidis 2017] and machine learning [Ray and Tadepalli 2010]. To find more recent papers, just Google "model-based reinforcement learning".

# 5    Some final thoughts ...

Much to my surprise, the **Robot Revolution** is happening now, in my lifetime. Things are moving much faster than I ever expected.

Learning will be used to reduce engineering costs, not replace engineering. It is important to keep in mind that robot behavior that is common, important, or risky is likely to be carefully engineered, and that learning might tune it or slightly improve it, rather than create it. For this reason, the emphasis in robot reinforcement learning on learning legged locomotion from scratch does not make much sense. If you want your work to be used broadly on real robots doing real jobs, figure out how to help engineers, not replace them. This is true of high performance legged robots such as Atlas or recent quadrupeds, and is likely to be true for self driving cars. As the field of robot learning matures, it will look more and more like traditional engineering.

On the academic side of robot learning, anything goes. Resist the siren call of knowledge-free learning. Join us to figure out how to achieve more human-like learning in the context of rich experience, personal knowledge, and cultural knowledge. Build a system that can learn from reading this paper, listening to a lecture, reading a textbook, or reading instructions on how to build a piece of furniture.

# Appendix 1: What are model-based and model-free reinforcement learning?

There is a confusion in reinforcement learning (RL) between using a model (model-based) or not (model-free) and algorithms that require a model (model-based) or can use a model or not (currently called model-free, which is a misnomer). Bertsekas [2019] refers to model-based and model-free *implementations* of various reinforcement learning algorithms. Bertsekas [2019] also views using a simulator or model to just produce state-action-outcome samples as model-free, while Sutton and Barto [2018] view this as model-based (Dyna), and I refer to it as weakly model-based. This writeup attempts to sort this mess out, and suggest new naming conventions.

There is a second confusion in RL that also makes this discussion difficult. I want to describe algorithms that compute the value function $V(\mathbf{x})$ as value function approaches, but often Q functions ($Q(\mathbf{x}, \mathbf{u})$) are also referred to as value functions. For the purposes of this discussion, value function means only $V(\mathbf{x})$ and not $Q(\mathbf{x}, \mathbf{u})$. In analogy to Q learning, value function approaches will be referred to as V learning. Sutton and Barto [2018] refer to $V(\mathbf{x})$ as the state-value function, and $Q(\mathbf{x}, \mathbf{u})$ as the action-value function, which is not bad terminology, except it does not match the older engineering literature. Bertsekas [2019] uses the term Q-factors for values of $Q(\mathbf{x}, \mathbf{u})$. Another common term is Q-values.

Usually value function approaches that update a representation of the value function $V(\mathbf{x})$ (V learning) are referred to as model-based because one needs a random access model to compute an arbitrary update. Trajectory-based optimization typically also require a model [Atkeson 1994, Atkeson and Stephens 2008, Schaal and Atkeson 2010, Atkeson and Liu 2013, Levine and Koltun 2013]. Q function approaches (Q learning) are usually referred to as model-free, but can be accelerated by using a model, so one could have a model-based Q function algorithm (as we will describe in this writeup). Policy gradient approaches are also usually referred to as model-free, but can be accelerated by using a model, so one could have a model-based policy optimization algorithm (also described in this writeup). Function optimization approaches to policy optimization can operate in the real world (model-free) or use a simulator or other model of a task (model-based).

The terms "model-based" and "model-free" should be reserved for describing whether an algorithm takes advantage of the additional information a model can provide in addition to samples along a trajectory of $(\mathbf{x}_k, \mathbf{u}_k, \mathbf{x}_{k+1})$, such as being able to efficiently estimate the derivatives of the dynamics, estimate an expected value of a random variable, propagate a probability distribution through time, represent and manipulate uncertainty over time, or choose which state and action to sample from when performing backups to estimate updated elements of value (V) or Q functions.

This writeup shows that value function, Q function, and policy gradient learning algorithms are basically the same for a linear system with quadratic costs, or a system modeled locally along a trajectory with local Taylor series approximations of nonlinear dynamics and more complex costs, independent of whether models are used or not (model-based vs. model-free). This is useful for trajectory-based reinforcement learning approaches.

We will assume the same ground truth model for value function, Q function, and policy gradient examples. Let's consider optimizing a policy for one time step of a deterministic discrete time system ($\mathbf{x}$ is the state and $\mathbf{u}$ are the actions) with linear dynamics ($\mathbf{x}_{k+1} = \mathbf{f}(\mathbf{x}_k, \mathbf{u}_k) = \mathbf{A}\mathbf{x}_k + \mathbf{B}\mathbf{u}_k$) and quadratic cost which is the sum of the one step cost $L(\mathbf{x}, \mathbf{u}) = \mathbf{x}^T\mathbf{Q}\mathbf{x} + \mathbf{u}^T\mathbf{R}\mathbf{u}$ and a terminal

penalty $\mathbf{x}^\mathrm{T}\mathbf{V_{xx}}\mathbf{x}$ where $\mathbf{Q}$, $\mathbf{R}$, and $\mathbf{V_{xx}}$ are symmetric positive semidefinite matrices. Eventually we will interpret $\mathbf{V_{xx}}$ as the second derivative (Hessian) with respect to the state of the value function $\mathrm{V}(\mathbf{x})$. This formulation is a version of a Linear Quadratic Regulator (LQR) formulation [Wikipedia 2020f], which was also examined in the model-free reinforcement learning context by [Bradtke 1994, ten Hagen and Kröse 1998] and in the general reinforcement learning context by Recht [2019]. Amos et al. [2018] discuss how to differentiate through receding horizon control using LQR. Marco et al. [2017] used model-based knowledge of the form of the cost function relating policy parameters to total cost to speed up a function optimization approach to "model-free" policy optimization.

The cost of this one-step task is:

$$cost = \mathbf{x}^\mathrm{T}\mathbf{Qx} + \mathbf{u}^\mathrm{T}\mathbf{Ru} + (\mathbf{Ax} + \mathbf{Bu})^\mathrm{T}\mathbf{V_{xx}}(\mathbf{Ax} + \mathbf{Bu}) \tag{1}$$

Value function (V, not Q) reinforcement learning (V learning) will compute $\partial cost/\partial \mathbf{u} = 0$ to find the optimal command, which results in an optimal policy, which turns out to be a deterministic linear function of the state.

$$\frac{\partial cost}{\partial \mathbf{u}} = 0 = 2(\mathbf{R} + \mathbf{B}^\mathrm{T}\mathbf{V_{xx}}\mathbf{B})\mathbf{u} + 2\mathbf{B}^\mathrm{T}\mathbf{V_{xx}}\mathbf{Ax} \tag{2}$$

$$\mathbf{u} = -(\mathbf{R} + \mathbf{B}^\mathrm{T}\mathbf{V_{xx}}\mathbf{B})^{-1}\mathbf{B}^\mathrm{T}\mathbf{V_{xx}}\mathbf{Ax} = \mathbf{Kx} \tag{3}$$

$$\mathbf{K} = -(\mathbf{R} + \mathbf{B}^\mathrm{T}\mathbf{V_{xx}}\mathbf{B})^{-1}\mathbf{B}^\mathrm{T}\mathbf{V_{xx}}\mathbf{A} \tag{4}$$

Note that, as predicted by the internal model principle, the model of the controlled system $(\mathbf{A}, \mathbf{B})$ is embedded in the optimal policy [Conant and Ashby 1970, Wonham 1975].

Q function reinforcement learning optimizes $\mathrm{Q}(\mathbf{x}, \mathbf{u})$ with respect to $\mathbf{u}$. We will present a simple version of how to do this, leaving out all the algorithmic tricks to make it work better, such as using experience replay and target representations. Let's parameterize $\mathrm{Q}(\mathbf{x}, \mathbf{u}, \mathbf{q})$ as the inner product of a vector of weights $\mathbf{q}$ and a feature vector $\phi(\mathbf{x}, \mathbf{u})$, so $\mathrm{Q}(\mathbf{x}, \mathbf{u}, \mathbf{q}) = \mathbf{q}^\mathrm{T}\phi(\mathbf{x}, \mathbf{u})$. $\phi(\mathbf{x}, \mathbf{u})$ is a vector of all linear and quadratic terms made up of products of the elements of $\mathbf{x}$ and $\mathbf{u}$, and 1. This representation can be trained using Temporal Difference (TD) learning in the model-free policy iteration case. Here is a SARSA style update ($\gamma$ is the discount factor, and $\alpha$ is a step size for gradient descent.) [Sutton and Barto 2018]:

$$\delta_k = \mathrm{L}(\mathbf{x}_k, \mathbf{u}_k) + \gamma\mathrm{Q}(\mathbf{x}_{k+1}, \mathbf{u}_{k+1}, \mathbf{q}_k) - \mathrm{Q}(\mathbf{x}_k, \mathbf{u}_k, \mathbf{q}_k) \tag{5}$$

$$\mathbf{q}_{k+1} = \mathbf{q}_k + \alpha\delta_k(\nabla_\mathbf{q}\mathrm{Q}(\mathbf{x}_k, \mathbf{u}_k, \mathbf{q}_k))^\mathrm{T} \tag{6}$$

$$\mathbf{q}_{k+1} = \mathbf{q}_k + \alpha\delta_k\phi(\mathbf{x}_k, \mathbf{u}_k) \tag{7}$$

Vectors by default are vertical, and the derivative of a scalar with respect to a vector generates a horizontal vector, hence the transpose. If $\mathrm{Q}(\mathbf{x}, \mathbf{u}, \mathbf{q})$ is parameterized in some more complex way, one uses Equation 6 to update $\mathbf{q}$. We are going to need some exploration to generate a rich distribution of $\mathbf{x}$ and $\mathbf{u}$, so we are going to add noise to $\mathbf{u}$. This means we have to use an off-policy deterministic policy gradient [Silver et al. 2014, Lillicrap et al. 2016]. This changes how

we compute $\delta_k$ above, based on the current deterministic policy $\mathbf{u} = \pi(\mathbf{x})$, where the policy is represented by a lookup table:

$$\delta_k = \mathbf{L}(\mathbf{x}_k, \mathbf{u}_k) + \gamma \mathbf{Q}(\mathbf{x}_{k+1}, \pi(\mathbf{x}_{k+1}), \mathbf{q}_k) - \mathbf{Q}(\mathbf{x}_k, \mathbf{u}_k, \mathbf{q}_k) \tag{8}$$

Improved actions at $\mathbf{x}$ are given by $\operatorname{argmin}_{\mathbf{u}} \mathbf{Q}(\mathbf{x}, \mathbf{u}, \mathbf{q})$. This optimization is straightforward since $\mathbf{u}$ appears in a known way and at most quadratically in $\phi(\mathbf{x}, \mathbf{u})$. It is possible to do batch computations of the gradient of $\mathbf{q}$, as long as states are included in the batch with the same distribution as the states visited. One can think of this as a form of experience replay.

Now let's look at the model-based version of Q learning. We know that $\mathbf{Q}(\mathbf{x}, \mathbf{u}) = cost$. By computing $\partial \mathbf{Q}(\mathbf{x}, \mathbf{u})/\partial \mathbf{u} = 0$ we can find the optimal policy directly, without gradient descent. We can rewrite the inner product $\mathbf{q}^\mathrm{T}\phi(\mathbf{x}, \mathbf{u})$ as a quadratic expression, and then we end up doing the same thing we did in the value (V) function case, so we get the same answer:

$$\mathbf{Q}(\mathbf{x}, \mathbf{u}) = \mathbf{x}^\mathrm{T}(\mathbf{Q} + \mathbf{A}^\mathrm{T}\mathbf{V}_{\mathbf{xx}}\mathbf{A})\mathbf{x} + \mathbf{u}^\mathrm{T}(\mathbf{R} + \mathbf{B}^\mathrm{T}\mathbf{V}_{\mathbf{xx}}\mathbf{B})\mathbf{u} + 2\mathbf{u}(\mathbf{B}^\mathrm{T}\mathbf{V}_{\mathbf{xx}}\mathbf{A})\mathbf{x} \tag{9}$$

$$\frac{\partial \mathbf{Q}(\mathbf{x}, \mathbf{u})}{\partial \mathbf{u}} = 0 = 2(\mathbf{R} + \mathbf{B}^\mathrm{T}\mathbf{V}_{\mathbf{xx}}\mathbf{B})\mathbf{u} + 2\mathbf{B}^\mathrm{T}\mathbf{V}_{\mathbf{xx}}\mathbf{A}\mathbf{x} \tag{10}$$

$$\mathbf{u} = -(\mathbf{R} + \mathbf{B}^\mathrm{T}\mathbf{V}_{\mathbf{xx}}\mathbf{B})^{-1}\mathbf{B}^\mathrm{T}\mathbf{V}_{\mathbf{xx}}\mathbf{A}\mathbf{x} = \mathbf{K}\mathbf{x} \tag{11}$$

$$\mathbf{K} = -(\mathbf{R} + \mathbf{B}^\mathrm{T}\mathbf{V}_{\mathbf{xx}}\mathbf{B})^{-1}\mathbf{B}^\mathrm{T}\mathbf{V}_{\mathbf{xx}}\mathbf{A} \tag{12}$$

Policy gradient algorithms use gradient descent with respect to the policy parameters to optimize the policy. Policy optimization enables us to choose any policy structure (policy parameterization) we think will work well. This is a way to sneak domain knowledge into the formulation. We will present a simple version of how to do this, again leaving out all the algorithmic tricks to make it work better. In this example we will choose the parameterization of the policy to be linear with respect to the state $\pi(\mathbf{x}, \boldsymbol{\theta}) = \mathbf{K}\mathbf{x}$, since we already know the optimal policy is linear in the state. The elements of the $m \times n$ matrix $\mathbf{K}$ are the unknown policy parameters $\boldsymbol{\theta}$. $m$ is the dimensionality of $\mathbf{u}$, and $n$ is the dimensionality of $\mathbf{x}$. Actually, we are going to formulate this in an equivalent way that simplifies what follows: $\boldsymbol{\theta} = \operatorname{vec}(\mathbf{K})$ where $\operatorname{vec}()$ stacks the columns of $\mathbf{K}$ in one large column vector [Wikipedia 2020j]. In this case:

$$\pi(\mathbf{x}, \boldsymbol{\theta}) = \mathbf{K}\mathbf{x} = (\mathbf{x} \otimes \mathbf{I}_m)\boldsymbol{\theta} \tag{13}$$

$$(\mathbf{x} \otimes \mathbf{I}_m) = \begin{pmatrix} \mathbf{x}^\mathrm{T} & \mathbf{0} & \dots & \mathbf{0} \\ \mathbf{0} & \mathbf{x}^\mathrm{T} & \dots & \mathbf{0} \\ \vdots & \vdots & \ddots & \vdots \\ \mathbf{0} & \mathbf{0} & \dots & \mathbf{x}^\mathrm{T} \end{pmatrix}_{m \times mn} \tag{14}$$

$$\boldsymbol{\theta} = \operatorname{vec}(\mathbf{K}) = (\mathbf{K}_{1,1}, \mathbf{K}_{2,1}, \dots, \mathbf{K}_{m,1}, \mathbf{K}_{1,2}, \mathbf{K}_{2,2}, \dots, \mathbf{K}_{m,2}, \dots, \mathbf{K}_{1,n}, \mathbf{K}_{2,n}, \dots, \mathbf{K}_{m,n})^\mathrm{T} \tag{15}$$

$\otimes$ forms a tensor product [Wikipedia 2020i].

In the model-free actor-critic case $Q(\mathbf{x}, \mathbf{u}, \mathbf{q})$ can again be trained using Temporal Difference (TD) learning (see above). Given $V(\mathbf{x}, \boldsymbol{\theta}) = Q(\mathbf{x}, \pi(\mathbf{x}, \boldsymbol{\theta}), \mathbf{q})$, improved actions are given by gradient descent with the gradient:

$$\nabla_{\boldsymbol{\theta}} V(\mathbf{x}, \boldsymbol{\theta}) = \nabla_{\mathbf{u}} Q(\mathbf{x}, \mathbf{u}, \mathbf{q}) \Big|_{\mathbf{u} = \pi(\mathbf{x}, \boldsymbol{\theta})} \nabla_{\boldsymbol{\theta}} \pi(\mathbf{x}, \boldsymbol{\theta}) \tag{16}$$

$$= \nabla_{\mathbf{u}} Q(\mathbf{x}, \mathbf{u}, \mathbf{q}) \Big|_{\mathbf{u} = \pi(\mathbf{x}, \boldsymbol{\theta})} \frac{\partial \left( (\mathbf{x} \otimes \mathbf{I}_m) \boldsymbol{\theta} \right)}{\partial \boldsymbol{\theta}} \tag{17}$$

$$= \nabla_{\mathbf{u}} Q(\mathbf{x}, \mathbf{u}, \mathbf{q}) \Big|_{\mathbf{u} = \pi(\mathbf{x}, \boldsymbol{\theta})} \mathbf{x} \otimes \mathbf{I}_m \tag{18}$$

Again, it is possible to do batch computations of the gradient of $\boldsymbol{\theta}$, as long as states are included in the batch with the same distribution as the states visited.

Now let's look at the model-based version of policy optimization. We say policy optimization instead of policy gradient because the optimal policy is computed without iteration and after only one gradient calculation. Model-based policy optimization calculates a policy for a region without sampling state-action-outcome transitions, since that information is in the known or learned model. At the optimum the derivative of the cost with respect to the policy parameters is zero. Substituting $\mathbf{Kx}$ for $\mathbf{u}$:

$$cost = \mathbf{x}^{\mathrm{T}} \mathbf{Q} \mathbf{x} + (\mathbf{Kx})^{\mathrm{T}} \mathbf{R} \mathbf{Kx} + (\mathbf{Ax} + \mathbf{BKx})^{\mathrm{T}} \mathbf{V}_{\mathbf{xx}} (\mathbf{Ax} + \mathbf{BKx}) \tag{19}$$

$$\frac{\partial cost}{\partial \mathbf{K}} = 0 = 2\mathbf{x}^{\mathrm{T}} ((\mathbf{R} + \mathbf{B}^{\mathrm{T}} \mathbf{V}_{\mathbf{xx}} \mathbf{B}) \mathbf{K} + \mathbf{B}^{\mathrm{T}} \mathbf{V}_{\mathbf{xx}} \mathbf{A}) \mathbf{x} \tag{20}$$

$$\mathbf{K} = -(\mathbf{R} + \mathbf{B}^{\mathrm{T}} \mathbf{V}_{\mathbf{xx}} \mathbf{B})^{-1} \mathbf{B}^{\mathrm{T}} \mathbf{V}_{\mathbf{xx}} \mathbf{A} \tag{21}$$

One benefit of algorithms that optimize a parametric policy is that policies applied to multiple models simultaneously can be optimized. Optimizing policies over a range of models simultaneously is a way to try to make policies more robust without sacrificing too much performance. Atkeson [2012] describes the details of how model-based policy optimization can be applied to multiple models simultaneously, in a generalization of model-based policy optimization. These models each have their own state vectors which can be of different sizes and have no relationship to state vectors of other models. This allows models with different model structures and simulated unmodeled dynamics to be included in the optimization.

I have just presented five reinforcement learning algorithms: model-based V learning, model-free and model-based Q learning, model-free policy gradient, and model-based policy optimization. These are all forms of reinforcement learning. Value (V) learning approaches need to use a model. Q learning doesn't need to use models, but benefits from them. There is both model-free (policy gradient) and model-based policy optimization. Model-free approaches use iterative gradient descent and samples of state-action-outcome transitions. Model-based approaches calculate the optimal policy directly without iterations, without the need of transition samples from the robot, and for a region. If the LQR model is global, the calculated policy is globally optimal. If the LQR model represents a region with locally linear dynamics (no discontinuities) and locally quadratic one step and terminal costs, the policy is optimal for that region, and we refer to it as a local policy. Atkeson [1994] describes how local policies along trajectories can be used to build a nonlinear globally optimal policy.

Value (V) learning and Q learning don't commit to a policy structure, but simply provide training data for supervised training of some policy structure such as a lookup table or deep neural network. You could pick multiple policy structures to train, for example, and see which one has the least error, or progress from simple to complex parameterizations as estimates of the policy and/or value/Q function converged.

Policy gradient approaches must commit to a policy structure before learning. Atkeson [1994] describes a nonparametric representation for policies that avoids committing to a particular policy structure, but this approach does not preserve the advantages a low complexity policy can provide. It should be possible to start with a low complexity policy to get fast initial learning, and then switch over to more complex policy structures during learning by using the current policy to generate training data to initialize the new policy, in a model-free way in that a model is not used. Using a model can greatly speed up training a new policy, because the model can be used to update entire regions of state space in one update. Unlike current deep reinforcement learning approaches which start out with complex representations, I would suggest starting with simple model and policy structures with few parameters, and then increasing the complexity of these representations by adding new terms or simply replacing the current models and policies. Unlike supervised learning, the size and complexity of the training data increases with time in reinforcement learning.

One can apply these approaches to trajectories, starting at the end of the trajectory (latest point in time) and working towards the beginning (earlier points in time). The parameters of the one step cost ($\mathbf{Q}$ and $\mathbf{R}$) and the system dynamics ($\mathbf{A}$ and $\mathbf{B}$) can change on each step. The terminal penalty $\mathbf{V_{xx}}(\mathbf{x})$ is updated to the optimal cost of the solved step expressed as a function of the state at that time step. In the example presented above, the terminal function for the previous step is given by a Riccati equation [Wikipedia 2020f]:

$$\mathbf{V_{xx}}_k = \mathbf{Q}_k + \mathbf{K}_k^{\mathrm{T}}\mathbf{R}_k\mathbf{K}_k + (\mathbf{A}_k\mathbf{x} + \mathbf{B}_k\mathbf{K}_k)^{\mathrm{T}}\mathbf{V_{xx}}_{k+1}(\mathbf{A}_k\mathbf{x} + \mathbf{B}_k\mathbf{K}_k) \tag{22}$$

This is just an application of the chain rule, or backpropagation through a model and through time, and replaces Temporal Difference (TD) learning. It does operate backwards in time, and so must operate on trajectories rather than individual state-action-outcome samples. Temporal difference learning operates forward in time since it builds a more global model of the value function in the form of a Q function. Atkeson [1994 1998], Atkeson and Liu [2013], Atkeson and Morimoto [2003], Atkeson and Stephens [2007 2008], Jacobson and Mayne [1970b], Levine and Koltun [2013], Liu et al. [2013], Morimoto et al. [2003], Schaal and Atkeson [2010], Tedrake [2009], Wikipedia [2020a], Yamaguchi and Atkeson [2015a 2016cd] discuss trajectory-based reinforcement learning in more detail.

For nonlinear dynamics the Differential Dynamic Programming (DDP) equations replace the LQR equations given here, see [Jacobson and Mayne 1970b, Wikipedia 2020a]. Local linear models of dynamics and local quadratic models of cost functions can model many complex systems and the approach is general. Model, value function and policy discontinuities can be handled with the local model approach [Atkeson 1994]. Additional discrete state variables can also be used to manage discontinuities such as in-contact and not-in-contact, and handle hybrid systems with both continuous and discrete states.

So if all of these RL approaches get the same answers whether you use models or not, why are we arguing about whether to use a model or not? In the model-free case where you don't

have or want to use a model, the environment provides training data and you simply don't need a model. Zhang et al. [2017] is an example of not using a model while using function optimization to optimize a policy (another form of RL). Just a small number of parameters (4) were learned in less than 100 trials over roughly two hours. The approach worked because the parameterized policy was carefully chosen using domain knowledge.

Is there a difference between using a model (model-based) and not using a model (model-free) in reinforcement learning in terms of performance limits or computational cost? The performance of policy gradient approaches is potentially limited by the choice of policy representation. Low complexity policy representations (such as linear policies) can rapidly improve performance initially, but may have limited ultimate performance. High complexity policies learn slowly, but can have high ultimate performance. Model-based approaches can use nonparametric representations of policies that become more complex as more training data is obtained and the complexity of the model increases, as training the model is a form of supervised learning [Atkeson et al. 1997ab].

Another potential benefit of using a model is reducing the computational cost of calculating derivatives. $\mathbf{A}$ and $\mathbf{B}$ are derivatives of the dynamics $\mathbf{f}(\mathbf{x}, \mathbf{u})$, and $\mathbf{Q}$ and $\mathbf{R}$ are second derivatives of a more general one-step cost $\mathbf{L}(\mathbf{x}, \mathbf{u})$. Models enable computing these derivatives analytically if the form of the model is known, while model-free approaches have to compute them from $N +$ 1 samples, where $N$ is the sum of the dimensionality of the states and actions. If analytically computing the derivative of the model is expensive, a model-based approach will probably also compute the derivative numerically based on samples, but for simple models this is a factor of $N + 1$ advantage of using a model. Model-based approaches do a better job at generating consistent and continuous derivatives along a line in state space.

A third benefit of using a model is reducing the computational cost of computing expectations in the stochastic case. A model can sometimes be used to analytically compute or approximate expectations of random variables in the above equations because the probability distributions are known, but without a model one has to estimate expectations by sampling from the distributions, which is often much slower. For example, in the case described above, the analytic cost is:

$$cost = \mathbf{x}^{\mathrm{T}}\mathbf{Q}\mathbf{x} + \mathbf{u}^{\mathrm{T}}\mathbf{R}\mathbf{u} + (\mathbf{A}\mathbf{x} + \mathbf{B}\mathbf{u})^{\mathrm{T}}\mathbf{V_{xx}}(\mathbf{A}\mathbf{x} + \mathbf{B}\mathbf{u}) + \mathrm{trace}(\mathbf{\Sigma}\mathbf{V_{xx}}) \qquad (23)$$

where $\mathbf{\Sigma}$ is the variance of $\mathbf{x}_{k+1}$, a result of any additive process noise [Wikipedia 2020h]. The additional term $\mathrm{trace}(\mathbf{\Sigma}\mathbf{V_{xx}})$ is ignored in the LQR case because it cannot be changed, $\mathbf{\Sigma}$ and $\mathbf{V_{xx}}$ are constants. However, in the nonlinear dynamics or non-quadratic cost cases, this term can change the choice of actions, choosing target states that can be achieved with lower variance, or portions of the space that have a smaller Hessian (curvature) of the value function. The term $\mathrm{trace}(\mathbf{\Sigma}\mathbf{V_{xx}})$ can be computed analytically in model-based approaches if the form of the model is known, but it must be estimated through sampling in model-free approaches.

A major computational cost difference difference between model-based and model-free approaches is that model-based approaches can update value functions and policies over regions instead of points, while current formulations of model-free approaches only update at points. Trajectory-based region updates using local Taylor series approximations to the model, value function, and policy based on the approaches described in this writeup have been proposed several times over the years [Atkeson 1994 1998, Atkeson and Liu 2013, Atkeson and Morimoto 2003, Atkeson

and Stephens 2007 2008, Levine and Koltun 2013, Liu et al. 2013, Morimoto et al. 2003, Schaal and Atkeson 2010, Tedrake 2009, Yamaguchi and Atkeson 2015a 2016cd]. These model-based approaches can be extended to modeling Q functions, as well as model-based policy optimization [Atkeson 2012]. Another way to speed up value and Q function updates is to randomly sample a small number of actions on each update [Atkeson 2007].

See the body of the paper for a more complete discussion of the pros and cons of model-based and model-free approaches.

So when should one use model-based vs. model-free reinforcement learning? I am recommending using both simultaneously if possible, as I am recommending learning parametric and nonparametric models simultaneously. One can capture the benefits of both approaches in a hybrid approach. In general, model-based approaches do well when the model is known or easy to learn, and model-free approaches do well when the policy representation is simpler than the model structure (such as a global linear policy and a nonlinear model). However, even with a low complexity policy, value functions may need to be represented at high resolution.

So here is one last attempt to resolve the confusion about "model-based" and "model-free" RL. RL algorithms that have to use a model and thus are definitely model-based RL:

- Algorithms that update a representation of the value function $V(\mathbf{x})$ using some variant of the Bellman equation involving $V(\mathbf{x})$.

- Algorithms that do trajectory optimization, or use some variant of LQR or DDP, such as Bubbles [Atkeson 1994 1998, Atkeson and Liu 2013, Atkeson and Morimoto 2003, Atkeson and Stephens 2007 2008, Liu et al. 2013, Morimoto et al. 2003, Schaal and Atkeson 2010, Yamaguchi and Atkeson 2015a 2016cd], LQR trees [Tedrake 2009], and Guided Policy Search [Levine and Koltun 2013].

Algorithms that can either use a model (and thus be model-based X) or not use a model (and thus be model-free X) are listed below. If I was ideologically rigid, I would call anything that used a simulator model-based. I would call anything that explicitly fit a model to data and used that to generate new samples $(\mathbf{x}_k, \mathbf{u}_k, \mathbf{x}_{k+1})$ model-based as well. But who would be so rigid? **These are not model-free algorithms, in that they can take advantage of a model.**

- Function optimization: Optimize policy parameters by running trials. (CMA-ES, CEM, your favorite function optimization algorithm goes here ...). Most work in this area uses a faster than real time simulator. Here is a truly model-free approach using no simulators [Zhang et al. 2017]. Doing things like using a known model or fitting a model to compute derivatives of performance vs. policy parameters or taking derivatives of your code are ways to apply models to speed up the optimization process.

- Q learning: anything based on a Q function, whether it explicitly also represents a policy or not. As shown in this writeup, LQR and DDP support the use of models to speed up Q learning.

- Policy optimization algorithms. Atkeson [2012] shows how to use a model to efficiently compute policy gradients along a trajectory.

So what terminology makes sense? Here is some suggested nomenclature:

- **Completely model-free** - no simulator, no learned model, and no use of a model.

- **Simulator-based** - uses a simulator.

- **Weakly model-based** - uses a learned model to generate training data.

- **Strongly model-based** or **Model-based** - maximizes use of a known or learned model.

Referring to the above terms as X, here are names for algorithms: V learning, Trajectory-based RL, X function optimization, X Q learning, and X policy optimization.

# Appendix 2: Unmodeled Dynamics

This appendix provides examples of the effect of unmodeled dynamics.

## A Simple Example [Atkeson 2012]

Consider a nominal linear plant which is a double integrator with the following discrete time dynamics (the sampling rate $T = 0.001s$):

$$\begin{pmatrix} \theta \\ \dot{\theta} \end{pmatrix}_{k+1} = \begin{pmatrix} 1 & T \\ 0 & 1 \end{pmatrix} \begin{pmatrix} \theta \\ \dot{\theta} \end{pmatrix}_k + \begin{pmatrix} 0 \\ T \end{pmatrix} \begin{pmatrix} u \end{pmatrix}_k \tag{24}$$

$\theta$ is the joint angle of a robot arm, and $\dot{\theta}$ is the joint velocity. $u$ is the joint torque.

In this example the feedback control law has the structure $\mathbf{u} = \mathbf{Kx}$. An optimal Linear Quadratic Regulator (LQR) is designed for the nominal double integrator plant with a one step cost function of $Ł(\mathbf{x}, \mathbf{u}) = 0.5(\mathbf{x}^{\mathrm{T}}\mathbf{Qx} + \mathbf{u}^{\mathrm{T}}\mathbf{Ru})$. In this example

$$\mathbf{Q} = \begin{pmatrix} 1000 & 0 \\ 0 & 1 \end{pmatrix} \qquad \mathbf{R} = \begin{pmatrix} 0.001 \end{pmatrix} \tag{25}$$

resulting in feedback gains of $\mathbf{K} = [973 \ 54]$. The planned step response for the nominal model from 1 to 0 is shown as the blue "nominal w/ original p(parameters)" trace in Figure 4. The position moves smoothly from 1 to 0 in a little more than 100ms.

The actual plant is also linear with the following unmodeled dynamics: a second order low pass filter on the input with a cutoff of 10 Hz. The transfer function for the unmodeled dynamics is $\omega^2/(s^2 + 2\gamma\omega s + \omega^2)$, with a damping ratio $\gamma = 1$ and a natural frequency $\omega = 20\pi$. There is no resonant peak and the unmodeled dynamics acts as a well behaved low pass filter. However, when the feedback gains above is applied to the actual system (black dotted line with legend "actual w/ original p") the system goes unstable with increasing oscillations (Figure 4). Any optimization or reinforcement learning algorithm that correctly optimizes the feedback gains given the nominal model and the cost function will get the same answer, and the policy will go unstable when applied to the actual system. This is the effect of model structure error (unmodeled dynamics) which cannot be replicated by using a stochastic transition function.

One way to design a robust control law is to optimize the parameters of the control law (in this case the position and velocity feedback gains) by evaluating them on several different models. The control law is simulated for a fixed duration $D$ on each of $M$ models for $S$ initial conditions (so that a reasonable distribution of states can be sampled), and the cost of each trajectory ($V_m(\mathbf{x}_s, \mathbf{p})$) using the original one step cost function is summed for the overall optimization criterion:

$$C = \sum_{m=1}^{M} \sum_{s=1}^{S} w(m, s) V_m(\mathbf{x}_s, \mathbf{p}) \tag{26}$$

$w(m, s)$ is a weight on each trajectory, $\mathbf{x}_s$ is the $s$th initial condition, and $\mathbf{p}$ are the parameters of the policy (in this case vec($\mathbf{K}$)). We will suppress the $m$ subscript on $V$ to simplify our results. We

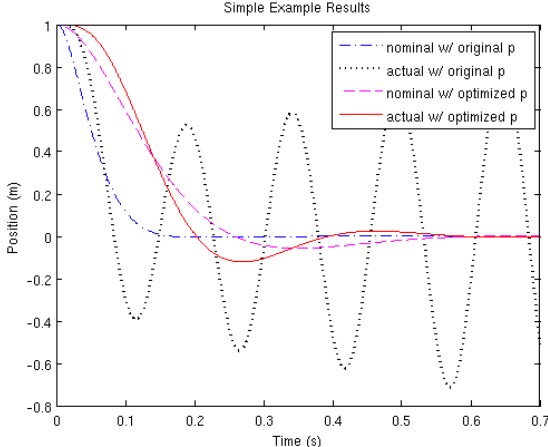

Figure 4: Simulations of the simple example.

assume each trajectory is created using the appropriate model and use the appropriate model dynamics to calculate derivatives of $V$ in what follows. First and second order gradients are summed using the same weights $w(m, s)$ as were used on the trajectories.

Optimizing the control law of the example for the nominal model and a model with similar unmodeled dynamics (second order low pass filter on the input with $\omega = 10\pi$ and $\gamma = 0.5$) with initial conditions $(1, 0)$ results in feedback gains of $[148\ 17]$ which are stable for the actual plant $(\omega, \gamma) = (20\pi, 1)$. Figure 4 shows simulations of the various conditions: the blue dot-dashed line is the nominal plant with the original gains $[973\ 54]$, and the black dotted line shows the actual plant with the original gains, which is unstable. The magenta dashed line shows the nominal plant with the gains optimized for multiple models $[148\ 17]$, and the red solid line shows the actual plant with the same gains. The multiple model gains are less aggressive than the original gains, and the actual plant is stable and reasonably well damped.

A model with the same model structure as the actual plant does not have to be included in the set of models used in optimization. Optimizing using the nominal double integrator model and the nominal model with a pure input delay of 50 milliseconds results in optimized gains of $[141\ 18]$, which provide about the same performance on the actual plant as the previous optimized gains. In addition, the new gains are stable for double integrator plants with delays up to 61 milliseconds, while the original gains of $[973\ 54]$ are stable for delays up to 22 milliseconds. We note that the nominal double integrator model, the model with an input filter, and the model with a delay all have different model structures (number of state variables for example), which a multiple model policy optimization approach should handle.

We believe using multiple models with different model structures is more effective than generating a distribution of models by varying model parameters. With different model structures we can easily add delay and various forms of model structure error.

We have also applied the method of common random numbers (which has been reinvented many times and is also known as correlated sampling, matched pairs, matched sampling, and Pegasus) [Heikes et al. 1976, Ng and Jordan 2000]. An array of random numbers is created, and

| Noise level | Maximum stable delay (msec) |
|---|---|
| 0 | 22 |
| 1 | 22 |
| 10 | 22 |
| 100 | 22 |
| 1000 | 30 |
| 10000 | 19 |
| 100000 | 16 |

Table 1: Input noise level vs. robustness.

that same array is used on each simulation to perturb the simulation, typically by adding process noise to the plant input **u**. This is an alternative approach to robust controller design through policy optimization. On the simple example, we found that the added process noise needed to be huge relative to the commands needed to move the arm ($\pm 1000$ uniformly distributed on each time step) for the generated controller to work reliably on the actual plant with the input filter with a cutoff of 10 Hz. However, there was only a narrow window of noise levels that worked reliably, and higher and lower levels of noise produced unstable controllers quite often. Table 1 shows how adding input noise to the input of the double integrator during optimization affects robustness to delays in the double integrator. Again, $\pm 1000$ uniformly distributed noise added to **u** on each time step provides the most robustness. However, this maximum robustness is less than that provided by optimizing with multiple models. We found that added noise is not a reliable proxy for unmodeled dynamics.

## A Real Robot Example [Bentivegna et al. 2007]

A linear model of a hip joint (fore-aft swing) of a Sarcos Primus System Humanoid is described. We can use a linear model since all the the other joints of the robot are fixed, and the robot pelvis is immobilized on a stand. Slides from a talk on system identification of the joint are presented as a series of figures. Think of this as a comic book. I even show some robot snuff porn. Check it out!

The joint is more complex than a simple joint torque driving rigid body links. This model tries to capture the actuator dynamics of the valve which controls oil flowing in and out of the piston and the dynamics of that flow. Even after our best efforts, we still have unmodeled dynamics, which show up as unexpected oscillations when we move the joint.

The model is presented in Figures 6, 7, and 8. We verify the model by comparing its frequency response to the actual frequency response of the robot (Figure 9). Then we look at step responses under LQR control (Figure 10). Looks good, but oscillations show that there are still unmodeled dynamics. Then we look at simulated torque control just using torque feedback (Figure 11). It doesn't work well. We have to take the actuator dynamics in to account. We can't really do torque step responses on the actual robot since that will cause the leg to kick into the mechanical stops. We would need to lock the joint in some way, which then means leg movement (changing position, and non-zero velocity) is not tested. We test torque control by commanding a zero torque and letting the leg swing freely like a pendulum (Figure 12). Some stepping in place data is shown, which is

very nice except for glitches where the actuator saturates (Figures 13-17). We use error correction learning to swing the leg and look at the command and joint torque, which look strange (Figure 18). We then tried a slightly faster trajectory and the "tendon" for the joint failed mid-movement. We show some robot snuff porn (Figure 19).

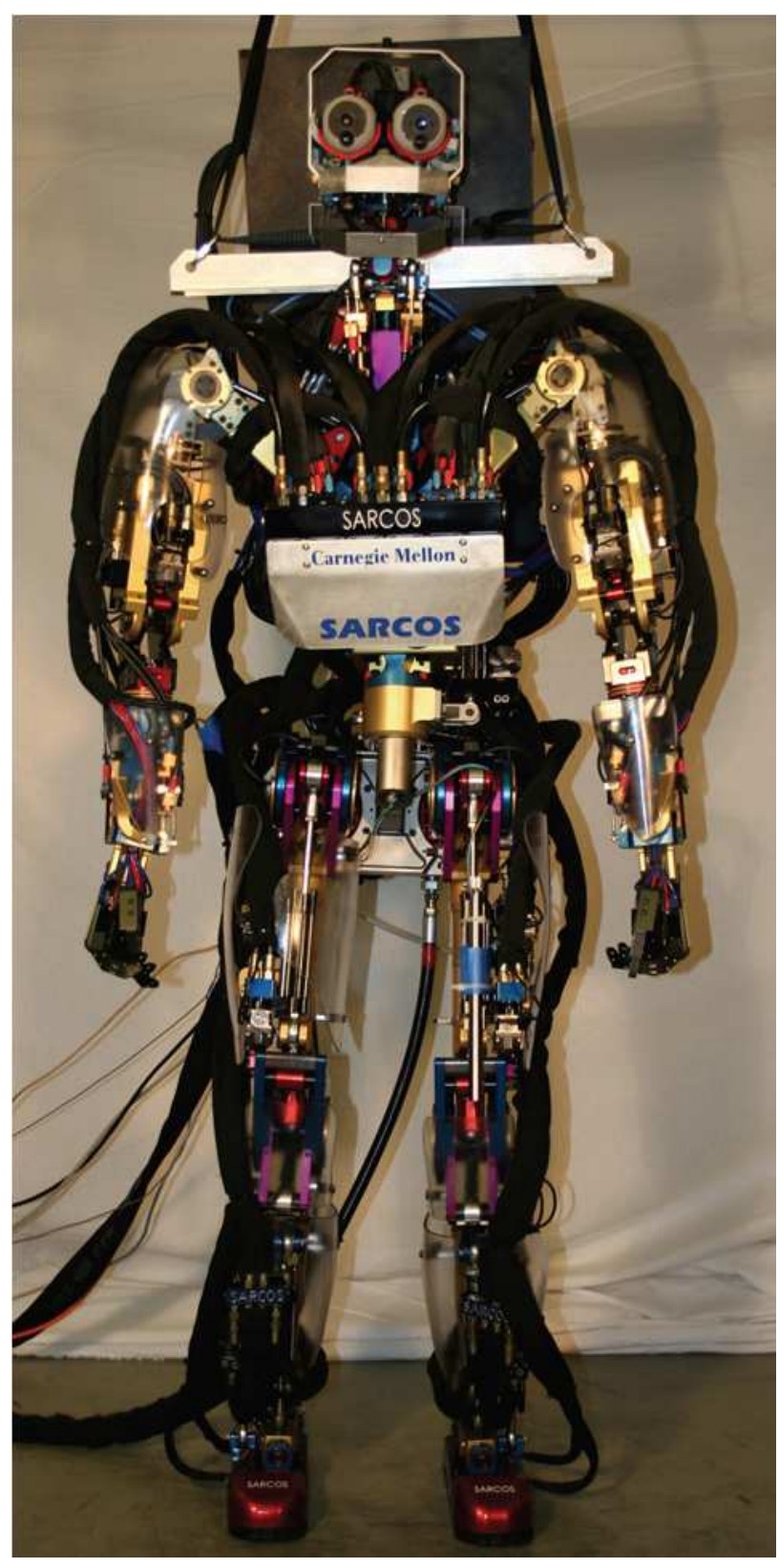

Figure 5: The Sarcos Primus System Humanoid we are going to model.

# Relevant components

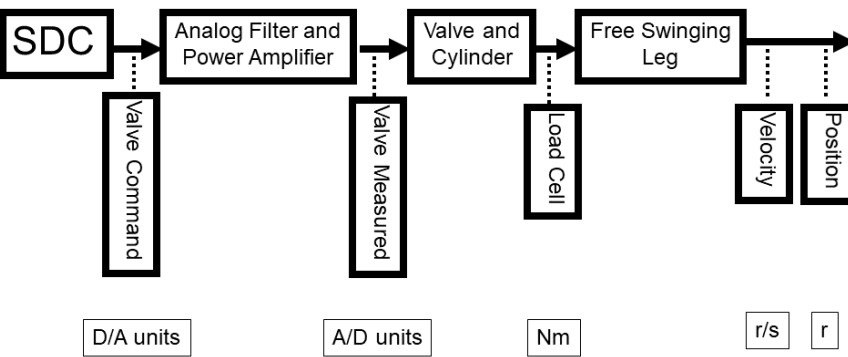

Figure 6: The components of a joint. The SDC is the microprocessor controlling the joint. It outputs a valve command, which is turned into current ("Valve Measured") into the valve, which lets oil into the piston and generates force ("Load Cell"), which cause the leg to accelerate, resulting in a changed velocity and position.

# Identified linear state space model

- State is $x_k$ = [ position velocity load valve-measured ]$^T$
- Command is $u_k$ = [ valve-command ]
- $x_{k+1} = A*x_k + B*u_k$ Timestep = 0.001s

$$B = \begin{bmatrix} 0 \\ 0 \\ 0 \\ 0.116 \end{bmatrix}$$

$$A = \begin{bmatrix} 1 & 0.001 & 0 & 0 \\ -0.0142 & 0.99902 & 0.00022 & 0 \\ 0 & -4.4413 & 0.98586 & 0.00197 \\ 0 & 0 & 0 & 0.884 \end{bmatrix}$$

Figure 7: Here is an identified model of the left hip fore-aft joint.

## Explanation of model

- State is $x_k$ = [ position velocity load valve-measured $]^T$
- Command is $u_k$ = [ valve-command ]
- $x_{k+1} = A*x_k + B*u_k$

$$B = \begin{bmatrix} 0 \\ 0 \\ 0 \\ 0.116 \end{bmatrix}$$

Velocity integrates to position

$$A = \begin{bmatrix} 1 & 0.001 & 0 & 0 \\ -0.0142 & 0.99902 & 0.00022 & 0 \\ 0 & -4.4413 & 0.98586 & 0.00197 \\ 0 & 0 & 0 & 0.884 \end{bmatrix}$$

## Explanation of model

- State is $x_k$ = [ position velocity load valve-measured $]^T$
- Command is $u_k$ = [ valve-command ]
- $x_{k+1} = A*x_k + B*u_k$

$$B = \begin{bmatrix} 0 \\ 0 \\ 0 \\ 0.116 \end{bmatrix}$$

Rigid body dynamics (stiffness due to gravity, damping, and mass terms)

$$A = \begin{bmatrix} 1 & 0.001 & 0 & 0 \\ -0.0142 & 0.99902 & 0.00022 & 0 \\ 0 & -4.4413 & 0.98586 & 0.00197 \\ 0 & 0 & 0 & 0.884 \end{bmatrix}$$

## Explanation of model

- State is $x_k$ = [ position velocity load valve-measured $]^T$
- Command is $u_k$ = [ valve-command ]

$$B = \begin{bmatrix} 0 \\ 0 \\ 0 \\ \end{bmatrix}$$

Second order system due to piston/oil movement. Valve opening causes oil flow, which integrates to pressure (load), which integrates to velocity. "Stiffness" is due to piston movement affecting flow similarly to valve opening. "Damping" is due to the effect of differential pressure (load) on flow. "Mass" is due to oil flow integrating to pressure.

$$A = \begin{bmatrix} 1 & 0.001 & 0 & 0 \\ -0.0142 & 0.99902 & 0.00022 & 0 \\ 0 & -4.4413 & 0.98586 & 0.00197 \\ 0 & 0 & 0 & 0.884 \end{bmatrix}$$

## Explanation of model

- State is $x_k$ = [ position velocity load valve-measured $]^T$
- Command is $u_k$ = [ valve-command ]
- $x_{k+1} = A*x_k + B*u_k$

$$B = \begin{bmatrix} 0 \\ 0 \\ 0 \\ 0.116 \end{bmatrix}$$

Analog filter and power amplifier act as first order low pass filter

$$A = \begin{bmatrix} 1 & 0.001 & 0 & 0 \\ -0.0142 & 0.99902 & 0.00022 & 0 \\ 0 & -4.4413 & 0.98586 & 0.00197 \\ 0 & 0 & 0 & 0.884 \end{bmatrix}$$

Figure 8: Explanation of the terms of the model.

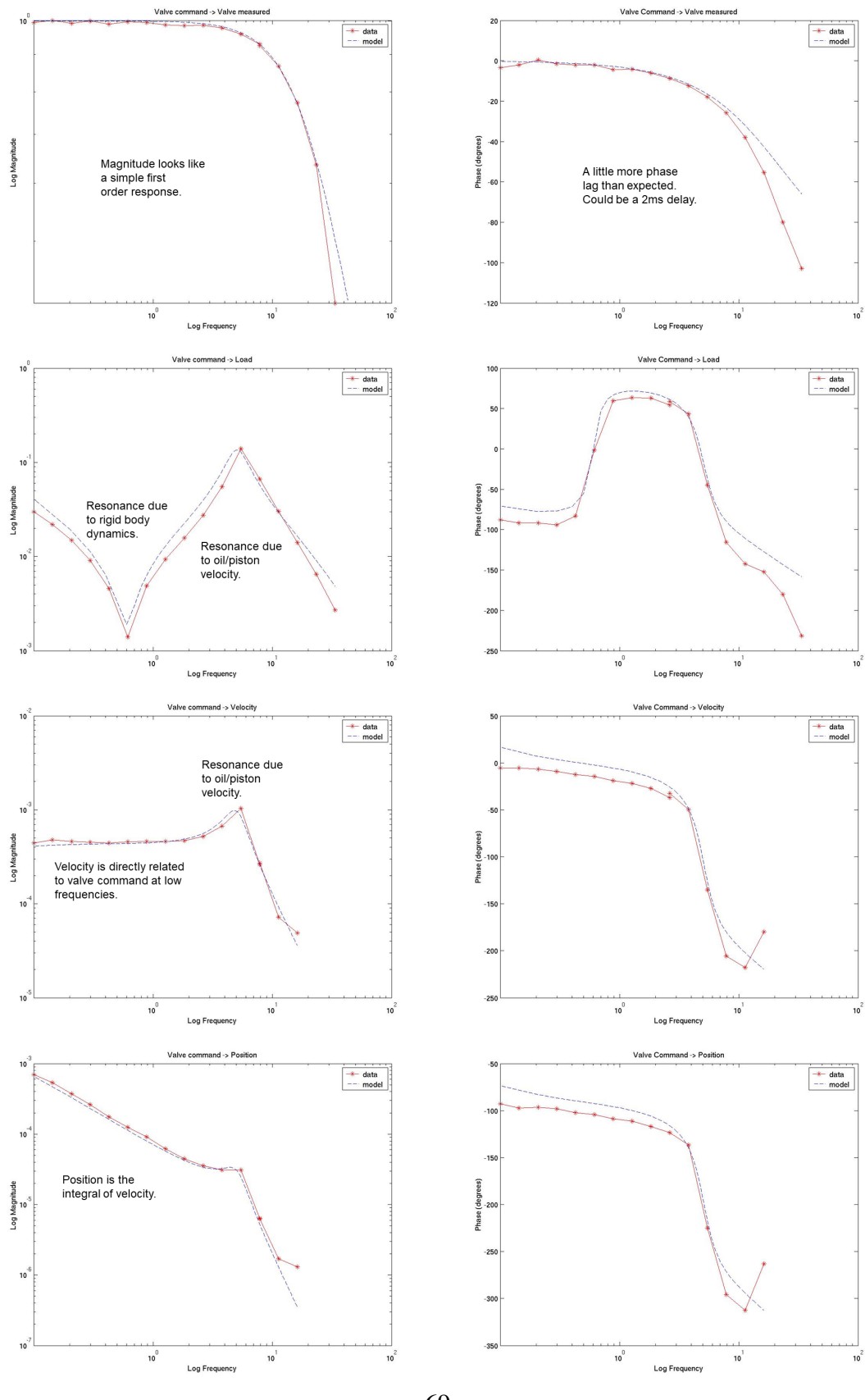

Figure 9: Frequency responses of the identified model compared to the actual robot. On the left are magnitude plots, and on the right are phase plots. The rows are the current into the valves ("valve measured"), the joint torque sensor, the joint velocity, and the joint position. This is a good fit up to about 10Hz. The joint did not respond much to frequencies higher than that, so we did not try to accurately model in that region. A constant phase offset at low frequencies is ok.

# Position Control LQR Design

- We can improve position control using full state feedback.
- K2 in next slide is best PD gain design. Limited by instability.
- K1 is full state LQR design given below.

$$Q = \begin{bmatrix} 1e6 & 0 & 0 & 0 \\ 0 & 3e3 & 0 & 0 \\ 0 & 0 & 0 & 0 \\ 0 & 0 & 0 & 0 \end{bmatrix} \quad R = \begin{bmatrix} 1e-7 \end{bmatrix}$$

$$K = \begin{bmatrix} 94156 & 5299.5 & 36.149 & 0.49774 \end{bmatrix}$$

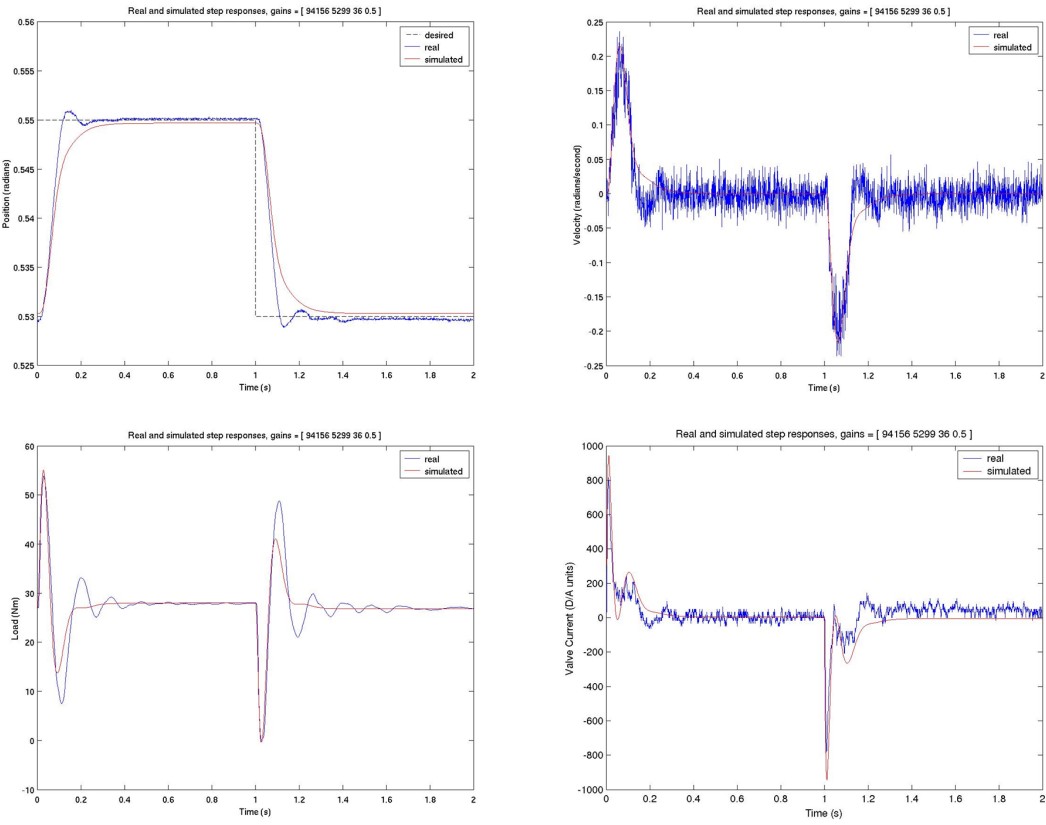

Figure 10: Position control using the full model and an LQR design. The first row of data is position and velocity, and the second row is joint torque ("Load (Nm)"), and the actual valve command. Note that the real joint moves faster than we had planned (see position plot), and there are a few oscillations after the movement, especially in the joint torque trace ("Load (Nm)") at about 8Hz. This model is pretty good, but there is still some unmodeled dynamics out there generating the oscillations. Note how noisy the velocity is. This is an example where we let the robot be the robot, and put feedback gains on joint torque and the valve current, rather than try to obey some idealized textbook model where only position and velocity feedback are used.

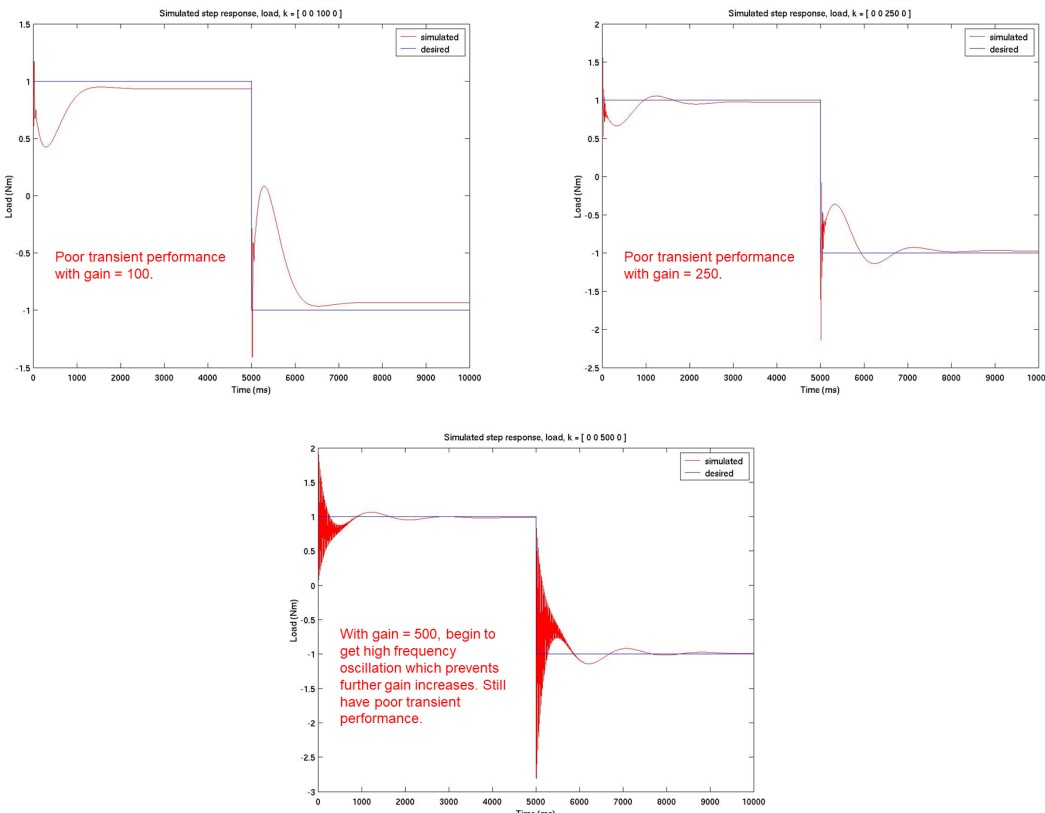

Figure 11: Simulated torque control where we just use torque feedback valve-command = gain*torque-error, applied to the identified model. Gains are 100, 250, and 500. This is an idealized textbook approach to force control. It doesn't work well. We can't avoid the poor transient performance, and at a gain of 500 we are getting a lot of oscillation. At about a gain of 700 the real joint goes unstable.

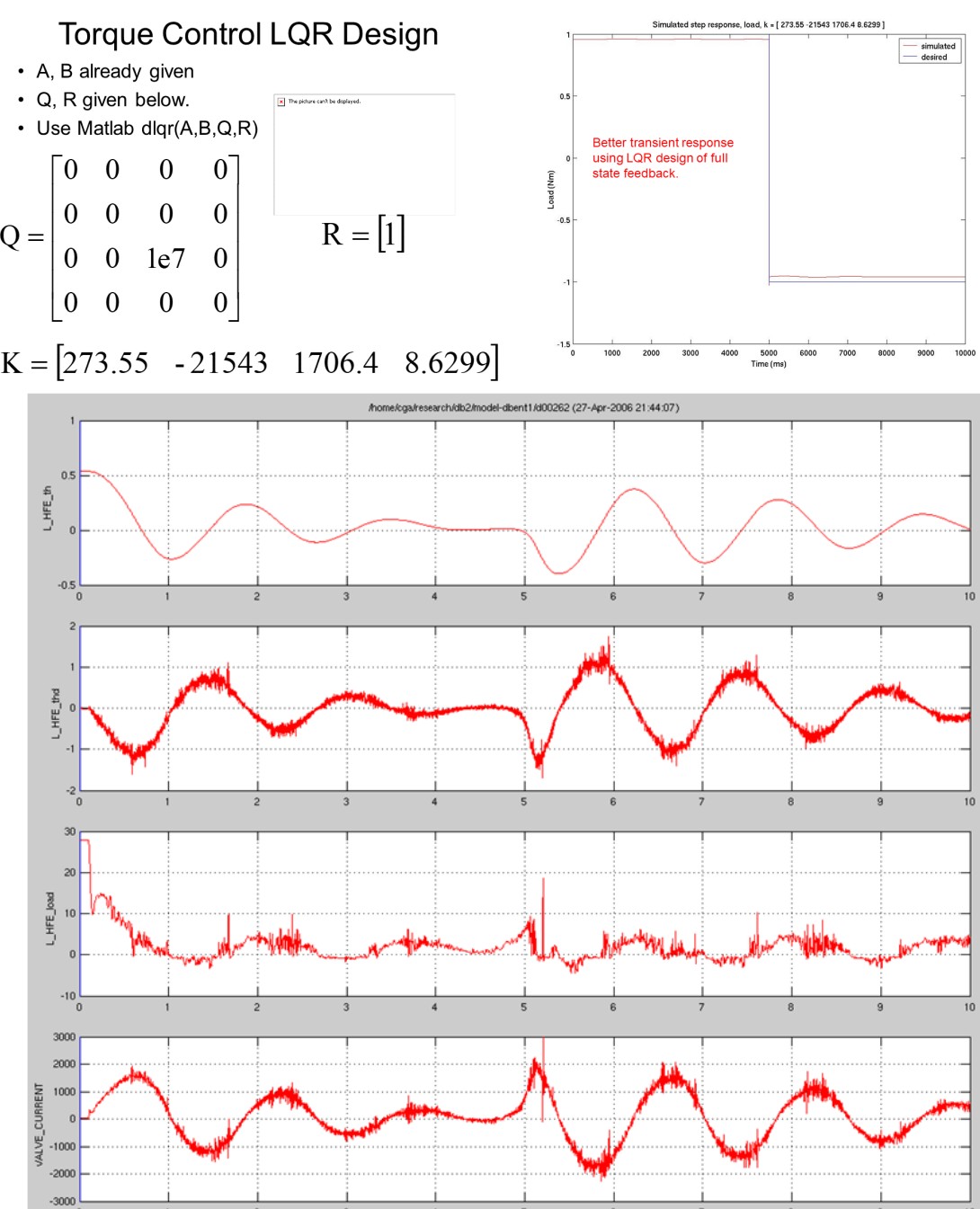

Figure 12: Torque control on the real robot using the full model and an LQR design. Here we are using the full model to do torque control. It works perfectly in simulation, as it should (upper right). "Simulation is doomed to succeed". In the bottom row the data plots show the joint swinging freely after being pushed at 0 and 5 seconds. The torque should be zero. It is close (mostly in the range ±5Nm). It is not perfect, as the leg swing loses energy when it should swing forever if there were perfect torque control. What looks like high frequency noise is quite repeatable, and thus is unmodeled dynamics. This is another example where we let the robot be the robot, and put feedback gains on joint torque and the valve current, rather than try to obey some idealized textbook model where only position and velocity feedback are used.

# HAA: Stepping in place.
## blue: left actual, green: left desired, red: right actual, cyan: right desired.

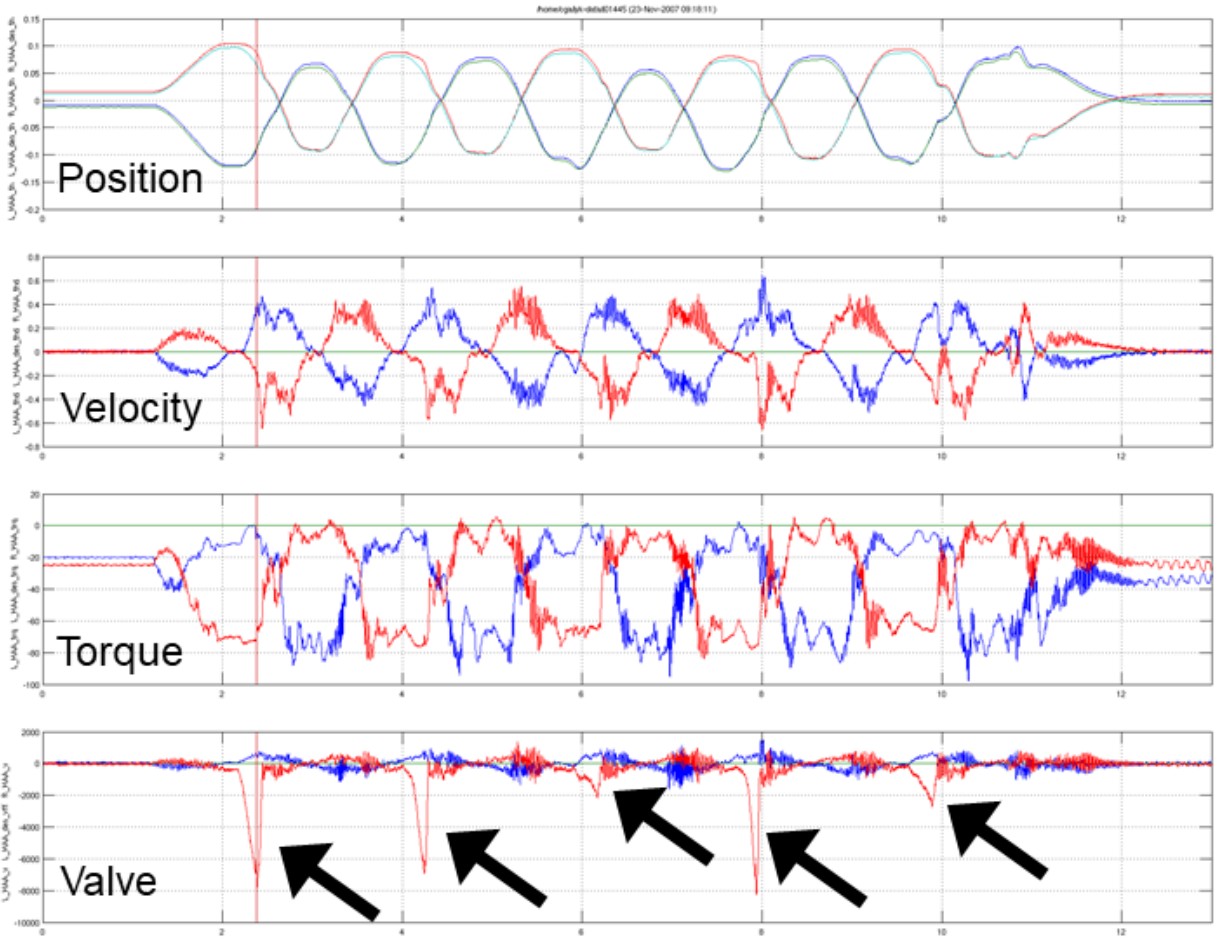

Figure 13: Hip sideways movement during stepping in place. Since we are close to actuator limits, sometimes the valve command becomes very large (arrows). The mid-frequency oscillations (about 4Hz) in torque are unmodeled dynamics. What looks like high frequency noise is quite repeatable, and thus is also unmodeled dynamics. Robust policies have to deal with this.

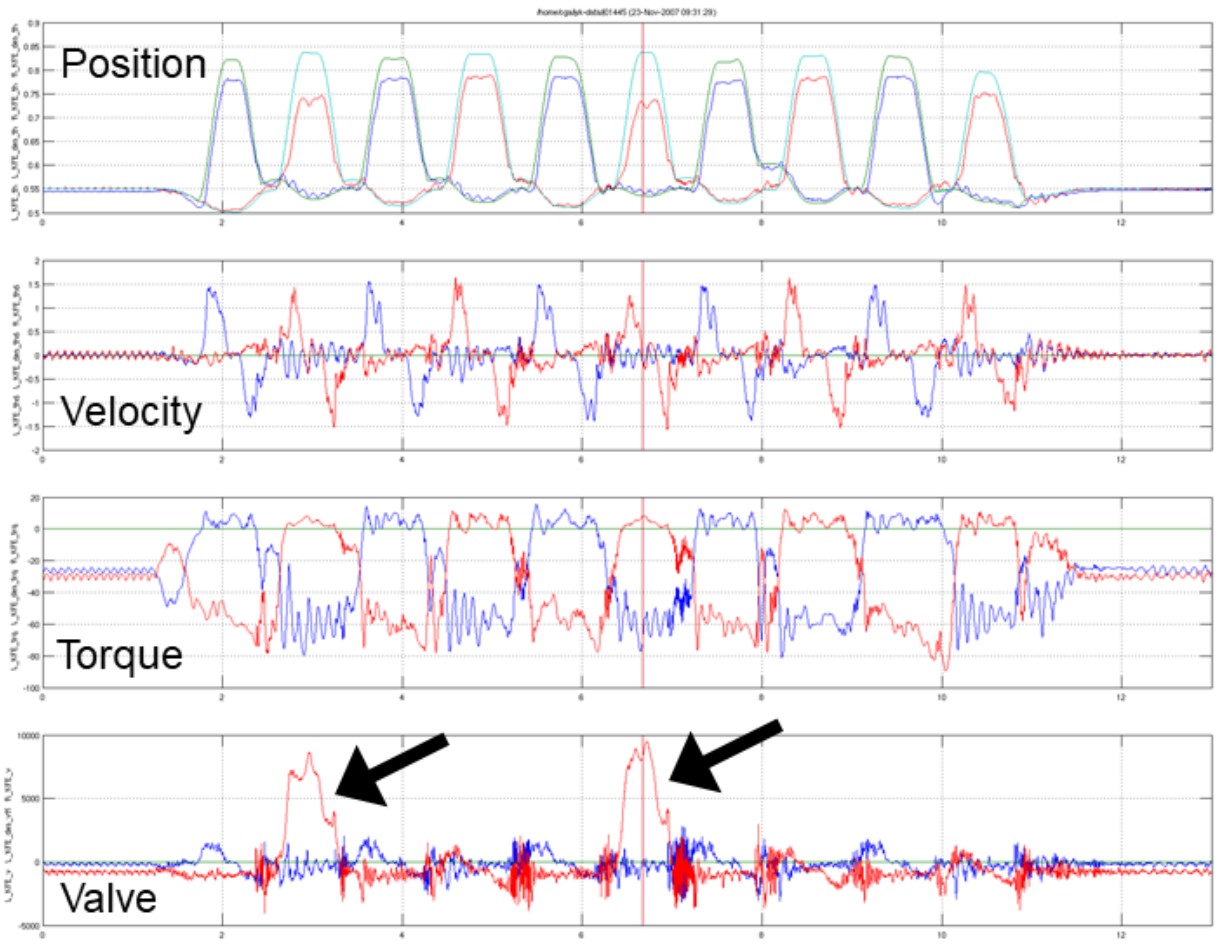

KFE: Stepping in place.
blue: left actual, green: left desired,
red: right actual, cyan: right desired.

Figure 14: Knee movement during stepping in place. Since we are close to actuator limits, sometimes the valve command becomes very large, and we get a smaller movement than we want (arrows). The mid-frequency oscillations (about 11Hz) in torque are unmodeled dynamics. What looks like high frequency noise is quite repeatable, and thus is also unmodeled dynamics. Robust policies have to deal with this.

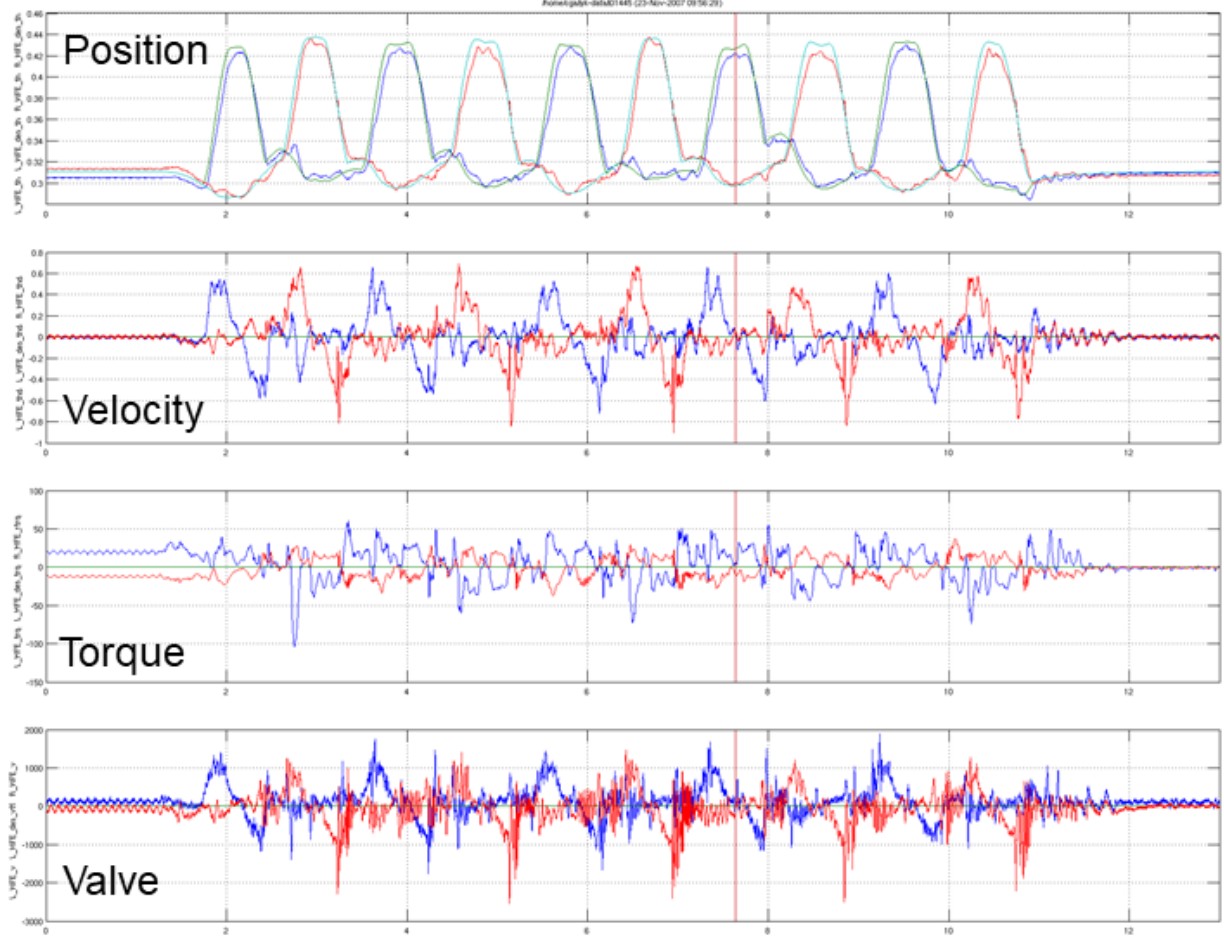

HFE: Stepping in place.
blue: left actual, green: left desired,
red: right actual, cyan: right desired.

Figure 15: Hip fore-aft movement during stepping in place. Everything is working well. Note how noisy the various signals are. What looks like high frequency noise is quite repeatable, and thus is also unmodeled dynamics. Robust policies have to deal with this.

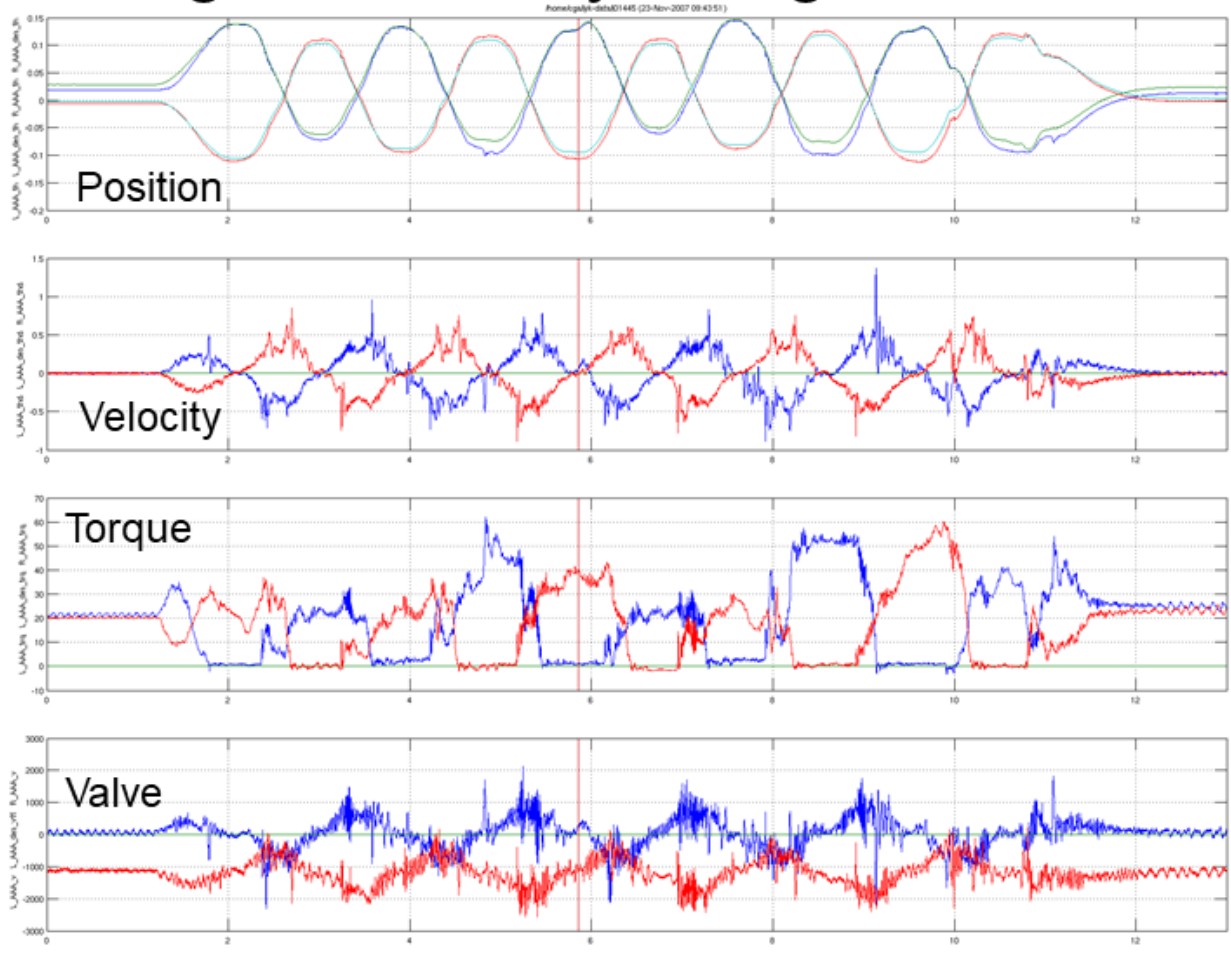

Figure 16: Ankle fore-aft movement during stepping in place. Note the torque saturation and how noisy the various signals are. What looks like high frequency noise is quite repeatable, and thus is also unmodeled dynamics. Robust policies have to deal with this.

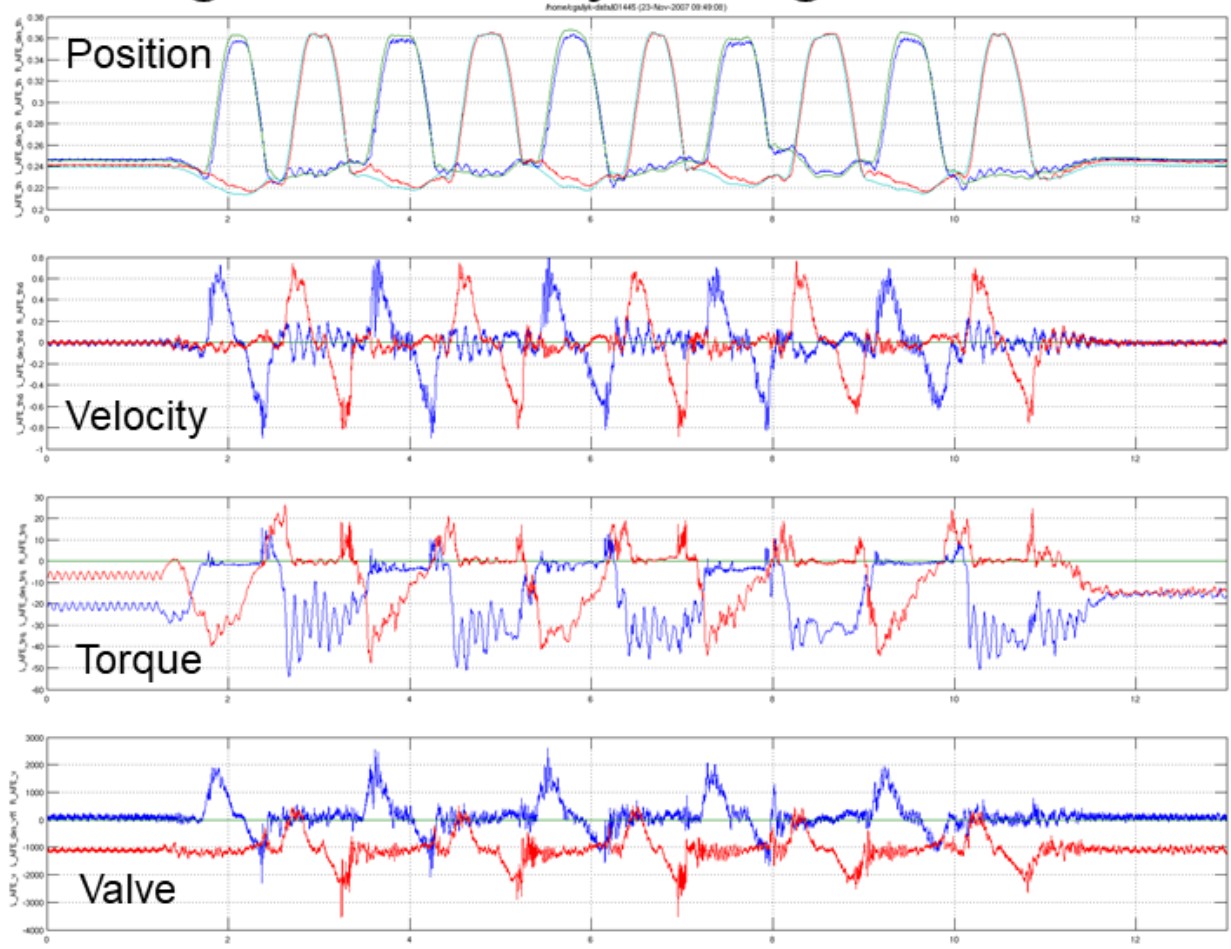

AAA: Stepping in place.
blue: left actual, green: left desired,
red: right actual, cyan: right desired.

Figure 17: Ankle sideways movement during stepping in place. Note the asymmetry in the torques and how noisy the various signals are. The mid-frequency oscillations (about 10Hz) in torque are unmodeled dynamics. What looks like high frequency noise is quite repeatable, and thus is also unmodeled dynamics. Robust policies have to deal with this.

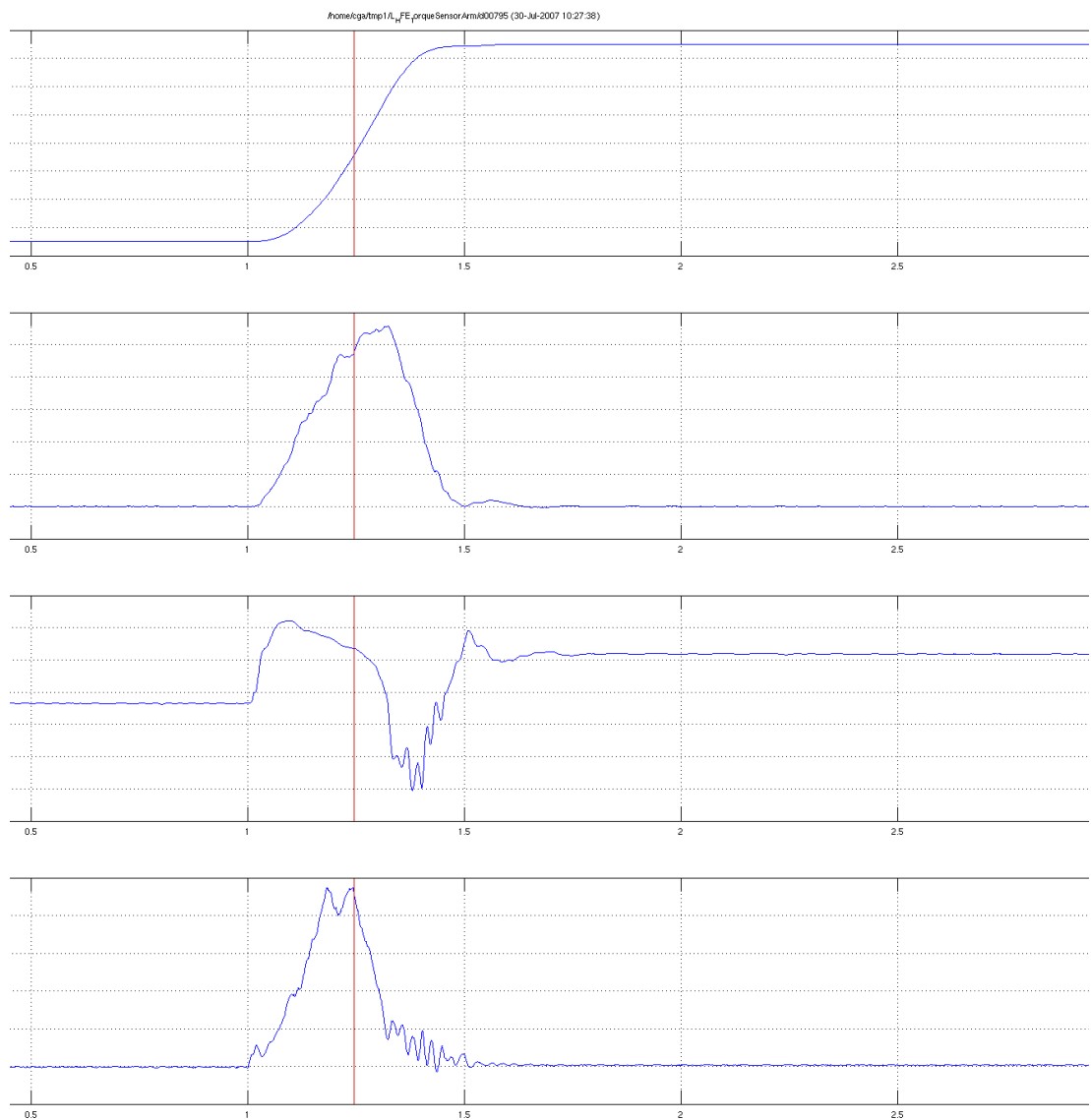

Figure 18: We use our error correction learning to get the robot joint to follow a smooth symmetric fifth order spline (a minimum jerk movement). The joint moves 1.4 radians in half a second, with a maximum velocity of 5.5 radians/second. The torque ranges from -150 to 110 Nm. The rows are (from the top) joint angle, joint velocity, joint torque, and valve command. Note how strange the torque profile and valve command are to achieve this simple movement. A slightly faster movement broke the robot, as you see on the next page.

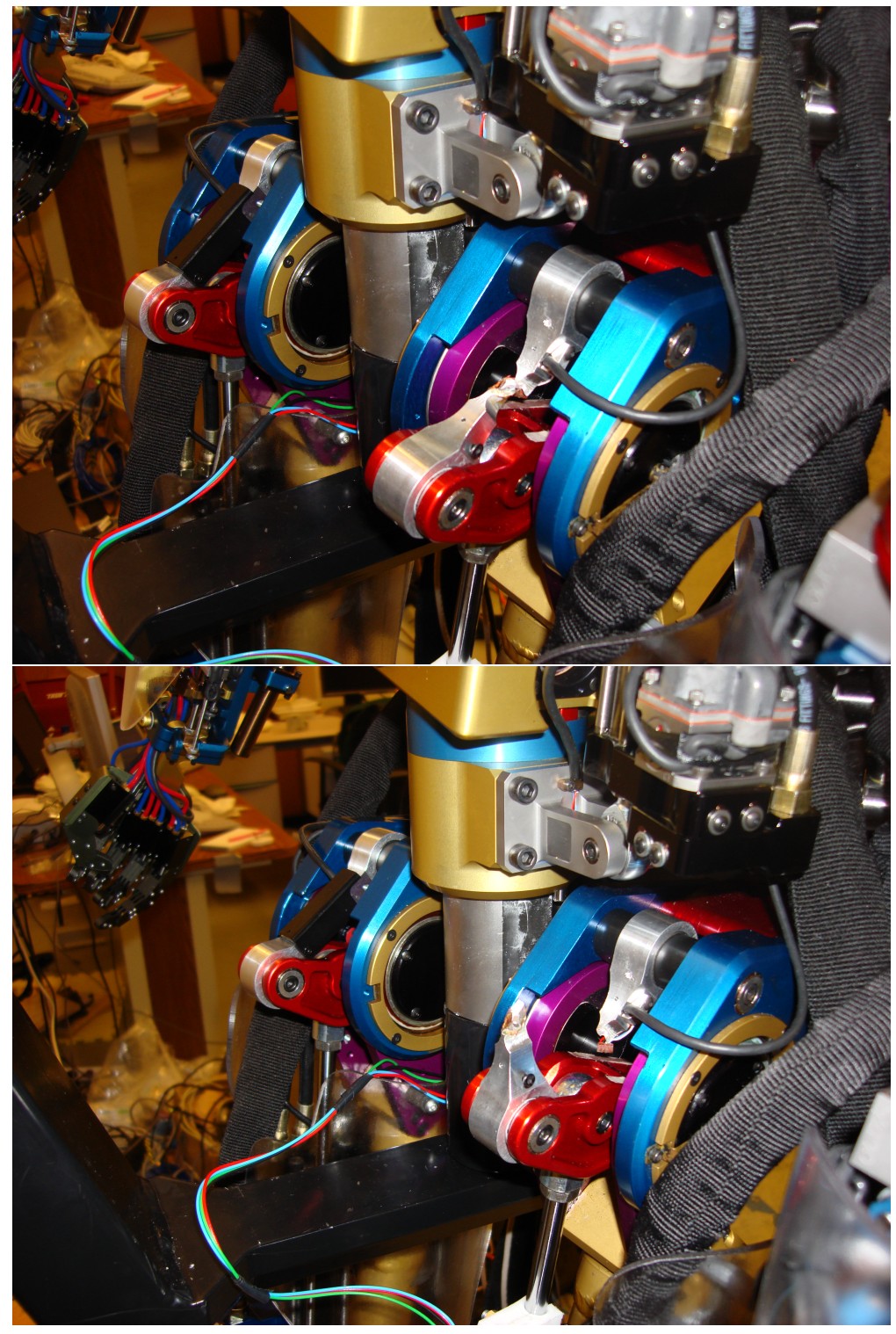

Figure 19: We broke the robot doing this modeling! These photos show that the left hip "tendon" ripped apart during a movement. Quite exciting! Leg flies in to the mechanical stop, and there is a loud crash. Scared the &#%@! out of us. That part needed to be beefed up.

# Appendix 3: Repeatability

This appendix will provide an example of how repeatable our Sarcos humanoid is. Data is taken from the modeling data collection we did to make the models like the one presented in the previous appendix, in this case for the left knee of the Sarcos Primus System Humanoid. We first show you one cycle of a sinusoidal movement (next page), and then we show you ten cycles plotted on top of each other to see which features of the curves are repeatable. **Quiz:** Before you turn to the page with the second figure, ask yourself which wiggles in the first figure do you think are repeatable, and which are noise. Then turn the page to the second figure and see. The last figure shows 10 seconds of the data the previous plots showed segments of.

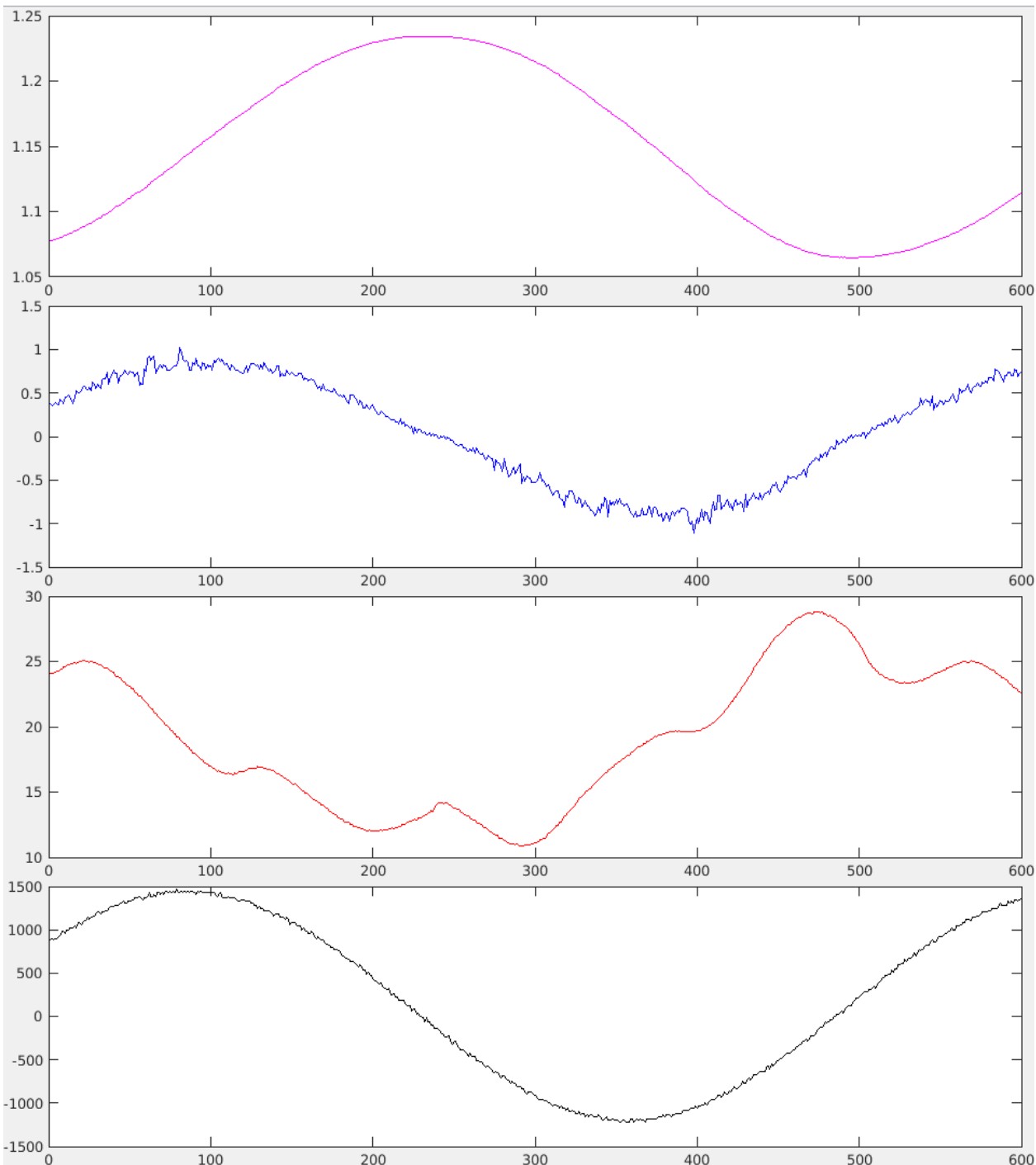

Figure 20: One cycle of a 1.83Hz sinusoidal movement of the left knee. 0.6s of data is shown, sampled at 1kHz. The traces from the top are left knee position (magenta), velocity (blue), torque (red), and valve command (black). We are interested in the velocity (blue) trace, where we see small high frequency noise due to the differentiation of the position sensor data, and also some slightly lower frequency larger features. Do you think they will be repeatable? The joint torque (red) sensor trace is much cleaner but has roughly 10Hz oscillations. Are they repeatable, or are they non-white noise in some range of frequencies? Before you turn the page, try to decide for yourself.

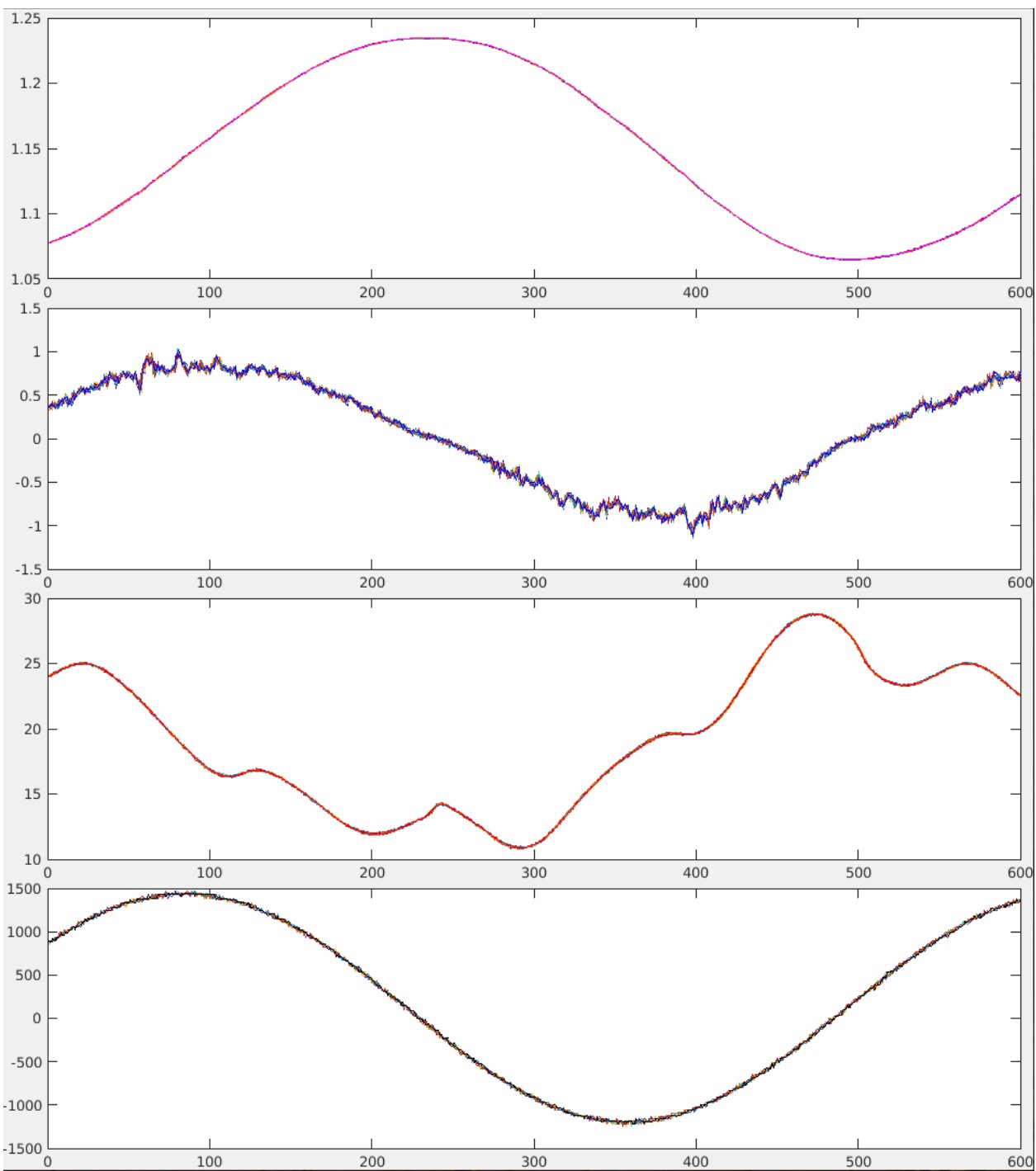

Figure 21: Ten cycles of a sinusoidal movement of the left knee plotted on top of each other to see how repeatable this robot is. The traces from the top are left knee position, velocity, torque, and valve command. It blows my mind how repeatable this robot is. The high frequency fuzz on the velocity is noise, but the slightly lower frequency bumps are repeatable. The torque oscillations are completely repeatable. Wow. The time alignment is limited to the integer sampling rate. The alignment could be better if we time-interpolated the data, but why bother. The point is already made.

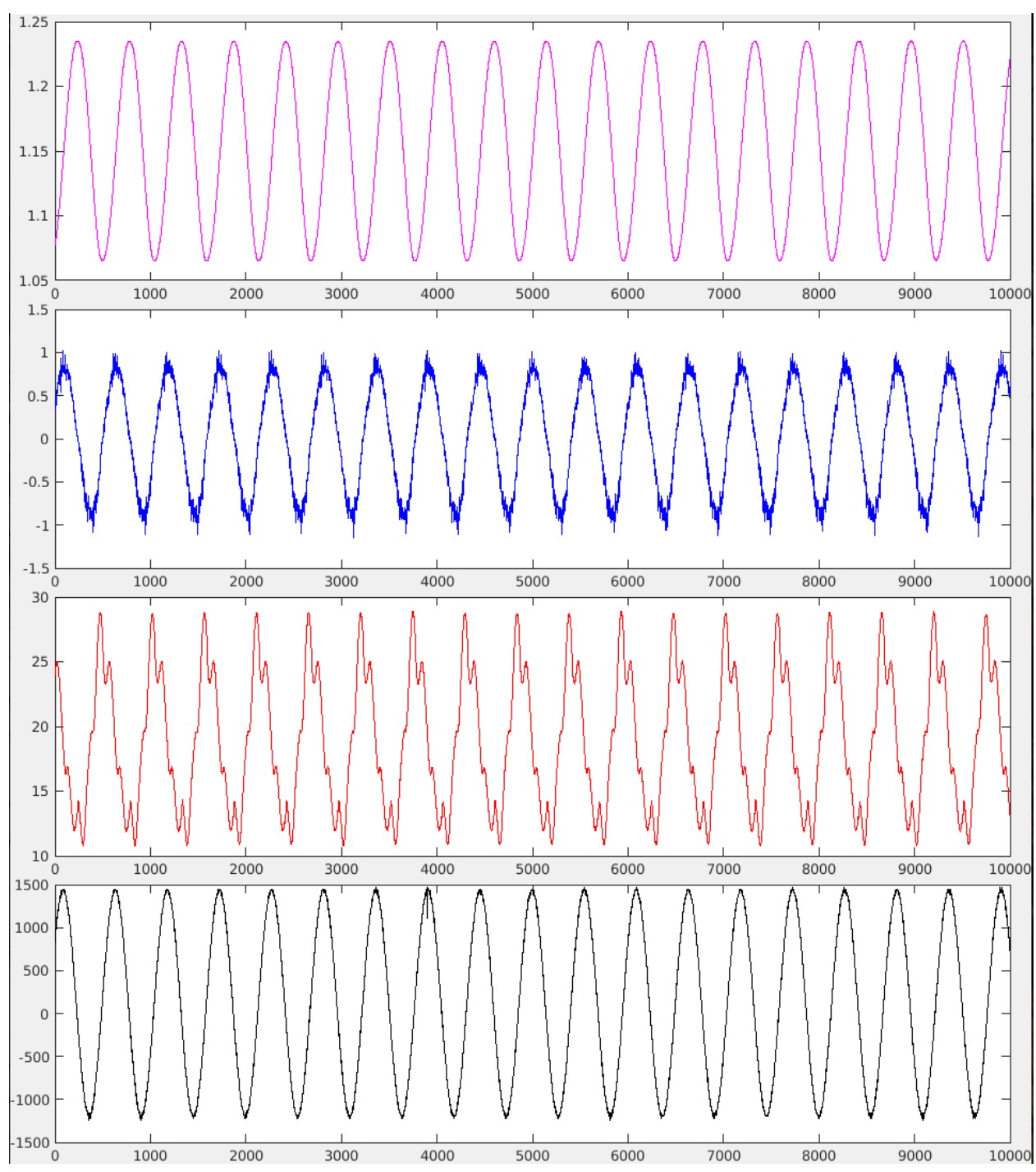

Figure 22: 10 seconds of the data the previous plots showed segments of. Like clockwork.

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

Wikipedia. Differential dynamic programming, 2020a. `en.wikipedia.org/wiki/Differential_dynamic_programming`.

Wikipedia. Fatigue (material), 2020b. `en.wikipedia.org/wiki/Fatigue_(material)`.

Wikipedia. H-infinity loop-shaping, 2020c. `en.wikipedia.org/wiki/H-infinity_loop-shaping`.

Wikipedia. Know nothing, 2020d. `en.wikipedia.org/wiki/Know_Nothing`.

Wikipedia. Thomas kuhn, 2020e. `en.wikipedia.org/wiki/Thomas_Kuhn`.

Wikipedia. Linear-quadratic regulator, 2020f. `en.wikipedia.org/wiki/Linear_quadratic_regulator`.

Wikipedia. Yann lecun, 2020g. `en.wikipedia.org/wiki/Yann_LeCun`.

Wikipedia. Quadratic form (statistics), 2020h. `en.wikipedia.org/wiki/Quadratic_form_(statistics)`.

Wikipedia. Tensor product, 2020i. `en.wikipedia.org/wiki/Tensor_product`.

Wikipedia. Vectorization (mathematics), 2020j. `en.wikipedia.org/wiki/Vectorization_(mathematics)`.

Wikipedia. Paul werbos, 2020k. `en.wikipedia.org/wiki/Paul_Werbos`.

Wikipedia. Chaos theory, 2020l. `en.wikipedia.org/wiki/Chaos_theory`.

Wikipedia. Intelligent design, 2020m. `en.wikipedia.org/wiki/Intelligent_design`.

Wikipedia. Dual control theory, 2020n. `en.wikipedia.org/wiki/Dual_control_theory`.

Wikipedia. End-to-end reinforcement learning, 2020o. `en.wikipedia.org/wiki/End-to-end_reinforcement_learning`.

Wikipedia. Fosbury flop, 2020p. `en.wikipedia.org/wiki/Fosbury_Flop`.

Wikipedia. Solved game, 2020q. `https://en.wikipedia.org/wiki/Solved_game`.

J Winn, C. M. Bishop, T. Diethe, J. Guiver, and Y. Zykov. *Model-Based Machine Learning*. Microsoft Research, 2020. `www.mbmlbook.com`.

M. Wisse, C. G. Atkeson, and D. K. Kloimwieder. Swing leg retraction helps biped walking stability. In *Proceedings of the 5th IEEE-RAS International Conference on Humanoid Robots (Humanoids)*, 2005.

Florentin Woergoetter and Bernd Porr. Reinforcement learning. *Scholarpedia*, 3(3):1448, 2008.

Walter Wonham. The internal model principle for linear multivariable regulators. *Applied Mathematics & Optimization*, 2:170–194, 01 1975.

Xinjilefu. *State Estimation for Humanoid Robots*. PhD thesis, Robotics Institute, Carnegie Mellon University, 2015. URL `www.cs.cmu.edu/~xxinjile/pdf/x_xinjilefu_robotics_2015.pdf`.

Xinjilefu and C.G. Atkeson. State estimation of a walking humanoid robot. In *Intelligent Robots and Systems (IROS), 2012 IEEE/RSJ International Conference on*, pages 3693–3699, 2012.

X Xinjilefu, Siyuan Feng, and Christopher G. Atkeson. Center of mass estimator for humanoids and its application in modelling error compensation, fall detection and prevention. In *IEEE-RAS International Conference on Humanoid Robots (Humanoids)*, 2015.

A. Yamaguchi and C. G. Atkeson. Combining finger vision and optical tactile sensing: Reducing and handling errors while cutting vegetables. In *IEEE-RAS International Conference on Humanoid Robotics*, 2016a.

Akihiko Yamaguchi and Christopher G. Atkeson. Differential dynamic programming with temporally decomposed dynamics. In *IEEE-RAS International Conference on Humanoid Robotics*, 2015a.

Akihiko Yamaguchi and Christopher G. Atkeson. A representation for general pouring behavior. In *in the Workshop on SPAR in the 2015 IEEE/RSJ International Conference on Intelligent Robots and Systems (IROS'15)*, 2015b.

Akihiko Yamaguchi and Christopher G. Atkeson. Stereo vision of liquid and particle flow for robot pouring. In *IEEE-RAS International Conference on Humanoid Robotics*, 2016b.

Akihiko Yamaguchi and Christopher G. Atkeson. Differential dynamic programming for graph-structured dynamical systems: Generalization of pouring behavior with different skills. In *IEEE-RAS International Conference on Humanoid Robotics*, 2016c.

Akihiko Yamaguchi and Christopher G. Atkeson. Neural networks and differential dynamic programming for reinforcement learning problems. In *the IEEE International Conference on Robotics and Automation (ICRA'16)*, 2016d.

Akihiko Yamaguchi and Christopher G. Atkeson. Recent progress in tactile sensing and sensors for robotic manipulation: can we turn tactile sensing into vision? *Advanced Robotics*, pages 1–13, 2019.

Akihiko Yamaguchi, Christopher G. Atkeson, Scott Niekum, and Tsukasa Ogasawara. Learning pouring skills from demonstration and practice. In *IEEE-RAS International Conference on Humanoid Robotics*, pages 908–915, Madrid, 2014.

Akihiko Yamaguchi, Christopher G. Atkeson, and Tsukasa Ogasawara. Pouring skills with planning and learning modeled from human demonstrations. *International Journal of Humanoid Robotics*, 12(3):1550030, 2015.

J. Zhang, P. Fiers, K. A. Witte, R. W. Jackson, K. L. Poggensee, C. G. Atkeson, and S. H. Collins. Human-in-the-loop optimization of exoskeleton assistance during walking. *Science*, 356(6344): 1280–1284, 2017.

Shan Zhong, Quan Liu, Zongzhang Zhang, and Qiming Fu. Efficient reinforcement learning in continuous state and action spaces with dyna and policy approximation. *Front. Comput. Sci.*, 13:106126, 2019.

M. Zucker, N. D. Ratliff, M. Stolle, J. E. Chestnutt, J. A. Bagnell, C. G. Atkeson, and J. Kuffner. Optimization and learning for rough terrain legged locomotion. *International Journal of Robotic Research*, 30(2):175–191, 2011.

