# OpenReview forum: "What advice would I give a starting graduate student interested in robot learning?"
_roboticsfoundation.org/RSS/2020/Workshop/RobRetro — RobRetro 2020_

### Official Review · AnonReviewer1 · 2020-06-24
**Are you looking for the history of robot learning? Here it is told by one of the pioneers in robot learning + advise and insights**

**Confidence:** 5
**Rating:** 9

**Review:**

The paper “What advice would I give a starting graduate student interested in robot learning? Models! “ is just really fun to read. While it is quite long (39 pages) it is full of useful insights, references and historical context, lessons learned and explicit advise to new graduate students. In that sense it is a true retrospective by one of the “old fogies [who] rise up and say “We already knew X, you are just using more computation.”” (Section 2.17 - Point 14). In addition, I also found myself often laughing out loud when reading for example Section 2.2 (Why I do what I do) or even about some academic failures (Section 2.4. at the very end). I’m looking forward to a discussion on this retrospective.

I only wish the paper was a bit more organised into history, insight, personal path etc.  These young people do not have sufficient attention span.

---

### Decision · Program_Chairs · 2020-06-25

Accept